# The NERP-4−SNAT2 axis regulates pancreatic β-cell maintenance and function

Weidong Zhang [1,2,16], Ayako Miura [2,11,16], Md Moin Abu Saleh [2,12], Koichiro Shimizu [2,13], Yuichiro Mita [2,14], Ryota Tanida [2,15], Satoshi Hirako [3], Seiji Shioda [4], Valery Gmyr [5], Julie Kerr-Conte [5], Francois Pattou [5], Chunhuan Jin [6], Yoshikatsu Kanai [6], Kazuki Sasaki [7], Naoto Minamino [8], Hideyuki Sakoda [1,2] & Masamitsu Nakazato [1,2,9,10] ✉

Insulin secretion from pancreatic β cells is regulated by multiple stimuli, including nutrients, hormones, neuronal inputs, and local signalling. Amino acids modulate insulin secretion via amino acid transporters expressed on β cells. The granin protein VGF has dual roles in β cells: regulating secretory granule formation and functioning as a multiple peptide precursor. A VGF-derived peptide, neuroendocrine regulatory peptide-4 (NERP-4), increases $Ca^{2+}$ influx in the pancreata of transgenic mice expressing apoaequorin, a $Ca^{2+}$-induced bioluminescent protein complex. NERP-4 enhances glucose-stimulated insulin secretion from isolated human and mouse islets and β-cell−derived MIN6-K8 cells. NERP-4 administration reverses the impairment of β-cell maintenance and function in *db/db* mice by enhancing mitochondrial function and reducing metabolic stress. NERP-4 acts on sodium-coupled neutral amino acid transporter 2 (SNAT2), thereby increasing glutamine, alanine, and proline uptake into β cells and stimulating insulin secretion. SNAT2 deletion and inhibition abolish the protective effects of NERP-4 on β-cell maintenance. These findings demonstrate a novel autocrine mechanism of β-cell maintenance and function that is mediated by the peptide−amino acid transporter axis.

Amino acids play pivotal roles not just in protein synthesis, but also in vital cellular processes, including hormone secretion and cell maintenance[1]. Postprandial elevations in plasma amino acids enhance insulin release from pancreatic β cells[2]. Amino acid transporters (AATs) are membrane-bound transport proteins that import or export amino acids (i.e., symporters and antiporters, respectively) to regulate cellular and circulating amino acid levels[3]. Several AATs are expressed in β cells, and stimulate postprandial insulin secretion by mediating the influx of amino acids[4,5]. After being transported into β cells, glutamine and alanine maintain these cells by activating ATP generation and by suppressing oxidative and endoplasmic reticulum (ER) stress[2,6,7]. The synthesis of AATs is regulated by various metabolic signals, such as amino acid availability and amino acid starvation[3,4]; however, no

endogenous substance that modulates the functional activity of AATs has been identified.

Altered intracellular $Ca^{2+}$ concentration is a common physiological response downstream of cell-surface receptors[8]. Aequorin, a 21-kDa protein purified from *luminous Aequorea*, is a $Ca^{2+}$-induced bioluminescent protein complex consisting of apoaequorin and coelenterazine as a luminous substrate[9]. Apoaequorin transgenic mice were generated to detect intracellular $Ca^{2+}$ mobilisation under the control of the CAG promoter[10]. Using the pancreata of these transgenic mice, we identified a novel 19−amino acid peptide from human neuroendocrine medullary thyroid carcinoma TT cells: NERP-4, corresponding to VGF 489−507[11,12]. VGF, a granin protein also named secretogranin VII, comprises 615 (human) or 617 (mouse/rat) amino acids and functions

in granule formation in neurons and endocrine cells, including β cells[13–16]. VGF also serves as a precursor of bioactive peptides that exhibit diverse biological activities, such as maintaining metabolic and glucose homeostasis[14,17–20].

Here we demonstrate that NERP-4 enhances glucose-stimulated insulin secretion (GSIS) by elevating intracellular $Ca^{2+}$ in β cells. NERP-4 administration reverses β-cell impairment by enhancing mitochondrial function and reducing oxidative and ER stress.    Using LRC-TriCEPS technology, binding assays, and cellular functional experiments, we establish NERP-4 binds to the sodium-coupled neutral AAT SNAT2, stimulates amino acid uptake, and improves β-cell maintenance and function. The present study demonstrates a critical role of the NERP-4–SNAT2 axis in β-cell biology.

## Results

### NERP-4 is co-localised with insulin in β cells, thus stimulating $Ca^{2+}$ influx

Aequorin-expressing tissues from apoaequorin transgenic mice were used to detect the elevation of intracellular $Ca^{2+}$ concentrations by ligands of interest, such as novel compounds and peptides[10]. We used pancreatic pieces because the islets would lose their original function once they were incubated with coelenterazine for 3 h. NERP-4 evoked transient luminescence in a piece of pancreatic tissue from apoaequorin transgenic mice, suggesting an increase in intracellular $Ca^{2+}$ level (Fig. 1b). NERP-4–induced $Ca^{2+}$ influx into β cells was investigated in MIN6-K8 cells, which are mouse insulinoma–derived β cells[21] that have been used to study the mechanism of glutamine-amplified $Ca^{2+}$ influx and insulin secretion[22,23]. NERP-4 enhanced $Ca^{2+}$ influx into MIN6-K8 cells under high glucose (Fig. 1c), but not under low glucose (Supplementary Fig. 1a). To explore whether the rise in $Ca^{2+}$ was due to extracellular $Ca^{2+}$ influx via $Ca^{2+}$ channels, we studied $Ca^{2+}$ influx into MIN6-K8 cells treated with EGTA (a calcium chelator) or nifedipine (an L-type $Ca^{2+}$ channel blocker). Both treatments suppressed NERP-4–induced $Ca^{2+}$ influx (Fig. 1d, e), implying that NERP-4 mediated $Ca^{2+}$ influx via $Ca^{2+}$ channels.

NERP-4 immunoreactivity was co-localised with insulin in C57BL/6 J mouse islets, rat islets, and MIN6-K8 cells (Fig. 1f–h). The specificity of an anti–NERP-4 antibody was confirmed in Vgf knockout (KO) mouse islets[24] (Fig. 1i). Immunogold electron microscopy revealed co-localisation of NERP-4 with insulin in storage granules (Fig. 1j). NERP-4 was infrequent in cells producing glucagon or somatostatin in C57BL/6 J mouse islets (Supplementary Fig. 1b). Immunoreactive NERP-4 was identified in human pancreatic extracts by reverse-phase high-performance liquid chromatography (RP-HPLC). NERP-4 immunoreactivity measured by radioimmunoassay (RIA) was eluted at the same position as synthetic human NERP-4 (Fig. 1k). In C57BL/6 J mouse islets, Vgf mRNA levels were 2–3 orders of magnitude lower than those of Ins1 and Ins2 (encoding insulin 1 and insulin 2, respectively) (Supplementary Fig. 1c). These results indicate that NERP-4 is produced in β cells and co-localises with insulin. NERP-4 increases $Ca^{2+}$ influx into β cells via $Ca^{2+}$ channels.

### NERP-4 stimulates insulin secretion

NERP-4 augmented insulin secretion in human and C57BL/6 J mouse islets and MIN6-K8 cells under high glucose, but not under low glucose (Fig. 2a–d). Co-administration of NERP-4 and NERP-2[VGF 310–347], another VGF-derived insulinotropic peptide, to Vgf knockdown MIN6-K8 cells additively increased GSIS (Supplementary Fig. 2a, b). In MIN6-K8 cells, NERP-4 enhanced the glucose-induced elevation of ATP concentration, but not that of intracellular cAMP concentration (Fig. 2e, f). Both EGTA and nifedipine treatments abolished NERP-4–induced GSIS in MIN6-K8 cells (Fig. 2g, h). Given that NERP-4 co-localised with insulin and directly enhanced GSIS by β cells, we investigated whether NERP-4 was secreted from β cells. High glucose stimulated both NERP-4 and insulin secretion (Fig. 2i, j). Anti–NERP-4

IgG administration to C57BL/6 J mouse islets suppressed GSIS below that observed with normal rabbit serum IgG (NRS IgG) (Fig. 2k). Thus, NERP-4 stimulates insulin secretion by enhancing $Ca^{2+}$ influx into β cells in an autocrine fashion.

### NERP-4 prevents oxidative and ER stress and mitochondrial dysfunction induced by palmitate and cytokines

Lipotoxicity impairs β-cell function by inducing oxidative and ER stress in diabetes[25]. Because two other insulinotropic peptides, TLQP-21[VGF 556–576] and glucagon-like peptide-1 (GLP-1), protected β cells from lipotoxicity[19,26], we investigated the roles of NERP-4 in β-cell maintenance in isolated C57BL/6 J mouse islets. Palmitate is a saturated fatty acid widely used to cause insulin secretory dysfunction, induction of ER stress, and apoptosis. Palmitate reduced GSIS and the mRNA levels of Ins1 and Ins2 (Fig. 3a–d). NERP-4 reversed these alterations (Fig. 3a–d). Palmitate increased the Vgf mRNA level and NERP-4 elevated it to an even greater extent (Fig. 3e). Palmitate increased the protein and mRNA levels of the ER stress marker CHOP, while NERP-4 reduced them (Fig. 3f, Supplementary Fig. 3). Palmitate reduced the protein and mRNA levels of the antioxidative response marker SOD2, while NERP-4 reversed these alterations (Fig. 3f, Supplementary Fig. 3). Nrf2 responds to oxidative stress by translocating from the cytosol to the nucleus and then inducing antioxidant genes[27]. Palmitate reduced Nrf2 translocation to the nucleus and Nrf2 mRNA level (Fig. 3g, Supplementary Fig. 3). NERP-4 reversed these alterations (Fig. 3g, Supplementary Fig. 3). Under non-stressed conditions, NERP-4 administration to isolated C57BL/6 J mouse islets increased the mRNA levels of Vgf, Sod2, and Nrf2 (Fig. 3e, Supplementary Fig. 3). NERP-4 also decreased the protein and mRNA levels of CHOP (Fig. 3f, Supplementary Fig. 3), increased the SOD2 protein level (Fig. 3f), and enhanced Nrf2 nuclear translocation (Fig. 3g).

To investigate the role of endogenous NERP-4 in β-cell maintenance, we administered NERP-4 IgG to C57BL/6 J mouse islets for 3 days. NERP-4 neutralisation reduced GSIS and the mRNA levels of Ins1, Ins2, Sod2, and Nrf2, and increased the Chop mRNA level relative to those seen following 3-day NRS IgG administration (Fig. 3h–m). These results suggest that NERP-4 protects β cells from metabolic stress in an autocrine fashion.

Glucose oxidation and ATP production in mitochondria are necessary for insulin biosynthesis, transport, and secretion[28,29]. Mitochondrial dysfunction causes β-cell failure and the development of diabetes[28,29]. We next studied the role of NERP-4 in mitochondrial function in MIN6-K8 cells. NERP-4 increased the mitochondrial oxygen consumption rate (OCR) in palmitate-treated MIN6-K8 cells (Fig. 4a, b). Palmitate reduced ATP content, the mRNA levels of the ATP synthases Atp5e and Atp5j2, and GSIS in MIN6-K8 cells (Fig. 4c–e, Supplementary Fig. 4a), while NERP-4 reversed these reductions (Fig. 4c–e, Supplementary Fig. 4a). Palmitate increased the Vgf mRNA level while NERP-4 further elevated it (Fig. 4f). NERP-4 suppressed the palmitate-induced production of reactive oxygen species (ROS) (Fig. 4g). NERP-4 administration to naïve MIN6-K8 cells also increased ATP content, the mRNA levels of Atp5e and Atp5j2, and cell viability, and decreased ROS production (Fig. 4c–e, g, j).

We also investigated the roles of NERP-4 in a model of cytokine-induced β-cell impairment. Interleukin-1β was previously shown to induce oxidative and ER stress in β cells[30]. A cytokine cocktail (tumour necrosis factor-α, interleukin-1β, and interferon-γ) upregulated the apoptotic marker protein cleaved caspase-3 and its mRNA, reduced GSIS and cell viability, and induced cell death in MIN6-K8 cells (Fig. 4h–k, Supplementary Fig. 4b). NERP-4 reversed these alterations. A cytokine cocktail increased the Vgf mRNA level and NERP-4 further elevated it (Supplementary Fig. 4b). Collectively, these data indicate that NERP-4 suppresses oxidative and ER stress, thereby improving β-cell mitochondrial function.

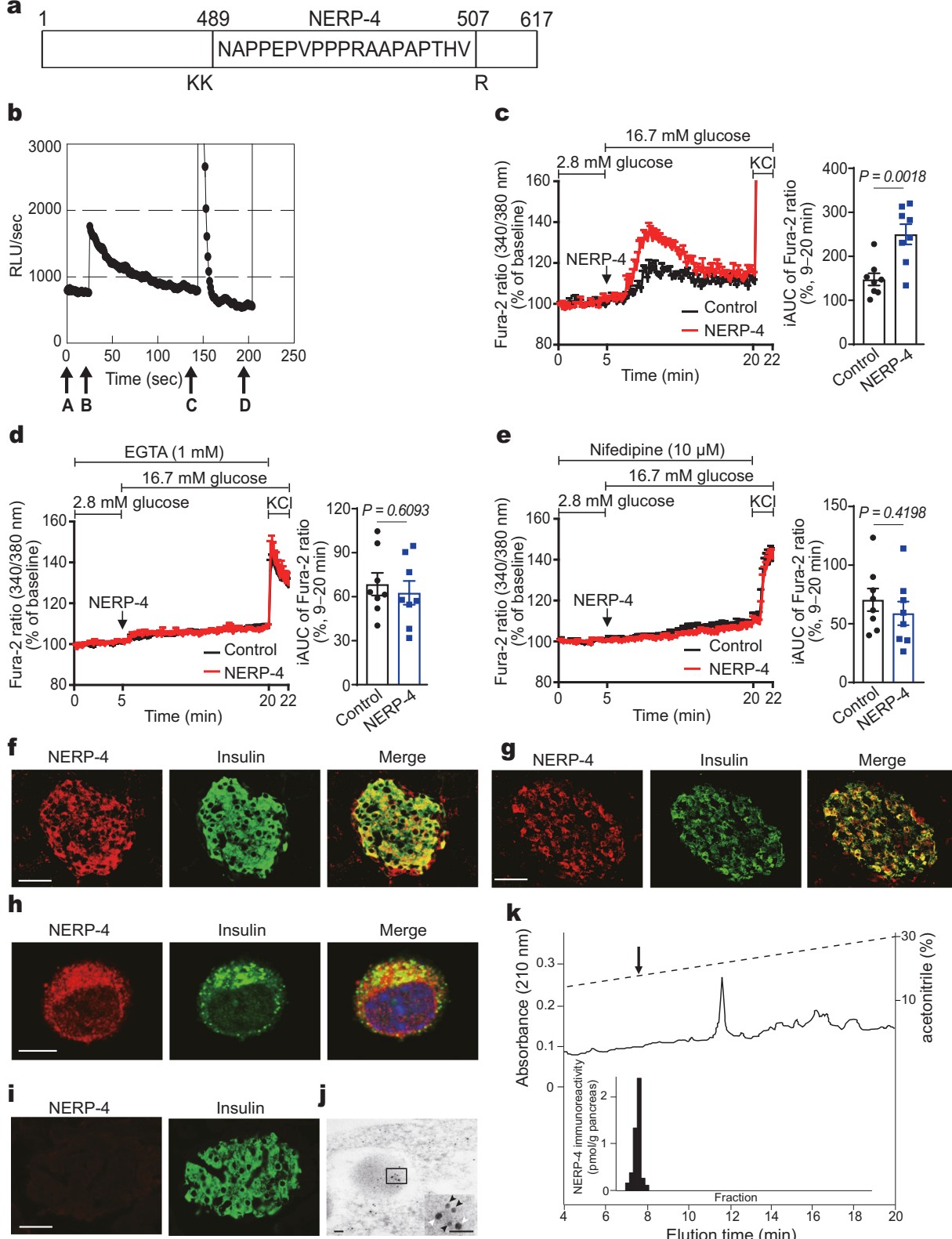

## NERP-4 reverses β-cell impairment in db/db mice

We next studied how NERP-4 impacted glucose metabolism and β-cell maintenance in obese, diabetic *db/db* mice. Insulin and NERP-4 immunoreactivities were reduced in *db/db* mice compared with *db/+* mice (Fig. 5a, b). The mRNA levels of *Vgf*, *Ins1*, and *Ins2* were lower in *db/db* mice compared with *db/+* mice (Fig. 5c). Two-week NERP-4 administration to *db/db* mice decreased fasting blood glucose and

plasma insulin levels compared with saline treatment, but did not change body weight or food intake (Fig. 6a, b, Supplementary Fig. 5a, b). Two weeks after NERP-4 administration, a glucose tolerance test (GTT) demonstrated lowered blood glucose and elevated plasma insulin (Fig. 6c, d), but an insulin tolerance test (ITT) showed no alteration in insulin sensitivity (Supplementary Fig. 5c). NERP-4 increased the number and mean diameter of insulin granules in β

**Fig. 1 | NERP-4 induces intracellular Ca²⁺ mobilisation. a** NERP-4 sequence in mouse VGF. NERP-4 is cleaved from VGF after a basic pair (lysine−lysine: KK) and before a single basic amino acid (arginine: R). **b** Representative profile of the relative luminescence evoked in the apoaequorin transgenic mouse pancreas. Arrows (A−D) represent the tested substances. Medium (A), 1 µM NERP-4 (B), 100 µM ATP (C), and 2.5% Triton X-100 (D) were administered before the time indicated by the corresponding arrow. ATP was used as a positive control. **c−e** Representative Fura-2-AM ratios in MIN6-K8 cells in response to NERP-4 with or without EGTA or nifedipine treatment (n = 8 cells), and average incremental AUC (iAUC) (9−20 min) of [Ca²⁺]ᵢ (n = 8 cells). Representative immunofluorescence images of NERP-4, insulin, and their merged images in C57BL/6 J mouse islet (**f**), rat islet (**g**), MIN6-K8 cell (**h**), and *Vgf* KO mouse islet (**i**). **j** A representative immunoelectron micrograph showing co-localisation of NERP-4 with insulin in C57BL/6 J mouse islets. Inset is a higher-magnification image (NERP-4: 5-nm gold particles; black arrowheads: insulin; 10-nm gold particles; white arrowheads). **k** RP-HPLC of immunoreactive NERP-4 extracted from human pancreas. The arrow indicates the elution position of synthetic human NERP-4. Results are representative of three (**c−e**) or two (**f−j**) independent experiments. Data are mean ± s.e.m (**c−e**). Unpaired two-tailed Student's *t*-test (**c−e**). Scale bars, 50 µm (**f, g, i**), 5 µm (**h**), 50 nm (**j**). Source data are provided as a Source data file.

cells of *db/db* mice, as observed by transmission electron microscopy (TEM) (Fig. 6e, f). NERP-4 also increased the following in *db/db* islets: *Ins1, Ins2, Iapp, Vgf*, and *Pdx-1* mRNA levels; insulin content; Ki67-positive cell number; and insulin staining intensity (Fig. 6g−o). NERP-4 did not change the *Gcg* mRNA level, plasma glucagon level, or the area of glucagon immunoreactivity (Supplementary Fig. 5d−f). Mitochondrial dynamics, such as fission and fusion, comprise a highly coordinated cycle that maintains the shape, distribution, and size of mitochondria and play a critical role in the regulation of pancreatic β-cell function[28,29]. Mitophagy acts to selectively degrade dysfunctional mitochondria[31,32]. NERP-4 increased the numbers of mitochondria and mitophagosomes and decreased mitochondrial size in β cells of *db/db* mice (Fig. 7a−d). NERP-4 upregulated the mRNA levels of mitochondrial markers of biogenesis (*Pgc1α*), fission (*Drp1*), fusion (*Mfn1*), and mitophagy (*Park2* and *Pink1*) (Fig. 7e). NERP-4 reduced the protein and mRNA levels of CHOP (Fig. 7f, Supplementary Fig. 6), and increased the protein and mRNA levels of SOD2 and the mRNA level of *Nrf2* (Fig. 7f, Supplementary Fig. 6). These results suggest that NERP-4 maintains β-cell function in *db/db* mice by improving mitochondrial dynamics and reducing oxidative and ER stress.

## SNAT2 is a target protein for NERP-4

To identify NERP-4 target candidates in MIN6-K8 cells, we used the ligand−receptor glycocapture technique LRC-TriCEPS[33], a highly specific and sensitive pull-down assay involving binding to glycosylated membrane proteins. The volcano plot in Fig. 8a compares proteins bound by NERP-4 and NERP-2. Only SNAT2 and lysosome membrane protein 2 (SCRB2) satisfied the criteria for specific binding to TriCEPS (Fig. 8b). We detected NERP-4 binding in the cell membrane protein fraction of SNAT2-overexpressing HEK293 cells with a radioisotope-labelled C-terminally tyrosyl fragment of NERP-4[8–19], [¹²⁵I]-Y-NERP-4[8–19], which was used in the RIA of NERP-4 (Fig. 8c). SCRB2 is a type III glycoprotein located primarily in limiting membranes of lysosomes and endosomes[34]. SCRB2 participates in membrane transportation and the reorganization of the endosomal/lysosomal compartment[34]. Our findings suggest that SCRB2 is not the target protein of NERP-4. AATs expressed on the cell membrane of β cells regulate β-cell behaviour by mediating amino acid uptake and release[3,4]. We therefore investigated SNAT2, a neutral AAT that enhances amino acid uptake into β cells to stimulate insulin secretion[5]. We studied the binding of NERP-4 to SNAT2. [¹²⁵I]-Y-NERP-4[8–19] bound to SNAT2-overexpressing HEK293 cells in a concentration-dependent manner (Fig. 8d, Supplementary Fig. 7). [¹²⁵I]-Y-NERP-4[8–19] binding was reduced by the addition of excessive NERP-4 (Fig. 8d), suggesting that the binding was specific for NERP-4. [¹²⁵I]-Y-NERP-4[8–19] binding was detected in the membrane fraction of SNAT2-overexpressing HEK293 cells (Fig. 8d). SNAT1, SNAT2, and SNAT4 are system A AATs that facilitate the uptake of neutral amino acids[34]. We detected the mRNAs of *Snat2, Snat3, Snat4*, and *Snat5*, but not *Snat1*, in C57BL/6 J mouse islets and MIN6-K8 cells, consistent with a previous study[22] (Fig. 8e). NERP-4 binding was detected only in SNAT2-overexpressing HEK293 cells, but not in HEK293 cells expressing other SNATs (Fig. 8f).

## NERP-4 increases amino acid uptake and GSIS via SNAT2

We next investigated the effect of NERP-4 on amino acid transport activity. Human and C57BL/6 J mouse islets and MIN6-K8 cells were administered NERP-4 with 1 µM [¹⁴C]-L-glutamine. NERP-4 increased the [¹⁴C]-L-glutamine uptake in both islets and MIN6-K8 cells (Fig. 9a−c). Next, MIN6-K8 cells were administered NERP-4 with 1 µM [¹⁴C]-L-alanine or 1 µM [³H]-L-proline. NERP-4 also increased their uptake (Fig. 9d, Supplementary Fig. 8a). NERP-4 increased the intracellular contents of glutamine, alanine, proline, glycine, and glutamic acid in MIN6-K8 cells (Supplementary Fig. 8b, Supplementary Table 1). Among system A transporters, proline is the preferential natural substrate of SNAT2, but not SNAT1 or SNAT4[35]. To demonstrate that the effect of NERP-4 depends only on SNAT2, we performed SNAT2 deletion and inhibitor experiments. *Snat2* knockdown in MIN6-K8 cells abrogated NERP-4−induced [¹⁴C]-L-glutamine and [¹⁴C]-L-alanine uptake (Fig. 9e, f, Supplementary Fig. 8c). α-Methylaminoisobutyric acid (MeAIB), a non-metabolisable L-alanine analogue, is an orthosteric inhibitor of system A transporter[36]. MeAIB also abrogated NERP-4−induced [¹⁴C]-L-glutamine and [¹⁴C]-L-alanine uptake (Fig. 9g, h). These results imply that NERP-4 acts only on SNAT2 to stimulate amino acid uptake. Glutamine and alanine are imported via several transporters in many types of cells[5]. In rat islets, SNAT2, SNAT4, ASC-1 (SLC1A10), ASCT2 (SLC1A5), and B⁰AT1 (SLC6A19) transport glutamine and alanine[5]. *Snat2* knockdown reduced glutamine and alanine uptake by 13.6% and 30.3%, respectively, compared with siSCR treatment (Fig. 9e, f). MeAIB reduced glutamine and alanine uptake by 17.8% and 40.7%, respectively (Fig. 9g, h). These results suggest that transporters other than SNAT2 also mediate glutamine and alanine uptake in β cells.

Given that NERP-4 increased GSIS by stimulating amino acid uptake, we assumed that β-cell exposure to higher glutamine concentrations could mitigate the effects of NERP-4 on GSIS. We studied the insulinotropic activity of NERP-4 in MIN6-K8 cells at glutamine concentrations from 1 µM to 1 mM in amino acid−free HEPES Krebs−Ringer bicarbonate (HKRB) buffer. Glutamine concentrations equal to or higher than 1 µM increased GSIS in a concentration-dependent manner (Supplementary Fig. 8d). NERP-4 augmented GSIS at glutamine concentrations between 1 µM and 100 µM, but not at 1 mM glutamine (Supplementary Fig. 8d). Both *Snat2* knockdown and MeAIB abrogated NERP-4−induced Ca²⁺ influx (Fig. 9i, Supplementary Fig. 8e, f) and GSIS (Fig. 9j, k) in MIN6-K8 cells. MeAIB also abolished NERP-4−induced GSIS in human and C57BL/6 J mouse islets (Fig. 9l, m). These findings reveal that NERP-4 acts specifically on SNAT2 to induce amino acid uptake, Ca²⁺ influx, and GSIS.

We used amino acid−free HKRB buffer in GSIS experiments. SNAT3, SNAT5, and ASCT2 function as exporters of glutamine under conditions in which extracellular glutamine is absent or very low[37,38]. To explore the source of amino acids that stimulate NERP-4−induced GSIS, we quantified amino acids released into the medium after GSIS experiments. MIN6-K8 cells released glutamine, alanine, proline, and other amino acids under both 2.8 mM and 16.7 mM glucose conditions (Supplementary Table 2). The concentrations of these amino acids were comparable between low and high glucose conditions. NERP-4 did not augment Ca²⁺ influx or GSIS under 2.8 mM glucose even when amino acids were present in the medium (Fig. 2b, c, Supplementary

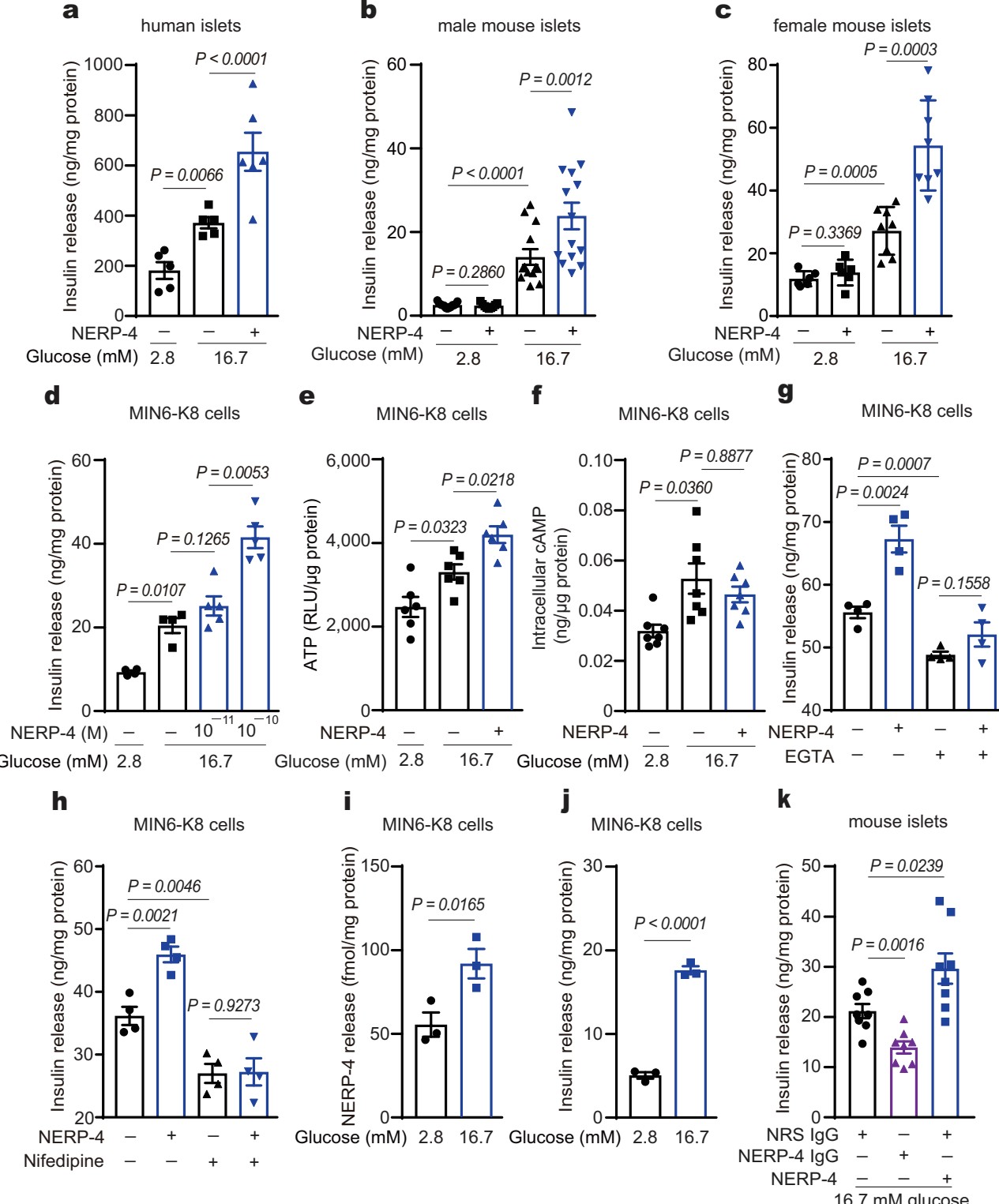

**Fig. 2 | NERP-4 induces insulin secretion.** NERP-4 enhances GSIS in human islets (**a**, *n* = 5, 5, 6 biological replicates), islets from male (**b**, *n* = 13, 9, 13, 14 biological replicates) and female (**c**, *n* = 6, 6, 8, 8 biological replicates) C57BL/6 J mice, and MIN6-K8 cells (**d**, *n* = 4, 4, 5, 5 biological replicates). **e** ATP production in MIN6-K8 cells (*n* = 6 biological replicates). **f** Intracellular cAMP level in MIN6-K8 cells incubated with NERP-4 (*n* = 7 biological replicates). GSIS from MIN6-K8 cells treated with EGTA (**g**) or nifedipine (**h**) (*n* = 4 biological replicates). Glucose-induced secretion of NERP-4 (**i**) and insulin (**j**) from MIN6-K8 cells (*n* = 3 biological replicates). **k** Anti−NERP-4 IgG suppression of insulin secretion in C57BL/6 J mouse islets (*n* = 8 biological replicates). Results are pooled from three (**b**) or two (**c**, **f**, **k**) independent experiments, or are representative of three (**a**, **d**) or two (**g**−**j**) independent experiments. Data are mean ± s.e.m (**a**−**k**). One-way ANOVA and Tukey's multiple comparisons test (**a**−**h**, **k**). Unpaired two-tailed Student's *t*-test (**i**, **j**). Source data are provided as a Source data file.

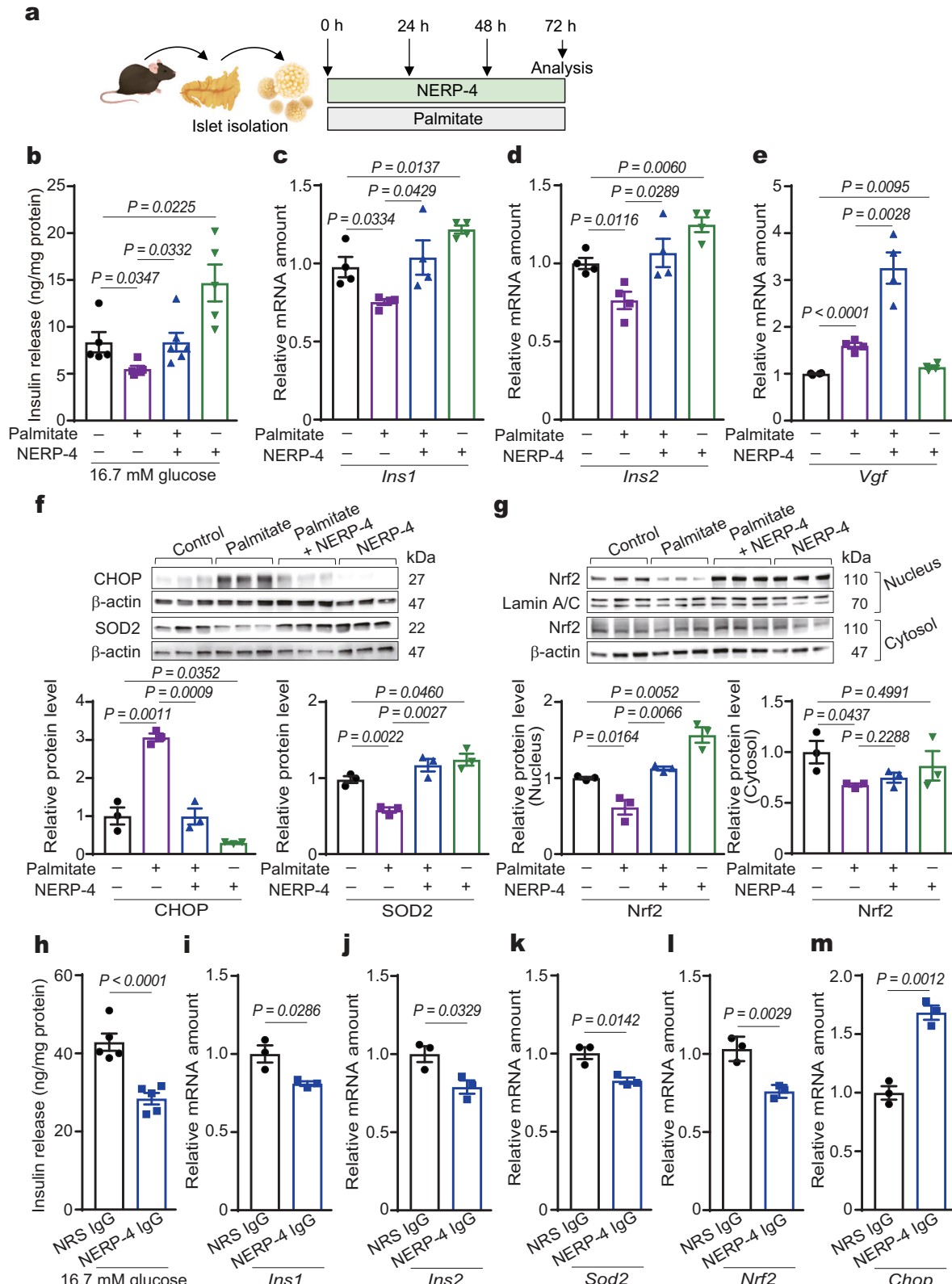

**Fig. 3 | NERP-4 reverses palmitate-induced β-cell dysfunction and NERP-4 IgG induces β-cell dysfunction in mouse islets. a** NERP-4 was administered at 0, 24, and 48 h to isolated C57BL/6 J mouse islets under palmitate. **b** GSIS from C57BL/6 J mouse islets (*n* = 5 biological replicates). **c–e** *Ins1, Ins2,* and *Vgf* mRNA amounts (*n* = 4 biological replicates). **f** Protein levels of CHOP and SOD2 (*n* = 3 biological replicates). **g** Nrf2 protein levels in nuclear and cytosolic fractions (*n* = 3 biological replicates). **h–m** C57BL/6 J mouse islets treated for 72 h with NRS IgG or NERP-4 IgG. **h** GSIS from C57BL/6 J mouse islets (*n* = 5 biological replicates). **i–m** *Ins1, Ins2, Sod2, Nrf2,* and *Chop* mRNA amounts (*n* = 3 biological replicates). Representative results of two independent experiments (**b–m**). Data are mean ± s.e.m (**b–m**). One-way ANOVA and Tukey's multiple comparisons test (**b–g**). Unpaired two-tailed Student's *t*-test (**h–m**). Source data are provided as a Source data file.

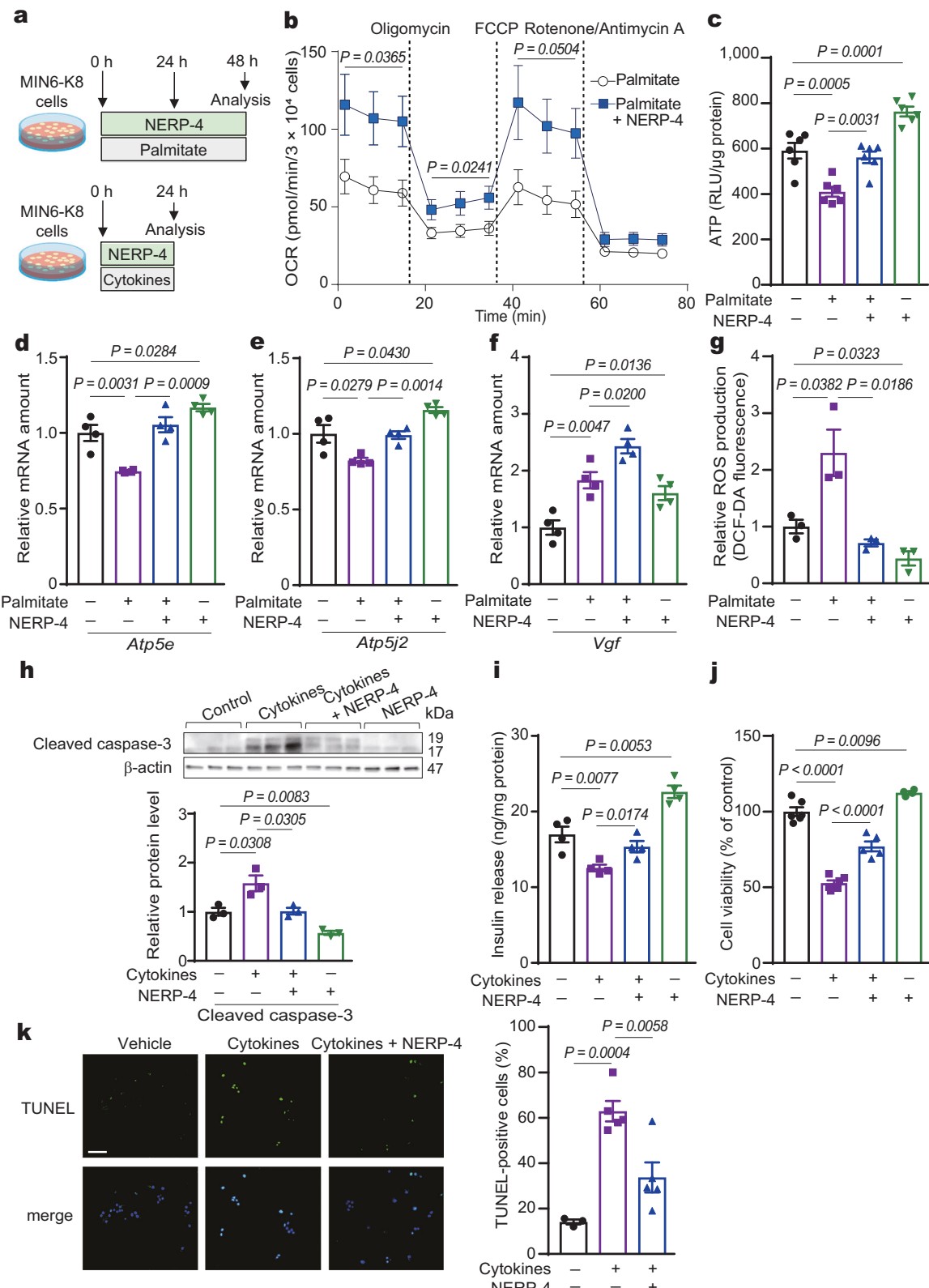

Fig. 1a). Another research group demonstrated that 2 mM glutamine administration to MIN6-K8 cells in medium containing 2.8 mM glucose did not affect GSIS or Ca²⁺ influx[23]. Together, these findings indicate that under high glucose conditions, NERP-4 functions as an insulino-tropic peptide by enhancing uptake of amino acids released from β cells.

NERP-4 augmented [$^{14}$C]-L-alanine uptake 1 min after its adminis-tration to MIN6-K8 cells (Supplementary Fig. 8g). To study the kinetic properties of SNAT2 activation by NERP-4, we determined the concentration-dependent uptake of [$^{14}$C]-MeAIB in the range of 1 μM to 2 mM at 2 min in the presence or absence of NERP-4. [$^{14}$C]-MeAIB uptake showed a good fit to a Michaelis–Menten curve, with a Km of

**Fig. 4 | NERP-4 reverses palmitate- or cytokine-induced β-cell dysfunction in MIN6-K8 cells. a** NERP-4 was administered at 0 and 24 h to MIN6-K8 cells under palmitate (**b**–**g**) or at 0 h under cytokines (**h**–**k**). **b** OCR analyses of MIN6-K8 cells (*n* = 3 biological replicates). The *P* values indicated the differences of basal respiration (1.6–14.8 min), ATP production (21.4–34.6 min), and maximal respiration (41.2–54.4 min) in palmitate-treated MIN6-K8 cells with or without NERP-4. **c** ATP production in MIN6-K8 cells treated with or without palmitate and NERP-4 (*n* = 6 biological replicates). *AtpSe* (**d**), *AtpSj2* (**e**), and *Vgf* (**f**) mRNA amounts in MIN6-K8 cells (*n* = 4 biological replicates). **g** ROS production in MIN6-K8 cells (*n* = 3

biological replicates). **h**–**k** MIN6-K8 cells were treated with a cytokine cocktail and NERP-4 for 24 h. **h** Protein level of cleaved caspase-3 (*n* = 3 biological replicates). **i** GSIS from MIN6-K8 cells (*n* = 4 biological replicates). **j** Cell viability (*n* = 6, 6, 5, 4 biological replicates). **k** Representative TUNEL images and per cent ratio of TUNEL-positive cells (green) to DAPI-positive cells (blue) (*n* = 5 independent samples). Representative results of two independent experiments (**b**–**k**). Data are mean ± s.e.m. (**b**–**k**). Unpaired two-tailed Student's *t*-test (**b**). One-way ANOVA and Tukey's multiple comparisons test (**c**–**k**). Scale bar, 100 μm (**k**). Source data are provided as a Source data file.

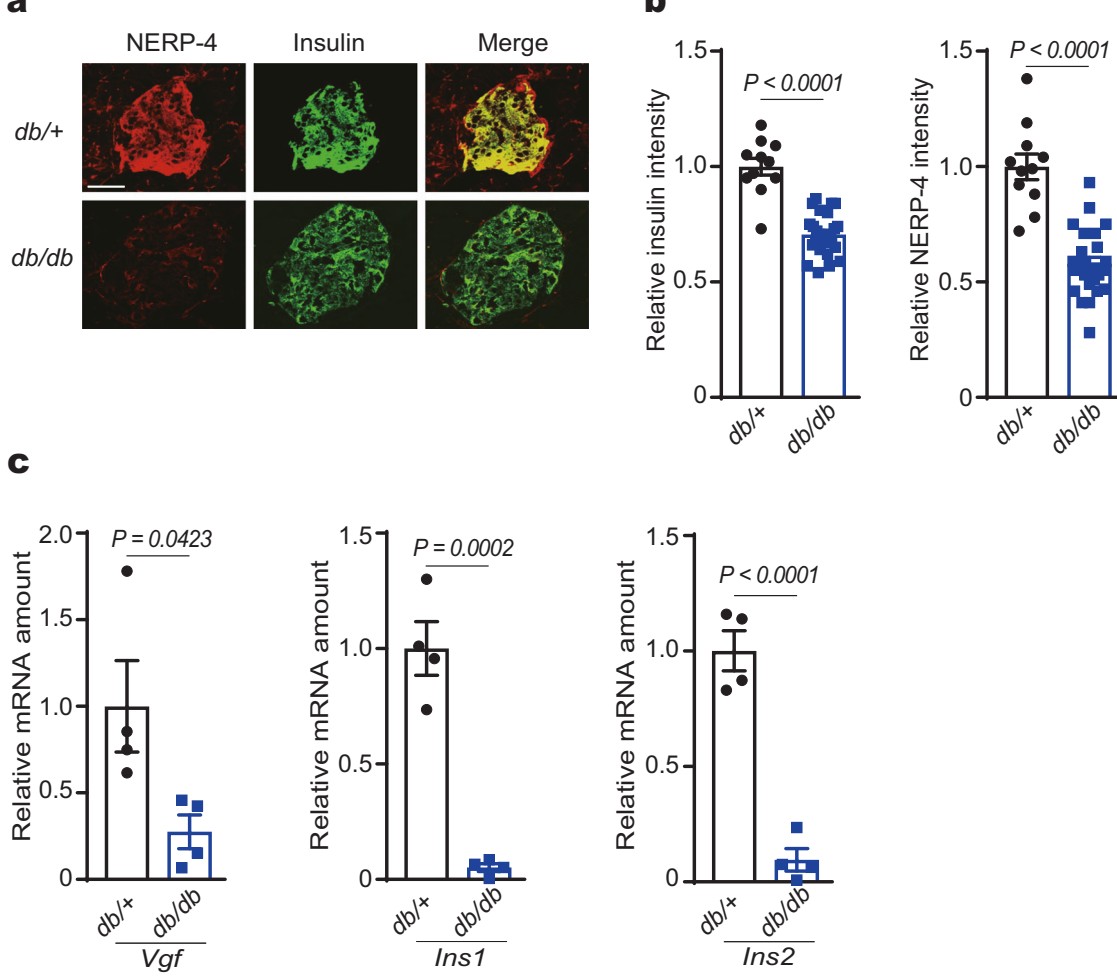

**Fig. 5 | NERP-4 is reduced in *db/db* mice. a** Representative NERP-4 (red) and insulin (green) immunoreactivities in pancreatic islets of 15-week-old *db/+* or *db/db* mice (*n* = 3 biological animals). **b** Relative fluorescence intensity of insulin or NERP-4 in pancreatic islets of 15-week-old *db/+* mice (11 islets from three mice) or *db/db* mice

(25 islets from three mice). **c** *Vgf*, *Ins1*, and *Ins2* mRNA amounts in islets of 10-week-old *db/+* mice and *db/db* mice (*n* = 4 biological animals). Data are mean ± s.e.m. Unpaired two-tailed Student's *t*-test (**b**, **c**). Scale bar, 50 μm (**a**). Source data are provided as a Source data file.

881.1 ± 220.7 μM and a Vmax of 2240 ± 242.3 pmol/mg protein/min (Fig. 9n). NERP-4 lowered the Km of [$^{14}$C]-MeAIB (412.6 ± 113.5 μM) without changing the Vmax. Furthermore, MeAIB did not alter [$^{125}$I]-Y-NERP-4[8–19] binding to MIN6-K8 cells (Supplementary Fig. 8h), supporting the finding that the NERP-4 binding site is different from the substrate binding site. Taken together, these results show that NERP-4 may modulate the binding affinity of SNAT2 to amino acids, implying that NERP-4 could act as a positive allosteric modulator (PAM).

## NERP-4 prevents β-cell impairment via SNAT2
Insulin increases protein and mRNA levels of SNAT2, thereby enhancing the amino acid supply and upregulating insulin-dependent

protein synthesis[39]. We found that insulin increased *Snat2* mRNA in naïve MIN6-K8 cells and C57BL/6 J mouse islets (Supplementary Fig. 9a, b). NERP-4 also increased *Snat2* mRNA in MIN6-K8 cells and C57BL/6 J mouse islets under non-stressed conditions (Supplementary Fig. 9c, d). SNAT2 is upregulated in β cells in diabetes as a response to the translational repression associated with ER stress[4]. We investigated *Snat2* expression in palmitate-treated C57BL/6 J mouse islets and *db/db* mouse islets. The *Snat2* mRNA level in C57BL/6 J mouse islets was increased by palmitate and was elevated to an even greater extent by NERP-4 (Fig. 10a). The *Snat2* mRNA level was higher in *db/db* mice compared with *db/+* mice (Fig. 10b). Two-week NERP-4 administration increased the *Snat2* mRNA level in *db/db* islets (Fig. 10c). The *Snat2*

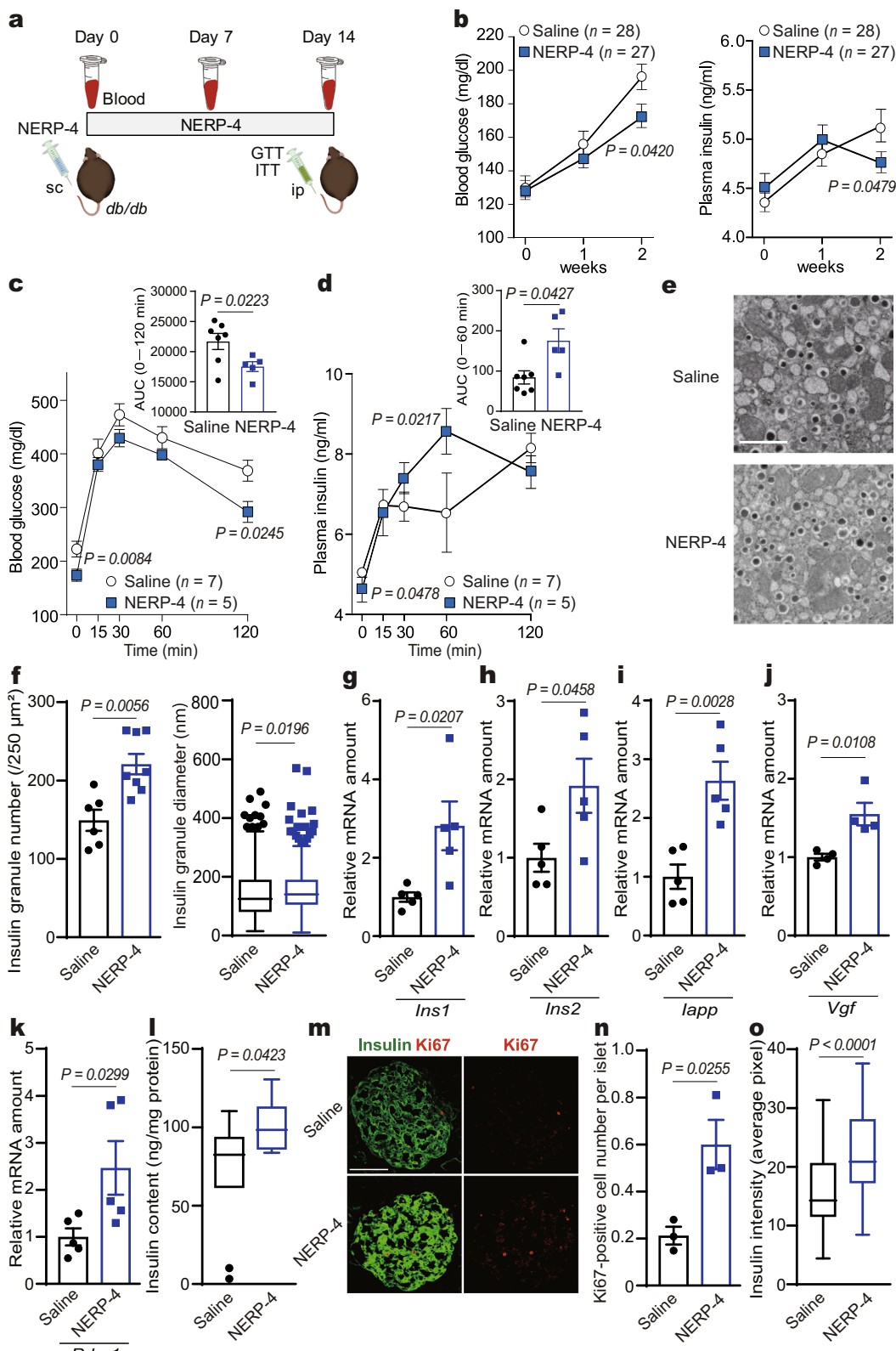

mRNA level was also increased in palmitate- or cytokine-treated MIN6-K8 cells after NERP-4 administration for 48 or 24 h, respectively (Supplementary Fig. 9e, f). SNAT2 upregulation induced by long-term NERP-4 administration under stress conditions could contribute to the effects of NERP-4 on β-cell viability and metabolism.

To verify that SNAT2 was responsible for NERP-4–induced β-cell maintenance, we examined how NERP-4 impacted the effects

of palmitate or cytokine treatment after *Snat2* knockdown (Fig. 10d). *Snat2* knockdown abolished the ability of NERP-4 to increase the OCR and the ATP production and to decrease ROS production in palmitate-treated MIN6-K8 cells (Fig. 10d–g). *Snat2* knockdown also abolished the NERP-4–induced recovery of cell viability in cytokine-treated MIN6-K8 cells (Fig. 10h). Together, these data show that NERP-4 enhances GSIS and reduces β-cell

**Fig. 6 | NERP-4 administration reverses β-cell impairment in *db/db* mice. a** Daily administration of NERP-4 or saline for 14 days to *db/db* mice. Blood was collected on Days 0, 7, and 14. GTT and ITT were performed on Day 14. **b** Changes in blood glucose and plasma insulin concentrations (*n* = 28, 27 biological animals). **c**, **d** Blood glucose and plasma insulin concentrations and their areas under the curves (AUCs) in an intraperitoneal GTT (*n* = 7, 5 biological animals). **e** Representative TEM micrograph of β cells. **f** Number of insulin storage granules (*n* = 6, 8 biological replicates). Box plots show the diameters of insulin storage granules (saline, *n* = 448; NERP-4, *n* = 790). **g**–**k** *Ins1*, *Ins2*, *Iapp*, *Vgf*, and *Pdx-1* mRNA amounts in pancreatic islets (*n* = 5 biological replicates). **l** Insulin content (*n* = 10 biological replicates). **m** Representative images of Ki67-positive cells (red) and insulin (green). **n** Number of Ki67-positive cells per islet (*n* = 3; saline, 145 islets from three mice and NERP-4, 147 islets from three mice). **o** Insulin intensity (saline, 78 islets from five mice and NERP-4, 69 islets from five mice). Results are pooled from four (**b**) or two (**c**, **d**, **l**) independent experiments or are representative of two independent experiments (**e**–**k**, **m**–**o**). Data are mean ± s.e.m (**b**–**d**, **f**–**k**, **n**). Centre line, median; box edges, first and third quartiles; whiskers, 1.5 times the inter-quartile range; outliers, individual points (**f**, **l**, **o**). Two-way ANOVA followed by Bonferroni's post-test for multiple comparisons (**b**–**d**). Unpaired two-tailed Student's *t*-test (**f**–**l**, **n**, **o**), Scale bars, 1 μm (**e**), 50 μm (**m**). Source data are provided as a Source data file.

impairment by enhancing both the expression and activity of SNAT2.

## Discussion

This study demonstrated that the VGF-derived peptide NERP-4 potently enhanced glutamine, alanine, and proline uptake into β cells via SNAT2, thereby stimulating mitochondrial ATP production and GSIS (Fig. 10i). NERP-4 maintains β cells by reducing oxidative and ER stress in *db/db* mouse islets and palmitate-treated C57BL/6 J mouse islets. Our findings showed that these effects of NERP-4 on β-cell maintenance resulted from the interaction between NERP-4 and SNAT2.

The granin protein VGF is processed to multiple bioactive peptides by the prohormone convertases PC1/3 and PC2 in the dense core vesicles of endocrine and neuroendocrine cells[17,18,20,40]. To identify novel bioactive peptides, we analysed the peptides secreted from TT cells[11], and isolated two C-terminally amidated VGF-derived peptides, NERP-1[VGF 281–306] and NERP-2. These two peptides were also expressed in human and mouse β cells[41,42]. NERP-2, but not NERP-1, enhanced GSIS by elevating intracellular Ca²⁺ in β cells[41]. NERP-4 shared no amino acid sequence homology with NERP-2. In the present study, NERP-4 additively enhanced GSIS when administered together with NERP-2, suggesting that the target protein of NERP-4 differs from that of NERP-2. VGF also produces three other insulinotropic peptides, namely TLQP-21, TLQP-62[VGF 556–617], and AQEE-30[VGF 588–617][19,43,44]. Chronic administration of TLQP-21 to prediabetic Zucker diabetic fatty rats preserved islet β-cell mass and slowed diabetes onset[19]. *C3aR1* knockout abolished the TLQP-21–induced anti-obesity effect seen in wild mice[45]. TLQP-62 induced a rapid increase in intracellular calcium mobilization, further increasing TLQP-62 secretion in a self-reinforcing manner[43]. AQEE-30 increased the phosphorylation of Akt and GSK3β in streptozotocin-treated β cells and suppressed β-cell death[46]. NERP-4 increased *Vgf* mRNA in *db/db* mouse islets, palmitate- or cytokine-treated β cells, and naïve β cells. VGF-derived peptides being induced by NERP-4 potentially mediate the long-term effects of NERP-4 on β-cell maintenance and function. Insulin itself has protective effects on β cells[47]. In the present study, NERP-4 recovered the suppression of insulin secretion caused by metabolic stress. NERP-4–induced insulin secretion could contribute to the effect of NERP-4 on β-cell maintenance.

To identify the NERP-4 target in β cells, we compared NERP-4 and NERP-2 in terms of the volcano plot results obtained by LRC-TriCEPS technology. NERP-4–SNAT2 interaction was verified by a binding assay, cellular functional experiments, and *Snat2* knockdown. SNAT2 has 11 putative transmembrane domains with two potential *N*-glyco-sylated sites, and imports amino acids in a Na⁺-cotransport manner[2,3,48]. Considering the kinetics analysis of the NERP-4–SNAT2 interaction and the lack of MeAIB interference with the binding between NERP-4 and SNAT2, NERP-4 may act as a PAM of SNAT2. In the process of drug discovery, PAMs have been developed for ion channels, kinases, phospholipases, and G protein–coupled receptors[49–52]. The binding of allosteric modulators to these target proteins had little impact on either the orthosteric or allosteric binding pockets[53]. The only endogenous PAM that has been identified thus far is the glutamate transporter–associated protein 3–18 (GTRAP3-18), which acts directly on an AAT named excitatory amino acid carrier 1 (EAAC1)[54]. The interaction between NERP-4 and SNAT2 provides evidence that an endogenous peptide can modulate the activity of an AAT. Cryo-electron microscopy studies have determined the architectures and transport mechanisms of some AATs[49,55,56]. The topological structure and amino acid transport site of SNAT2 have yet to be determined. Future structural analyses of the molecular features of the SNAT2-binding site of NERP-4 could elucidate how NERP-4 modulates SNAT2 activity.

SNAT2 imports glutamine, alanine, proline, serine, and other amino acids[57]. Among the SNAT2 substrates, glutamine, alanine, and proline regulate insulin secretion[2,58,59]. SNAT3 and SNAT5 have bidirectional transport properties depending on the metabolic/nutritional status[3]: they import glutamine, asparagine, alanine, and histidine, and export glutamine and glycine[57,60]. In addition to its role in amino acid transfer, SNAT2 possesses a transceptor-like function[61]. It detects whether there are sufficient amounts of the amino acids that regulate its expression and stability[61,62]. SNAT2-mediated Na⁺ incorporation plays a potential role in membrane depolarisation and increases both intracellular Ca²⁺ levels and insulin secretion[59]. SNAT2 activation can also activate several transduction pathways that support cell growth and proliferation[61–63]. In the present study, NERP-4 increased SNAT2 expression, suggesting that NERP-4 not only contributes to the transport of Na⁺ and amino acids via SNAT2, but also increases SNAT2 expression that can enhance its downstream signalling.

After being imported into β cells, glutamine and alanine are metabolised in the tricarboxylic acid (TCA) cycle to produce ATP, and eventually potentiate Ca²⁺ influx and insulin secretion[2]. Cytosolic glutamine is converted to glutamate, which yields the antioxidant glutathione, thereby reducing ROS production and improving mitochondrial dysfunction[64,65]. Glutamate also yields γ-aminobutyric acid, which contributes to β-cell survival[3]. Alanine is rapidly converted to pyruvate, and then to glutamate, aspartate, and lactate via the TCA cycle[2]. It is currently unknown which SNAT is responsible for alanine export from β cells. Proline plays key roles in protein structure and function, and also in the maintenance of cellular redox homeostasis[66]. Proline also stimulates insulin secretion through plasma membrane depolarisation as well as cell glycolytic and oxidative metabolism[58,59]. In this study, NERP-4 increased the intracellular contents of SNAT2 substrates and stimulated GSIS from MIN6-K8 cells prepared under amino acid–free conditions. Amino acids secreted from the cells potently mediated NERP-4–induced GSIS. Identifying a novel mechanism whereby the peptide–AAT axis regulates amino acid transport could increase our understanding of how amino acids regulate β-cell biology.

Glucotoxicity and lipotoxicity in type 2 diabetes stress β cells by increasing insulin biosynthesis[6,25]. β cells are susceptible to ROS-induced cellular stress and protein misfolding in the context of excessive caloric intake[25,67]. Mitochondrial quality is dynamically maintained by the regulation of biosynthesis, fission, fusion, and mitophagy[68]. Mitochondrial dysfunction and both oxidative and ER

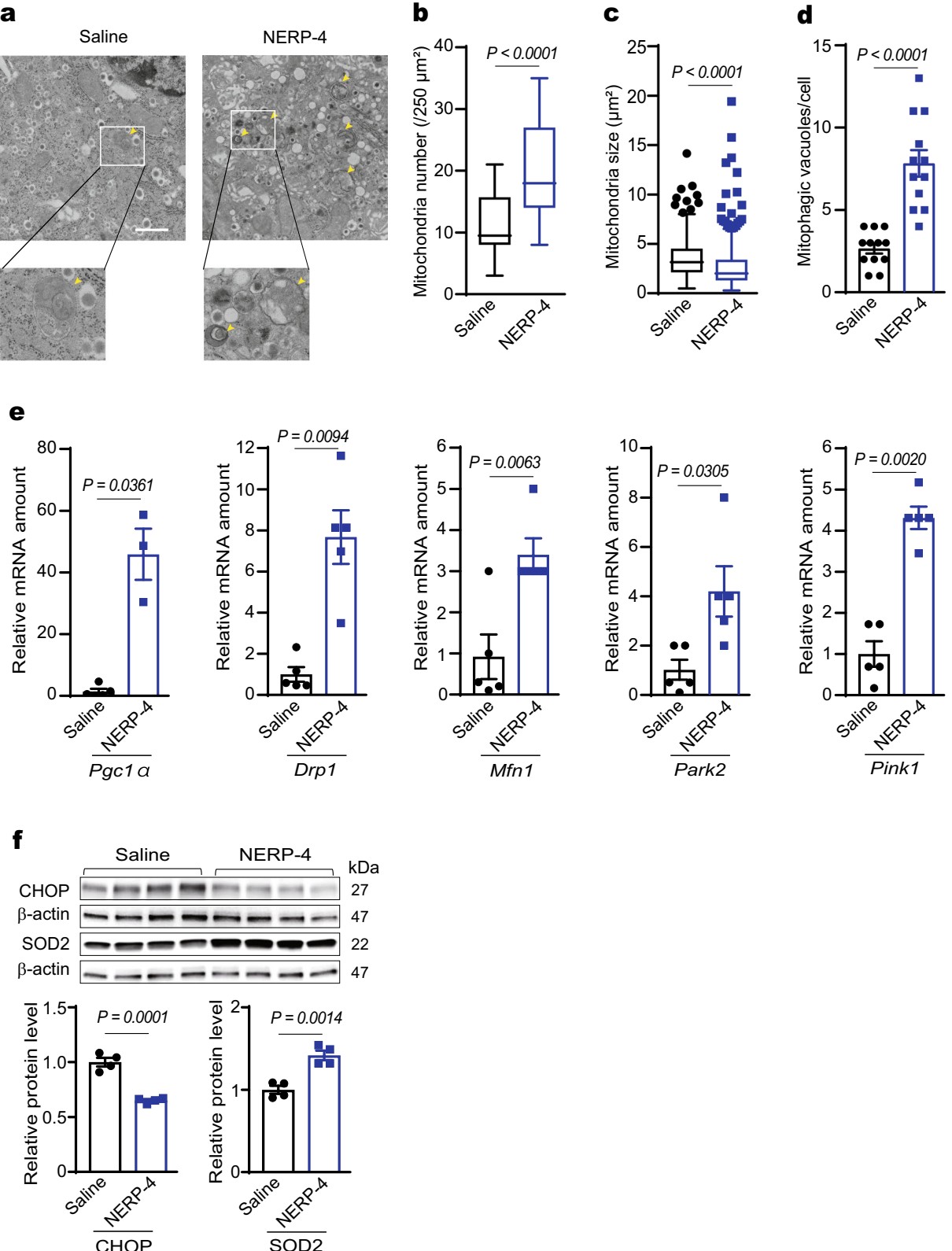

**Fig. 7 | NERP-4 administration reverses mitochondrial dynamics in the islets of *db/db* mice. a** Representative TEM images of mitochondria in β cells. Organelles surrounded by single or double membranes represent mitophagic vacuoles containing mitophagosomes (yellow arrows). Box plots displaying mitochondrial number (**b**) and size (**c**) (saline, 386 mitochondria from three mice and NERP-4, 417 mitochondria from three mice). **d** Quantification of the number of mitophagic vacuoles (12 cells from five mice). **e** *Pgc1α, Drp1, Mfn1, Park2,* and *Pink1* mRNA

amounts in pancreatic islets (*n* = 5 biological replicates). **f** Protein levels of CHOP and SOD2 (*n* = 4 biological replicates). Representative results of three (**e**) or two (**f**) independent experiments. Data are mean ± s.e.m (**d**–**f**). Centre line, median; box edges, first and third quartiles; whiskers, 1.5 times the interquartile range; outliers, individual points (**b**–**c**). Unpaired two-tailed Student's *t*-test (**b**–**f**). Scale bar, 1 μm (**a**). Source data are provided as a Source data file.

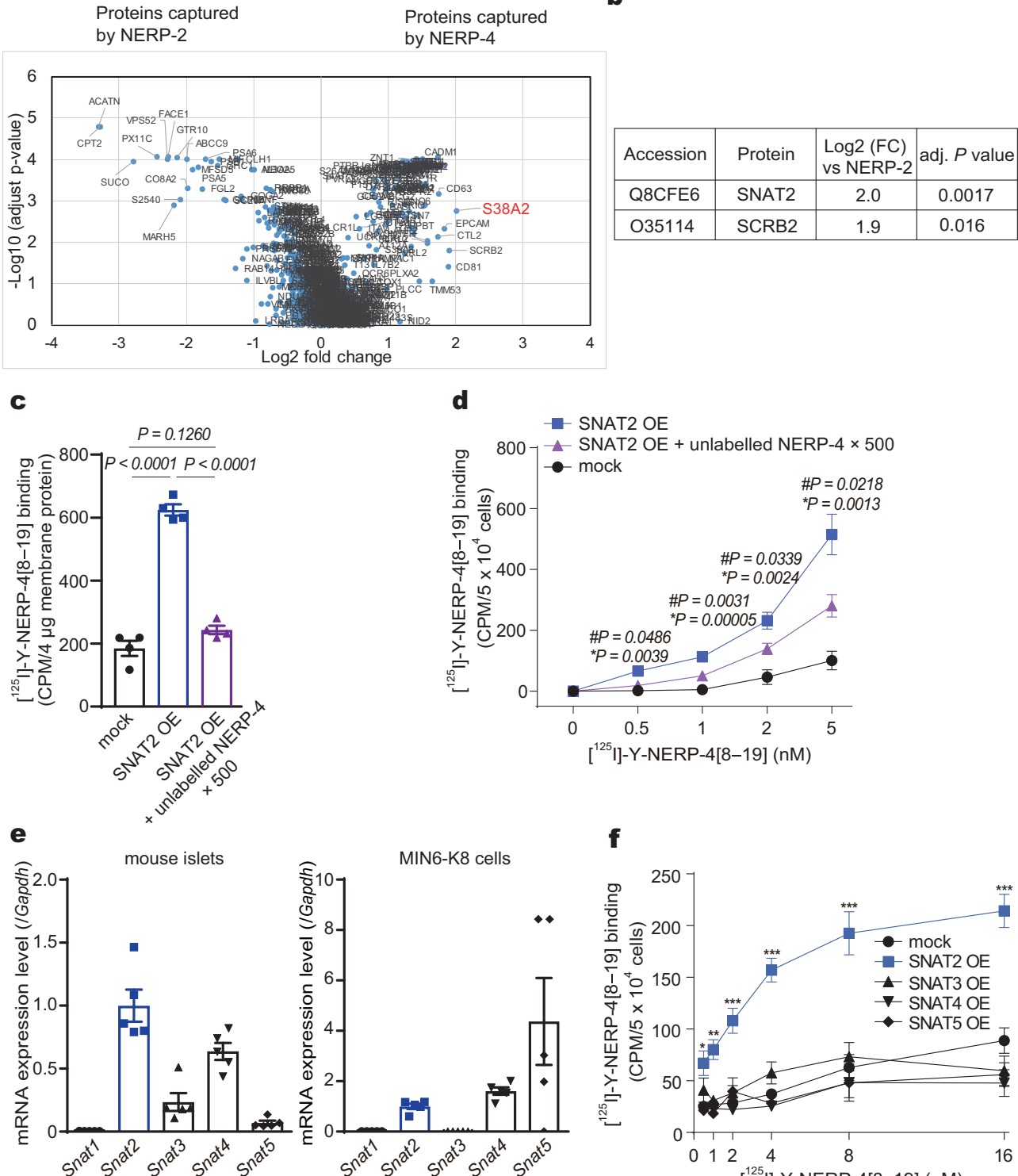

**Fig. 8 | SNAT2 is a target protein candidate for NERP-4. a** Volcano plot depicting a comparison of proteins captured by NERP-4 or NERP-2. Data are shown at the protein level and are annotated using the UniProt mouse database. *Y*-axis = −Log10 (adjusted *P* value), *X*-axis = Log2 fold change compared to the other samples (*n* = 3 independent experiments). **b** Target candidates identified by the TriCEPS™-based ligand–receptor capture method. SCRB2: lysosome membrane protein 2. **c** Binding of [125I]-Y-NERP-4[8–19] to the membrane of SNAT2-OE HEK293 cells with or without unlabelled NERP-4 (*n* = 4 biological replicates). **d** Binding of [125I]-Y-NERP-4[8–19] with or without unlabelled NERP-4 in SNAT2-overexpressing (OE) HEK293 cells (*n* = 4 biological replicates). **e** mRNA levels of *Snat1*, *Snat2*, *Snat3*, *Snat4*, and

*Snat5* in C57BL/6 J mouse islets and MIN6-K8 cells (*n* = 5 biological replicates). **f** Binding of [125I]-Y-NERP-4[8–19] to mock, SNAT2-, SNAT3-, SNAT4-, or SNAT5-OE HEK293 cells (*n* = 3 biological replicates). Representative results of two independent experiments (**c**–**f**). Data are mean ± s.e.m (**c**–**f**). Differential protein abundance was tested using a statistical ANOVA model followed by multiple testing corrections (**a**, **b**). One-way ANOVA and Fisher's LSD test, #*P* value, SNAT2 OE vs. SNAT2 OE plus unlabelled NERP-4; *P* value, SNAT2 OE vs. mock (**c**). One-way ANOVA and Tukey's multiple comparisons test (**d**, **e**). Two-way ANOVA and Tukey's multiple comparisons test, *P* = 0.0444, **P* = 0.0051, ***P* < 0.0001 vs. mock; not significant, SNAT3/SNAT4/SNAT5 vs. mock (**f**). Source data are provided as a Source data file.

stress in β cells exacerbate diabetes[6,31]. In addition to insulin, β cells produce peptides that preserve β-cell longevity by protecting against metabolic stress[69]. In this study, NERP-4 administration to *db/db* mice upregulated genes related to mitochondrial biogenesis and dynamics. Under both stressed and non-stressed conditions, NERP-4 administration to MIN6-K8 cells improved β-cell viability and function. By contrast, NERP-4 neutralisation in isolated mouse islets aggravated oxidative and ER stress and impaired GSIS, suggesting that endogenous NERP-4 functions in β-cell behaviour.

The present study has expanded on the role of SNAT2 in β-cell maintenance. The biosynthesis of SNAT2 is regulated by the concentrations of insulin and amino acids[4]. Under ER stress in diabetes, ATF4 upregulates SNAT2 mRNA in β cells to stimulate amino acid influx and protein synthesis[4,7]. Palmitate induces oxidative stress by impairing cytosolic and mitochondrial $Ca^{2+}$ homeostasis, thereby causing mitochondrial dysfunction and β-cell apoptosis[70,71]. In this study, NERP-4 upregulated SNAT2 mRNA in *db/db* islets, both palmitate- and cytokine-treated β cells, and naïve β cells. By contrast, *Snat2* knockdown abolished the protective effects of NERP-4 on mitochondrial function, ROS production, and cell viability. In addition to the insulinotropic activity of NERP-4, which is mediated by SNAT2, NERP-4 protects β cells from oxidative and ER stress via SNAT2.

The present study revealed that NERP-4 acts on SNAT2 to regulate β-cell function and maintenance. NERP-4 is secreted together with insulin following glucose stimulation, and modulates SNAT2 activity in β cells. Understanding the NERP-4–SNAT2 axis may clarify how glucose and amino acids interact in β-cell biology, thus serving as a foundation for future research and therapeutic strategies in diabetes.

## Methods

### Animals
C57BL/6 J mice, *db/+* and *db/db* mice, Wistar rats (Charles River Laboratories), *Vgf* KO mice (Dr. S.R. Salton, Mount Sinai School of Medicine)[24], and apoaequorin transgenic mice[10] were maintained under controlled temperature (21–23 °C) and light (light on: 08:00–20:00) conditions with free access to a standard diet (CLEA Japan, CE-2) and water. Male mice were used for the experiments. For GSIS studies, islets were isolated from both male and female C57BL/6 J mice. All animal experiments were performed in accordance with the Japanese Physiological Society guidelines for animal care and were approved by the Ethics Committee on Animal Experimentation of the University of Miyazaki. The mice and rats were humanely euthanized by intraperitoneal injection with a combination anesthetic, consisting of 0.3 mg/kg of medetomidine, 4.0 mg/kg of midazolam, and 5.0 mg/kg of butorphanol. Pancreas tissues and islets were collected after euthanasia.

### Statement of ethics
Human tissues were studied after approval by the Ethics Committee of the University of Miyazaki (approval date: April 23, 2017; Approval No. O-136). Human pancreatic islets isolated from a heart-beating cadaveric donor (female, 53 years old) as described previously[72] next-of-kin consent for research, were kindly provided by the University Hospital of Lille via the European Consortium for Islet Transplantation human islet distribution program supported by the Juvenile Diabetes Research Foundation. Islet purity was assessed as the percentage of endocrine clusters positive to dithizone staining (range: 80–90%). Participates provided written informed consent. Permissions for research use of human specimens and exportation of human islets were granted by the French Ministry of Higher Education and Research and the Agence de la biomédecine, respectively.

### Cultured cells
MIN6-K8 cells were provided by Dr. Jun-ichi Miyazaki (Osaka University). MIN6-K8 cells are mouse insulinoma–derived β cells[21] that

have been used to study the mechanism of glutamine-amplified $Ca^{2+}$ influx and insulin secretion[22,23]. HEK293 cells were from ATCC (CRL-1573). They were maintained in Dulbecco's modified Eagle's medium (DMEM, 044-29765, Wako) supplemented with 10% fetal bovine serum (FBS), 25 mM glucose, 4 mM glutamine, 100 U/mL benzylpenicillin, and 100 mg/mL streptomycin at 37 °C in a humidified atmosphere of 5% $CO_2$.

### Quantitative real-time PCR (RT-PCR)
mRNA was extracted from MIN6-K8 cells and mouse islets with a Ribopure™ Kit (Thermo Fisher Scientific). First-strand cDNA was synthesised with high-capacity cDNA reverse transcription kits (Applied Biosystems). RT-PCR was performed with the TaqMan/ Applied Biosciences primers shown in Supplementary Table 3. Quantitative RT-PCR was performed with TaqMan Fast Universal PCR Master Mix (Thermo Fisher Scientific) and a Thermal Cycler Dice Real Time System II (Takara Bio). PCR products were normalised to the expression of glyceraldehyde3-phosphate dehydrogenase (*Gapdh*) mRNA.

### Digital PCR
Primers (Supplementary Table 3), QS3D Chips, and the ProFlex PCR System (Thermo Fisher Scientific) were used for amplification. QS3D Chips were analysed using QuantStudio 3D Analysis Suite Cloud Software (Thermo Fisher Scientific).

### Short interfering RNA (siRNA) treatment
MIN6-K8 cells ($1 \times 10^5$ cells/well in 24-well plates) were treated with mouse *Snat2* siRNA (s85605, Thermo Fisher Scientific, 20 pmol/mL), mouse *Vgf* siRNA (s117809, Thermo Fisher Scientific, 20 pmol/mL), or scrambled control siRNA (siSCR) (4390843, Thermo Fisher Scientific) using Lipofectamine 2000 (Thermo Fisher Scientific) for 72 h. Knockdown efficacy was studied by RT-PCR.

### Apoaequorin transgenic mice
We generated apoaequorin transgenic mice to explore transient luminescence caused by intracellular $Ca^{2+}$ mobilisation as described elsewhere[10]. Mouse pancreatic tissues were minced in RPMI-1640 (Invitrogen) to prepare small pieces. In order to reconstitute aequorin, 1–2 $mm^3$ pieces were incubated with coelenterazine (Molecular Probes) for 3 h. Pancreatic pieces were sequentially treated with 1 μM NERP-4, 100 μM ATP, and 2.5% Triton X-100. The light emission of apoaequorin was continuously monitored every second as relative luminescence units in an AutoLumat LB 953 luminometer (Berthold Technologies).

### Measurement of intracellular $Ca^{2+}$ influx
Naïve, siSCR-treated, or *Snat2* knockdown MIN6-K8 cells were cultured in 35-mm culture plates (Becton Dickinson Labware). Cells were loaded with 1 μM fura-2 acetoxymethyl ester (Fura-2-AM, Dojindo) in HKRB containing 2.8 mM glucose for 30 min at 37 °C. Culture plates were placed on the stage of an integrated fluorescence microscope (BZ-X700, Keyence). Images were captured at 10-s intervals, and 340- and 380-nm excitation filters were used for Fura-2-AM dual-wavelength excitation-ratio imaging. For $[Ca^{2+}]_i$ measurements, cells were exposed to HKRB containing 2.8 mM glucose for 30 min, then to HKRB containing 16.7 mM glucose with or without $10^{-10}$ M NERP-4. In some experiments, 1 mM EGTA (Nacalai Tesque), 10 μM nifedipine (Sigma-Aldrich), or 10 mM MeAIB (Abcam) was administered when 2.8 mM glucose started. $[Ca^{2+}]_i$ was also measured in MIN6-K8 cells incubated under 2.8 mM glucose with $10^{-10}$ M NERP-4. The fluorescence ratio was recorded for 22 min. At the end of each experiment, cells were exposed to 35 mM KCl for 2 min. All data are expressed as per cent changes relative to the average fluorescence ratio in 2.8 mM glucose.

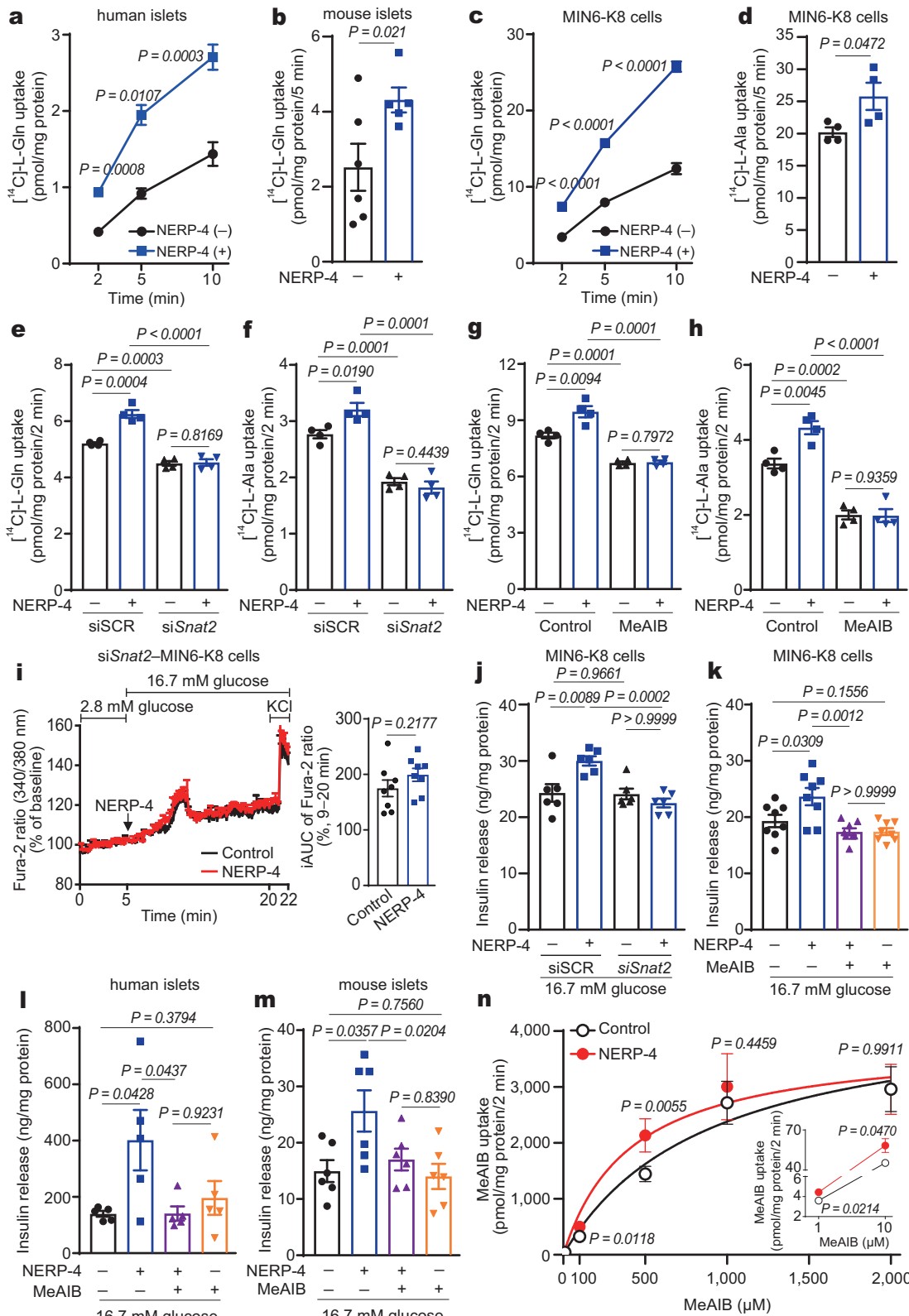

## Antibody preparation

Acetyl AC-NERP-4[8−19] (Supplementary Table 6) was conjugated with maleimide-activated keyhole limpet hemocyanin (Pierce) and used to immunise rabbits, as described elsewhere[20]. Rabbits were immunized with each conjugate emulsified with an equal volume of Freund's complete adjuvant. Anti−NERP-4 IgG and NRS IgG were purified from acetyl AC-NERP-4[8−19] antiserum and normal rabbit serum (FUJIFILM

Wako Pure Chemical), respectively, over a Protein G Sepharose 4 Fast Flow column (Merck Millipore).

## Immunohistochemistry and immunocytochemistry

Pancreata were infiltrated with 4% paraformaldehyde overnight at 4 °C and embedded in OCT compound (Sakura Finetek). MIN6-K8 cells were fixed with 4% paraformaldehyde for 15 min at room temperature

**Fig. 9 | NERP-4 stimulates glutamine and alanine uptake into β cells via SNAT2.**
NERP-4–induced [$^{14}$C]-L-glutamine uptake into human islets (**a**, $n = 4$ biological replicates), C57BL/6 J mouse islets (**b**, $n = 6$, 5 biological replicates), and MIN6-K8 cells (**c**, $n = 4$ biological replicates). **d** [$^{14}$C]-L-Alanine uptake into MIN6-K8 cells ($n = 4$ biological replicates). [$^{14}$C]-L-Glutamine (**e**) and [$^{14}$C]-L-alanine (**f**) uptake in si*Snat2*-MIN6-K8 cells ($n = 4$ biological replicates). [$^{14}$C]-L-Glutamine (**g**) and [$^{14}$C]-L-alanine (**h**) uptake by MIN6-K8 cells with or without NERP-4 and MeAIB ($n = 4$ biological replicates). **i** Representative Fura-2-AM ratios in si*Snat2*–MIN6-K8 cells in response to NERP-4 ($n = 8$ cells), and average iAUC (9–20 min) of [Ca$^{2+}$]$_i$ ($n = 8$ cells). Effects of *Snat2* knockdown (**j**, $n = 6$ biological replicates) and MeAIB (**k**, $n = 8$ biological

replicates) on NERP-4–induced GSIS in MIN6-K8 cells. Effect of MeAIB on NERP-4–induced GSIS in human islets (**l**, $n = 5$ biological replicates) and C57BL/6 J mouse islets (**m**, $n = 6$ biological replicates). **n** Concentration dependence of MeAIB uptake into MIN6-K8 cells in the presence or absence of NERP-4 ($n = 4$ biological replicates). Inset shows 1 or 10 μM MeAIB uptake. All experiments were performed under 16.7 mM glucose. Representative results of two independent experiments (**b–n**). Data are mean ± s.e.m (**a–n**). One-way ANOVA and Tukey's multiple comparisons test (**a**, **c**, **e–h**, **j–n**). Unpaired two-tailed Student's *t*-test (**b**, **d**, **i**). Source data are provided as a Source data file.

(RT). Antibodies and staining conditions are described in Supplementary Table 4. For double immunofluorescence staining[41], pancreata or MIN6-K8 cells were incubated overnight at 4 °C with the first primary antibodies and then overnight with the second primary antibodies (Supplementary Table 4). They were then incubated at RT with corresponding Alexa Fluor 488– or 594–conjugated secondary antibodies at RT for 1 h. Samples were observed using an AX-7 fluorescence microscope (Olympus) or C2 confocal microscope (Nikon). The fluorescence intensity and area were measured using ImageJ software (National Institutes of Health).

### Subcellular fractionation
NERP-4–induced Nrf2 translocation into the nucleus was studied in isolated mouse islets. NE-PER Nuclear and Cytoplasmic Extraction Reagents (Thermo Fischer Scientific) was used to prepare cytosol and nuclear fractions. The cytosol extracts and nuclear extracts were transferred to a new prechilled tube and stored at −80 °C until used. The protein contents of the extracts were determined by a Bradford assay.

### Western blotting
Proteins (20 μg) extracted from HEK293 cells, MIN6-K8 cells, and mouse islets were analysed by western blot with the indicated antibodies (Supplementary Table 4), as described elsewhere[73]. The cells and islets were ruptured with RIPA buffer (Nacalai Tesque). The protein samples were resolved by SDS–PAGE and transferred onto PVDF membranes, blocked for 1 h with 5% skim milk, and incubated overnight at 4 °C with primary antibodies. After three washes with TBST, the membranes were incubated for 1 h with the appropriate secondary antibodies and developed using chemiluminescent substrates. Fusion Edge software (Vilber Lourmat) was used for quantification.

### Double immunoelectron microscopy
Double immunoelectron microscopy was performed as described elsewhere[74]. Ultrathin sections of the pancreata of C57BL/6 J mice were incubated with guinea pig anti-insulin antibody for 12 h at 4 °C, followed by incubation with 10 nm of gold-labelled goat anti–guinea pig IgG (BB International) for 3 h at RT. They were next incubated with anti–NERP-4 antibody for 12 h at 4 °C and further with 5 nm of gold-labelled goat anti–rabbit IgG (BB International) for 3 h at RT. They were stained with uranium acetate–lead citrate and examined using an H-7600 transmission electron microscope (Hitachi).

### NERP-4 secretion from MIN6-K8 cells
MIN6-K8 cells at 80% confluence were incubated in HKRB containing 2.8 mM or 16.7 mM glucose for 30 min. The supernatant was applied to a Sep-Pak C-18 cartridge (Waters), and peptides bound to the resin were eluted with a 60% acetonitrile solution containing 0.1% trifluoroacetic acid (TFA), as described elsewhere[11]. A portion of the eluate was subjected to radioimmunoassay for NERP-4.

### Quantification and chromatographic characterisation of NERP-4 in human pancreas
Noncancerous human pancreatic tissues (total wet weight 7.95 g) were obtained from four patients undergoing surgery for pancreatic cancer.

The study was carried out in accordance with the principles of the Declaration of Helsinki, and written informed consent was obtained. The tissues were subjected to the peptide isolation procedure, as described previously[11]. The pancreatic extract was applied to a Sep-Pak C-18 cartridge. The eluted peptides were subjected to RP-HPLC analysis using a TSKgel ODS-120T column (Tosoh) and a linear gradient of 10–60% acetonitrile containing 0.1% TFA for 80 min at a flow rate of 1.0 mL/min.

### Radioimmunoassay (RIA)
RIA was carried out using antiserum for NERP-4 and [$^{125}$I]-Y-NERP-4[8–19]. The half-maximum-inhibition amount of ligand binding in the RIA was 30 fmol/tube of NERP-4. The specificity of NERP-4 RIA was examined for 20 peptides, including the VGF-derived peptides listed in Supplementary Table 5.

### Islet isolation
Mouse pancreatic islets were isolated as described previously[75]. Each minced pancreas was digested in 10 mL collagenase P (1 mg/mL in HBSS, Roche Diagnostics) for 15 min at 37 °C. The pellet was resuspended in RPMI-1640 supplemented with 10% FBS, 1% penicillin–streptomycin, 2 mM glutamine, and 11 mM glucose. Pancreatic islets were selected under a stereomicroscope.

### Insulin secretion and content in MIN6-K8 cells and islets
Naïve, siSCR-treated, and *Snat2* or *Vgf* knockdown MIN6-K8 cells ($2 \times 10^5$ cells/well) were pre-incubated for 30 min in HKRB containing 2.8 mM glucose, and then incubated for 30 min in HKRB containing 16.7 mM glucose with or without $10^{-10}$ M NERP-4. EGTA (1 mM), nifedipine (10 μM), glutamine (1 μM to 1 mM, Nacalai Tesque), MeAIB (10 mM), or NERP-2 (1 μM)[42] was administered in experiments described in the text. Ten size-matched islets from C57BL/6 J mice were pre-incubated in 2.8 mM glucose for 1 h, then incubated for 1 h with 2.8 mM or 16.7 mM glucose, or with 16.7 mM glucose containing $10^{-8}$ M NERP-4. To examine the effect of palmitate plus low glutamine, islets were cultured in RPMI-1640 medium supplemented with 0.5 mM palmitate (Sigma-Aldrich), 1 mM glutamine, 10% FBS, 100 U/mL benzylpenicillin, and 100 mg/mL streptomycin, and were treated with $10^{-8}$ M NERP-4 daily for 3 d. MIN6-K8 cells were cultured in DMEM supplemented with 0.5 mM palmitate, 2 mM glutamine, 10% FBS, 100 U/mL benzylpenicillin, and 100 mg/mL streptomycin, and were treated with $10^{-10}$ M NERP-4 daily for 2 d. To examine the effects of cytokines, MIN6-K8 cells were treated with a cytokine cocktail (50 ng/mL mouse tumour necrosis factor-α, 10 ng/mL mouse interleukin-1β, and 50 ng/mL mouse interferon-γ) (Mitltenyi Biotec) and $10^{-10}$ M NERP-4 in DMEM (045-30285, Wako) supplemented with 2 mM glutamine, 10% FBS, 100 U/mL benzylpenicillin, and 100 mg/mL streptomycin for 24 h. To measure insulin contents, MIN6-K8 cells and islets were treated with 1% hydrochloric acid–ethanol for 24 h at 4 °C after ultrasonic homogenisation. Mouse insulin and human insulin were measured with an ultra-sensitive Mouse Insulin ELISA Kit (Morinaga Institute of Biological Science) and a Mercodia Insulin ELISA Kit (Mercodia), respectively. Protein content was determined with a Protein Assay Kit I (Bio-Rad). All GSIS experiments were conducted in amino acid–free buffer.

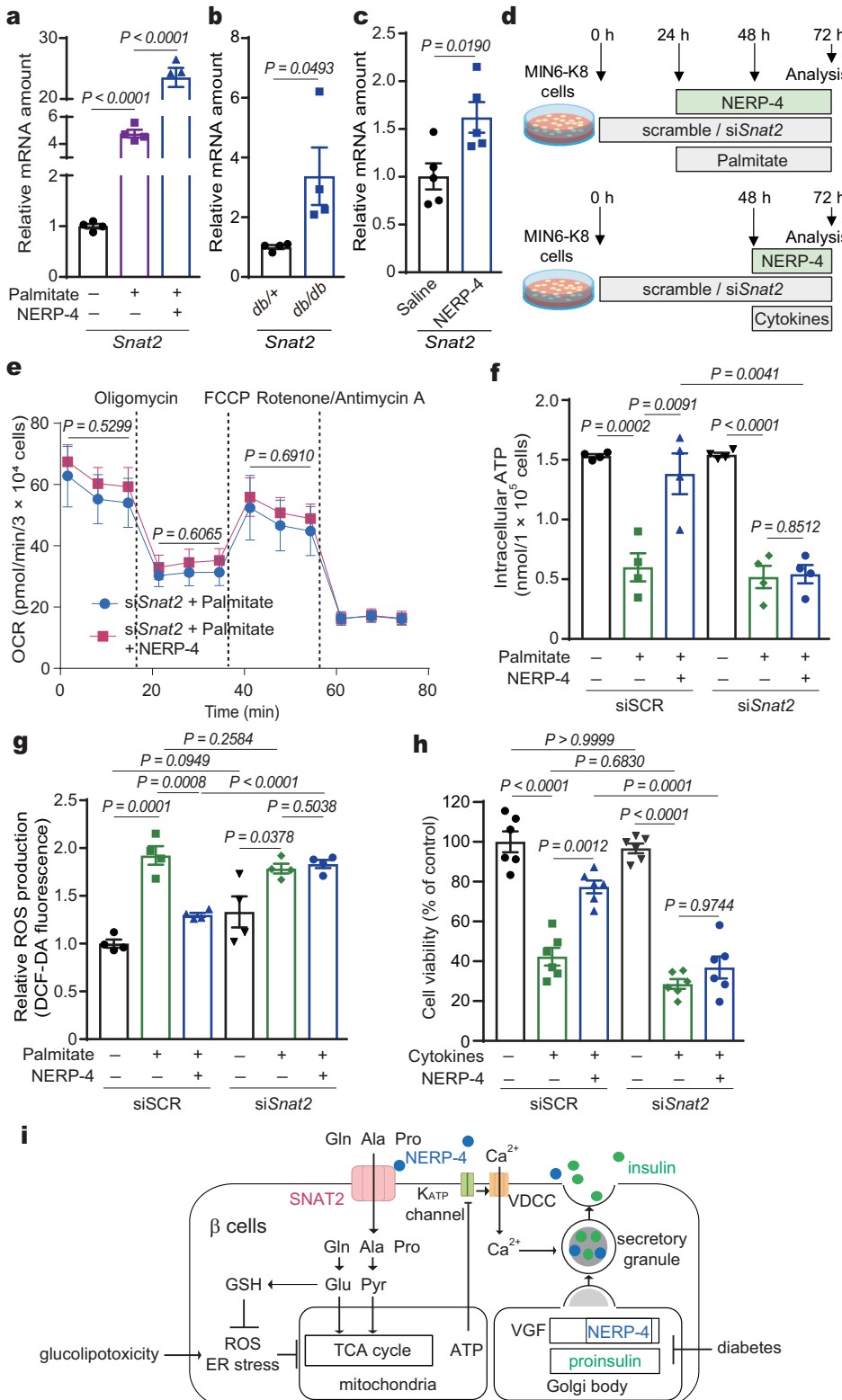

## Neutralisation of NERP-4 in experiments with mouse islets

To study the short-term effect of NERP-4 IgG, C57BL/6 J mouse islets were pre-incubated for 3 h with 3 µg/mL anti–NERP-4 IgG or NRS IgG in HKRB buffer containing 2.8 mM glucose. Islets were incubated for 1 h with 3 µg/mL anti–NERP-4 IgG, 3 µg/mL NRS IgG, or 3 µg/mL NRS IgG plus $10^{-8}$ M NERP-4 in HKRB buffer containing 16.7 mM glucose. To study the long-term effect of NERP-4 IgG, C57BL/6 J mouse islets cultured in RPMI-1640 medium containing 1 mM glutamine were treated

with 3 µg/mL anti–NERP-4 IgG or NRS IgG daily for 3 d. The islets were stimulated by 16.7 mM glucose for 1 h after pre-incubation for 1 h in 2.8 mM glucose.

### ATP assay

Naïve, siSCR-treated or *Snat2* knockdown MIN6-K8 cells were incubated with 2.8 mM or 16.7 mM glucose with or without $10^{-10}$ M NERP-4 for 30 min. The cells were treated with 0.5 mM palmitate in DMEM

**Fig. 10 | NERP-4 reverses β-cell impairment via SNAT2. a,** *Snat2* mRNA amounts in mouse islets under palmitate treatment with or without NERP-4 ($n = 4$ biological replicates). **b** *Snat2* mRNA amounts in 10-week-old *db/+* mouse and *db/db* mouse islets ($n = 4$ biological animals). **c** *Snat2* mRNA amounts in islets from *db/db* mice administered NERP-4 for two weeks ($n = 5$ biological replicates). **d** NERP-4 was administered at 24 and 48 h after the start of siSCR treatment or *Snat2* knockdown to MIN6-K8 cells under palmitate (**e–g**) or at 48 h under cytokines (**h**). **e** OCR ($n = 3$ biological replicates). The *P* values indicated the differences of basal respiration (1.6–14.8 min), ATP production (21.4–34.6 min), and maximal respiration (41.2–54.4 min) in palmitate-treated si*Snat2*-MIN6-K8 cells with or without NERP-4. **f** ATP production ($n = 4$ biological replicates). **g** ROS production ($n = 4$ biological replicates). **h** Cell viability ($n = 6$ biological replicates). Representative results of two independent experiments (**a–c**, **e–h**). Data are mean ± s.e.m. (**a–c**, **e–h**). One-way ANOVA and Tukey's multiple comparisons test (**a**, **e–h**). Unpaired two-tailed Student's *t*-test (**b**, **c**). **i** Schematic of NERP-4 roles in pancreatic β cells. NERP-4 is processed from VGF, a granin protein that is critical for granule biogenesis in pancreatic β cells. NERP-4 is packed with insulin in secretory granules and secreted by glucose. NERP-4 binds to SNAT2 to stimulate amino acid uptake into β cells, thereby enhancing mitochondrial ATP production, glucose-induced Ca²⁺ mobilisation into β cells, and GSIS. NERP-4 expression is reduced in β cells in *db/db* mice. NERP-4 protects β cells from glucolipotoxicity by reducing ROS production and ER stress, thereby enhancing mitochondrial biogenesis and dynamics. Source data are provided as a Source data file.

containing 2 mM glutamine for 48 h, and treated with $10^{-10}$ M NERP-4 once daily during palmitate treatment. They were lysed and the ATP content was determined with a CellTiter-Glo® Luminescent Cell Viability Assay Kit (Promega) or a Colorimetric/Fluorometric Assay Kit (BioVision).

## Measurement of intracellular cAMP
MIN6-K8 cells were pre-incubated with HKRB buffer containing 1 mM glucose and 250 μM 3-isobutyl-1-methylxanthine (Sigma-Aldrich) for 1 h at 37 °C. They were treated with 2.8 mM glucose or 16.7 mM glucose with or without $10^{-10}$ M NERP-4 for 30 min. cAMP was measured with a cAMP Biotark enzyme immunoassay system (GE Healthcare).

## Cell viability assay and terminal deoxynucleotidyl transferase dUTP nick end labelling (TUNEL) assay
Cell viability was studied with a Cell Counting Kit-8 (Dojindo). Cells were treated with a cytokine cocktail (50 ng/mL IFN-γ, 50 ng/mLTNFα, 10 ng/mL IL-1β; Supplementary Table 6) and $10^{-10}$ M NERP-4 in DMEM containing 2 mM glutamine for 24 h. A TUNEL assay was carried out using an In Situ Cell Death Detection Kit, Fluorescein (Roche Diagnostics). TUNEL-positive cells were counted in five random areas.

## TriCEPS™-based ligand–receptor glycocapture
The putative target protein for NERP-4 was identified with TriCEPS™-based ligand–receptor capture (LRC-TriCEPS, P05201; Dualsystems Biotech)[33]. In brief, 300 μg NERP-4 or NERP-2 used as a control was reacted with TriCEPS™ reagent in 25 mM HEPES buffer for 2 h at 22 °C. Cell surface proteins in MIN6-K8 cells ($1.2 \times 10^8$) were oxidised with 1.5 mM NaIO₄. TriCEPS™-coupled NERP-4 or NERP-2 was incubated with cell surface proteins for 90 min; the proteins were then purified by solid-phase chromatography. Proteins were reduced, alkylated, and digested with trypsin. Tryptic peptides were analysed on a Thermo LTQ Orbitrap XL spectrometer fitted with an electrospray ion source. The experiment was performed two times. Progenesis software was used for raw file alignment and feature detection, the Comet search engine was used for spectra identification, and the Trans proteomic pipeline was used for statistical validation of putative identifications and protein inference[33]. Upon protein inference, relative quantification of controls and ligand samples was performed on the basis of ion intensity. Differential protein abundance was tested using a statistical ANOVA model followed by multiple testing corrections. The results are presented as a volcano plot, with the *X*-axis representing the mean ratio fold change (on a log2 scale) and the *Y*-axis representing the statistical significance *P* value of the ratio fold change for each protein (on a $-\log_{10}$ scale). The criteria for considering a protein as a candidate for interacting with the ligand of interest were an adjusted *P* value < 0.01 and a fold change >3.5 for NERP-4 compared to NERP-2, according to the manufacturer's recommendation[33].

## Stable expression of SNAT family in HEK293 cells
Human SNAT2/SLC38A2 (Myc-DDK–tagged)-CMV 6-entry vector was purchased from Origene (RC201892). Human SNAT3, SNAT4 and SNAT5 were amplified by PCR and inserted into the *Bgl*II and *Xho*I sites of pCMV6-Entry with an In-Fusion HD Cloning Kit (Z9640N, Takara Bio). These constructs were transfected into HEK293 cells with ViaFect (Promega). The transfected cells were cultured in a medium containing 0.5 mg/mL G418 (Promega) for 2 weeks, and single colonies were subsequently isolated.

## Binding assay
SNAT2-expressing HEK293 cells were incubated in HEPES buffer (DMEM containing 0.1% bovine serum albumin, 0.1% NaN₃ and 50 mM HEPES (pH 7.4)) containing 0.5–16 nM [¹²⁵I]-Y-NERP-4[8–19] with or without unlabelled NERP-4 for 1 h at 37 °C. The radioactivity of cell lysates was measured with an automatic γ-counter (Accu-FLEXγ ARC-7001, Hitachi). Nonspecific binding was determined in the presence of a 500-fold excess of unlabelled NERP-4. Cell membrane proteins (4 μg) were extracted from SNAT2-expressing HEK293 cells, as described previously[76]. Cells were scraped into HEPES buffer (25 mM HEPES, pH 7.4, 10 mM MgCl₂, and 0.25 M sucrose) and homogenized on ice. The homogenate was centrifuged at $800 \times g$ at 4 °C for 20 min and the supernatant was centrifuged twice at $100,000 \times g$ at 4 °C for 1 h. The membrane proteins were incubated with 2 nM [¹²⁵I]-Y-NERP-4[8–19] for 3 h at 25 °C. Bound [¹²⁵I]-Y-NERP-4[8–19] was isolated by vacuum filtration, and the radioactivity was measured. SNAT3-, SNAT4-, or SNAT5-expressing HEK293 cells were also studied in the above binding assay.

## Transport assay and inhibition experiments
MIN6-K8 cells and islets were preincubated with HKRB or HBSS buffer supplemented with 2.8 mM glucose for 1 h at 37 °C. They were incubated for 2, 5, or 10 min with HKRB or HBSS buffer containing 2.8 mM or 16.7 mM glucose and 1 μM [¹⁴C]-ʟ-glutamine (NEC451, PerkinElmer), 1 μM [¹⁴C]-ʟ-alanine (MC-466, Moravek), or 1 μM [³H]-ʟ-proline (NET483001MC, PerkinElmer) with or without NERP-4 (MIN6-K8 cells, $10^{-10}$ M and islets, $10^{-8}$ M). In an inhibition experiment of SNAT2, 10 mM MeAIB was administered to MIN6-K8 cells for 30 min. Uptake of glutamine, alanine, and MeAIB into si*Snat2* MIN6-K8 cells was also studied. The kinetic parameters of SNAT2 uptake activation in MIN6-K8 cells were determined by the uptake of [¹⁴C]-MeAIB (NEC671050UC, PerkinElmer) for 2 min at concentrations of 1, 10, 100, 500, 1000, and 2000 μM in HKRB buffer containing 16.7 mM glucose, with or without NERP-4 ($10^{-10}$ M). The uptake values were plotted against [¹⁴C]-MeAIB concentration and fitted to a Michaelis–Menten curve. The Michaelis constant (Km) and the maximum velocity values (Vmax) were determined by nonlinear regression using the enzyme kinetics module of GraphPad Prism 7 statistical software (GraphPad). Radioactivity was measured in Tri-Carb 2810TR (PerkinElmer).

## Amino acid determination

MIN6-K8 cells were preincubated for 1 h in HKRB containing 2.8 mM glucose. To measure amino acids released into the medium, MIN6-K8 cells (80–90% confluency in a 100 mm cell culture dish) were incubated for 30 min in HKRB containing 2.8 mM or 16.7 mM glucose. Supernatants were applied to 30 kDa Amicon Ultra (Merck Millipore) to remove albumin, and the eluates were subsequently lyophilised. To measure the intracellular contents of amino acids in MIN6-K8 cells ($4 \times 10^6$ cells/well in a 6-well plate), they were incubated for 5 min in HKRB containing 16.7 mM glucose with or without NERP-4. After washing the cells 3 times with cold PBS, the cell lysates were collected with RIPA buffer (Nacalai Tesque) and stored at −80 °C until used. Amino acids were determined by liquid chromatography/mass spectrometry (SRL Inc.).

## Administration of NERP-4 to db/db mice

*db/db* mice (8-week-old male, $n = 35$ for each group) were subcutaneously injected with NERP-4 (600 nmol/kg) or saline once daily (16:00) for 2 weeks. Blood samples were collected from their tail veins after a 16-h fast once per week. Two weeks after the start of NERP-4 or saline administration, *db/db* mice were intraperitoneally administered glucose (0.5 g/kg BW) or insulin (1.5 U/kg BW, Humulin, Eli Lilly) after a 16-h fast. Blood was collected at time points from 0 to 120 min after administration.

## TEM

Pancreata of *db/db* mice were fixed with a mixture of 2% paraformaldehyde and 2.5% glutaraldehyde overnight at 4 °C. They were then embedded in epoxy resin. Ultrathin sections mounted on gold meshes were stained with uranyl acetate–lead citrate and examined using an HT7700 transmission electron microscope (Hitachi)[6]. Sizes of insulin granules (saline, 594; and NERP-4, 305) and mitochondria (saline, 386; and NERP-4, 417) of each of three *db/db* mice administered NERP-4 or saline were measured.

## Mitochondrial respiration

The OCR was determined using an Extracellular Flux Analyzer XFe (Agilent Technologies). MIN6-K8 cells ($3 \times 10^4$ cells/well) were placed in minimal XF assay medium (Agilent Technologies) containing 16.7 mM glucose with palmitate (0.5 mM) or palmitate plus NERP-4 ($10^{-10}$ M) for 48 h. Basal respiration, ATP-coupled respiration, proton leakage and maximal respiratory capacity were measured by adding their respective inhibitors as described elsewhere[77]: the electron transport chain inhibitor oligomycin (4 μM), the proton ionophore FCCP (2 μM), rotenone (5 μM) and antimycin A (5 μM) (Agilent Technologies). All OCR measurements were corrected for non-mitochondrial OCR.

## Measurement of ROS production

Naïve, siSCR-treated, or *Snat2* knockdown MIN6-K8 cells were seeded on a 96-well black plate (Costar, Corning). The cells were treated with 0.5 mM palmitate in DMEM containing 2 mM glutamine for 48 h, and were administered $10^{-10}$ M NERP-4 once daily during palmitate treatment. The cells were loaded with 50 μM 2,7-dichlorodihydrofluorescein diacetate (Dojindo) for 30 min at 37 °C. Fluorescence intensity was measured using a VICTOR Nivo (PerkinElmer) with excitation at 488 nm and emission at 522 nm.

## Statistical analysis

Statistical analyses were performed with GraphPad Prism 7 statistical software (GraphPad) using two-way ANOVA followed by Bonferroni's post-test for multiple comparisons, One-way ANOVA and Tukey's multiple comparisons test, one-way ANOVA and Fisher's LSD test, and the unpaired two-tailed Student's *t*-test. Outliers were identified by ROUT method with Prism in all experiments. The statistical results were the same in all the experiments when outliers were included or excluded. We showed the figures that include outliers. All data are expressed as means ± s.e.m. $P < 0.05$ was considered statistically significant.

## Reporting summary

Further information on research design is available in the Nature Portfolio Reporting Summary linked to this article.

## Data availability

The data needed to reproduce the findings described in this manuscript can be found in the manuscript figures and supplementary materials. Source data are provided with this paper.

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

## Acknowledgements
The authors thank Mikiya Miyazato and Kenji Mori (National Cerebral and Cardiovascular Center Research Institute) for suggesting the useful methods employed in this study. The authors thank Stephen R.J. Salton (Mount Sinai School of Medicine, NY) for providing *Vgf*$^{-/-}$ mice, and Junichi Miyazaki (Osaka University) for providing MIN6-K8 cells. The authors thank Yoshiteru Goto, Itsuki Morinaga, and Eiko Kurata (University of Miyazaki) for their technical support. Part of this work was performed at the Frontier Science Research Centre, University of Miyazaki. This study was supported in part by the Japan Society for the Promotion of Science (JSPS) KAKENHI (25293216, and 15K09439) and the Agency for Medical Research and Development–Core Research for Evolutional Science and Technology (AMED-CREST, 19gm0610016h0006) (to M.N.).

## Author contributions
W.Z., A.Miura., A.Moin., H.S., N.Minamino. and M.Nakazato. designed the study; W.Z., A.Miura., A.Moin., K.Shimizu., Y.M., R.T., S.Hirako., S.S., C.J., Y.K., K.Sasaki., N.Minamino., V.G., J.K.C. and F.P. conducted the experiments and analysed the data; W.Z., A.Miura., H.S., and M.Nakazato. wrote the manuscript. All authors commented on and approved the final version of the manuscript. M.Nakazato. is the guarantor of this work, has full access to all the data in the study, and takes responsibility for the integrity and accuracy of the data.

## Competing interests
The authors declare no competing interests.

## Additional information

[1]Department of Bioregulatory Sciences, Faculty of Medicine, University of Miyazaki, Miyazaki, Japan. [2]Division of Neurology, Respirology, Endocrinology and Metabolism, Department of Internal Medicine, Faculty of Medicine, University of Miyazaki, Miyazaki, Japan. [3]Department of Health and Nutrition, University of Human Arts and Sciences, Saitama, Japan. [4]Department of Clinical Pharmacy, Faculty of Pharmaceutical Sciences, Shonan University of Medical Sciences, Yokohama, Japan. [5]Université de Lille, Inserm, Campus Hospitalo-Universitaire de Lille, Institut Pasteur de Lille, U1190-EGID, F-59000 Lille, France. [6]Department of Bio-system Pharmacology, Graduate School of Medicine, Osaka University, Osaka, Japan. [7]Department of Peptidomics, Sasaki Foundation, Tokyo, Japan. [8]Department of Molecular Pharmacology, National Cerebral and Cardiovascular Center Research, Suita, Japan. [9]Institute for Protein Research, Osaka University, Osaka, Japan. [10]AMED-CREST, Japan Agency for Medical Research and Development, Tokyo, Japan. [11]Present address: Department of

Pharmacology, Faculty of Medicine, University of Miyazaki, Miyazaki, Japan. [12]Present address: Department of Postgraduate Studies and Research, Royal College of Surgeons in Ireland – Bahrain, Busaiteen, Bahrain. [13]Present address: Division of Hematology, Diabetes, and Endocrinology, Department of Internal Medicine, Faculty of Medicine, University of Miyazaki, Miyazaki, Japan. [14]Present address: Systems Life Sciences Laboratory, Department of Medical Life Systems, Faculty of Life and Medical Sciences, Doshisha University, Kyoto, Japan. [15]Present address: Department of Endocrinology and Metabolism, Kanazawa University Graduate School of Medical Sciences, Kanazawa, Japan. [16]These authors contributed equally: Weidong Zhang, Ayako Miura. ✉e-mail: nakazato@med.miyazaki-u.ac.jp

