## [Peer Review File · Nature Communications]

The NERP-4–SNAT2 axis regulates pancreatic β -cell maintenance and functionREVIEWER COMMENTS

Reviewer #1 (Remarks to the Author):

The contribution by Zhang et al. concerns the description of a protective effect on beta-cell function by a VGF-derived peptide, NERP-4. NERP-4 increases intracellular Ca²⁺, enhances glucose-stimulated insulin secretion and promotes β -cell survival and function, lowering oxidative and ER stress. Authors propose that these effects are mediated by the stimulation of the neutral amino acid transporter SNAT2. Authors describe an increase in glutamine and alanine uptake caused by NERP-4, an effect suppressed, along with metabolic protection, in cells with SNAT2 knocked-down. From these results, Authors propose a novel, NERP-4 and SNAT2 mediated autocrine mechanism of β -cell maintenance and functional stimulation.

The novelty of the study is due to the putative identification of an endogenous positive allosteric activator of the SNAT2 transporter and to the proposal of a thus far unknown role of the axis NERP-4/SNAT2 in beta cell maintenance and function. Besides clarifying another mechanism for amino acid-dependent regulation of insulin secretion, the axis NERP-4/SNAT2 may potentially yield novel therapeutic targets for diabetes.

However, although these elements support a potentially significant impact of these data and hypotheses, the study presents several points that elicit my criticism.

1) Under several experimental conditions adopted, insulin secretion is stimulated. Yet, with the exception of results shown in Figs. 2 and 3, insulin levels are not checked. How can authors exclude that the effects recorded involve also insulin?

2) Many effects are only checked or inferred from changes in gene expression, several of which are really modest. An evaluation of protein expression should be added.

3) Moreover, in some instances, the functional consequences of changes in gene expression are simply presumed but not demonstrated. For instance, a 50% increase in caspase-3 expression can really explain the decrease in cell viability and the increase in TUNEL positivity detected? Actually, caspase-3 activation, rather than expression, should be linked to cell death. The same kind of argument should be raised for Nrf2.

4) The only Figure in which the effects of NERP-4 are evaluated under control conditions is Figure 4c. In this Figure, the peptide increases ATP independently of palmitate, and the effect is roughly comparable in palmitate-treated and untreated models, casting some doubts on the specificity of the claimed anti-lipotoxic effect. I think that this kind of control

(i.e. NERP-4 effect under not-stressed conditions) should be made whenever stressful conditions are adopted.

5) The expression of Snat2-5 is reported in Fig. 8. What about Snat1? It is claimed that this is present in the Discussion (line 241) but no data are reported. Given the very high homology between SNAT1 and SNAT2, this omission is important for a correct interpretation of the results presented in Figure 8f.

6) Authors report an increase in Gln transport induced by NERP-4. Yet, in many cell models Gln interacts with several transporters. On the other hand, MeAIB only inhibits a portion of Gln transport (Fig. 9, j, k). What are the other transporters involved? Do they change upon NERP-4 stimulation?

7) SNAT2 is claimed to be operative in glutamine and alanine (+ cysteine) transport. Yet, the specificity of the transporter is much larger. For instance, a preferential natural substrate of the transporter is Pro, which poorly interacts with most of the other transporters for neutral amino acids, thus constituting a reliable indicator of SNAT2 activity. Have the authors any data about changes in Pro transport induced by NERP-4?

8) Have the authors any evidence that NERP-4 effectively change the intracellular concentration of amino acids? Have they measured the cell content of Gln (or Ala) in NERP-4-treated models?

9) The results reported in Fig. 9e are quite expected, and cannot be taken per se as an evidence for a process coupled with insulin secretion. Obviously, when incubated in amino acid-free saline, amino acids go out of cells down their transmembrane gradient. Have the authors any evidence that the amino acid concentrations they measure outside are larger in conditions of insulin secretion stimulated (e.g. high glucose) compared to control (low glucose)? Moreover, an interesting functional proof of their model would be to demonstrate that this efflux is stimulated by NERP-4. Have the authors checked other amino acids? In other words, the three amino acid reported are the only released?

10) More importantly, authors state that NERP-4 stimulates GSIS even under amino acid-free conditions (line 250-251). Thus, how can SNAT2-mediated Gln transport works under those conditions? It is very unlikely that the extremely low quantities of Gln released from cells (in the order of microM) can justify a significant influx mediated by SNAT2 (whose K_m for Gln is > 1 mM).

11) It has been known since many years that SNAT2 is insulin-sensitive. This notion is briefly

mentioned (line 258) but it should be more extensively discussed. Is there any evidence (original or in literature) that SNAT2 is induced by insulin in beta cells?

12) From the results presented, it is tentative to assume two different time schedules for NERP-4 effects. One, rapid, for the stimulation of insulin secretion, attributable to the allosteric mechanism proposed in the manuscript, and the other, slower, for effects on cell viability and metabolism. Under this regard, it would be very interesting to know if NERP-4 stimulates the expression of SNAT2? This is an easy question but, apparently, there is no clear answer. The only data are, apparently, those in Fig. 10, where, however, NERP-4 effects are seen in palmitate-treated islets (again the experiment in control conditions is missing) or in db mice. What about human cells and normal conditions or models?

13) If Gln uptake in beta cells were mainly due to SNAT2 operation (but this, however, awaits experimental demonstration), then the effects of SNAT2 knockdown (Fig. 10) would be those generically expected for an amino acid deprivation response rather than specifically attributed to an interference with NERP-4 effects.

Minor

1) Results. The first sentence of the second paragraph (line 103) refers to results presented in the first paragraph and should be omitted.

2) The characteristics of MIN6-K8 cells should be briefly given, especially as far as the responsiveness of their insulin secretion to amino acids is concerned

3) "several orders of magnitude" should be cautionary "2-3 orders of magnitude" (line 100)

Reviewer #2 (Remarks to the Author):

In this interesting and well executed study, Zhang and coworkers focused on a previously unknown role of the VGF-derived peptide NERP4 on beta cell function. The Nakazato lab identified NERP4 recently, and now they demonstrate that the peptide regulate amino acid uptake and insulin secretion. They also identify SNAT2 as a binding partner of NERP4 and investigate its role in beta cell biology.

The manuscript has several strengths including the rigorous experimental design and experiment execution. Additionally, the role on SNAT2 in NERP4 activity, and more in

general in beta cell biology and insulin secretion, is novel and very interesting. The study includes several independent beta cell lines - from human and mice - as well as in vivo data in relevant diabetic models. A large number of techniques are used to test the role of NERP4.

However, there are a few major and some minor concerns that need to be addressed, in my opinion.

Major:

1) Figure 1 shows that NERP4 mediates rapid calcium influx. However, how is unclear. Critically, the relationship between NERP4-mediated calcium influx and NERP4-mediated activation of SNAT2 is uncertain. SNAT2 is an AA-Na²⁺ exchanger but I don't think that SNAT2 activation can elicits calcium influx directly. Conversely, the kinetic of calcium flux in figure 1 is compatible with cation channel activation mechanism (ion channel coupled to another GPCR? Ion channel receptor?), while it appears incompatible with an indirect mechanism as proposed in in fig 10i. The major problem here is that if calcium influx is independent from SNAT2, NERP4 might activate a second binding partner (a membrane receptor?), and thus the mechanism proposed here for insulin secretion remains incompletely characterized. This must be ruled out, e.g. by showing that calcium influx is abrogated by SNAT2 deletion? Otherwise it must be recognized that NERP4 mediated effects are not only mediated by SNAT2 but by other mechanisms as well (see #3).

2) The identification of a protein-mediated mechanism for NER4 is very exciting because for only 1 other VGF-derived peptide, i.e. TLQP-21, a receptor-mediated mechanism has been identified (see <https://pubmed.ncbi.nlm.nih.gov/34626205/>). Thus, this project fills an important gap of knowledge in the field. However, the evidence that NERP4's sole binding partner is SNAT2 is not that strong in fig 8a; and its role as an allostatic activator is weak (see #4). Also, while pharmacological data convincingly demonstrate that SNAT2 activity is required for NERP4 biological effect, the evidence that NERP4-mediates amino acid uptake and insulin secretion is exerted uniquely via SNAT2 is not that strong (Fig 9f only).

3) Several C-terminal VGF-derived peptides (TLQP-62 in particular) are potent insulin secretagogues and mediate GSIS as well as they increase VGF mRNA expression in beta cells (<https://pubmed.ncbi.nlm.nih.gov/25917832/>). The mechanism and relevance of the autocrine effect of VGF peptides on VGF expression is still unclear. However, authors should quantify VGF mRNA (in addition to Ins and other genes) in all of their experiments (in vivo and in vitro). If VGF mRNA is indeed induced by NERP4 [and thus all VGF peptides are potentially produced and secreted] as well, some of its long-term effects (in vitro and in vivo) can be mediated by other VGF-derived peptides as well.

4) Much emphasis is given in the discussion to NERP4 being a PAM. However, this conclusion is not supported by rigorous data, but only inferred from indirect evidence (Fig 9k). At the minimum an in-silico model of the mode of binding on SNAT2 (+/- MeAIB)-NERP4 should be presented if authors want to sustain this conclusion.

Specific comments:

5) The result section is difficult to read. Authors are encouraged to use a more narrative style of presenting the results.

6) The section on ER stress and mito dysfunction is not well connected to the rest of the story. I'm not sure it is really required to support the main conclusion and mechanism.

7) Page 9. The "criteria for specific binding" is not explained in the manuscript (or I cannot find it). As I mentioned above the specificity/selectivity of NERP4-SNAT2 binding is not very convincing in my opinion.

8) The statistical section describes the use of Bonferroni post hoc while the figure legends describe the Tukey. Please clarify.

9) Fig 1k seems irrelevant, it is quite expected that beta cells express insulin to a level higher than VGF.

Reviewer #3 (Remarks to the Author):

This is an interesting and novel manuscript describing studies demonstrating that the peptide NERP-4, which is secreted from pancreatic beta cells can regulate various aspects of beta cell health and function by interacting with an amino acid transporter, SNAT2 in beta

cells. NERP-4 was discovered by the authors and it has many actions, supposedly all due to receptor-induced changes in amino acid flux into beta cells.

While the paper is of interest, there are a number of weaknesses that limit enthusiasm for the current version, as detailed below.

1. While the manuscript describes potentially mechanistic studies, the calcium studies shown early in the paper (Fig 1) are insufficiently described and their importance is not clear for the secretory phenotype. For example, in Fig 1b it is shown that the addition of NERP-4 raises free calcium in 'pancreatic extracts', leading to a sharply rising spike that declines over 100 seconds. It is not clear what the preparation is here. Do they mean 'islets' or cells from the pancreas of some kind? Plus, how was a dose of 1 μ M chosen here while the dose used in the center panel (on MIN6-K8, what is K8, a subclone?) was only 0.1 nM? The MIN6-K8 run was carried out in 22 mM glucose, which is well above the physiological range and yet no transients or oscillations in MIN6 calcium were seen during the first measurement period, which is odd. The authors also present no evidence that the rise in Ca to NERP-4 is due to intracellular Ca release and not influx—or both? Properly, the experiment should be run in Ca free media to ensure the rise in Ca is due to release. As the authors no doubt know, Ca is not the only regulator of glucose induced insulin secretion and it is possible that NERP-4 affects not the triggering but the amplification pathway.

2. Another problem with the data in Fig. 1 is that even if true, it is not clear what the functional role of the Ca rise is for the data shown in the rest of the paper? Thus, is the Ca rise the result of binding of NERP-4 to the amino acid transporter and amino acid flux or is it unrelated and due to another pathway? What does preventing the Ca rise do to the subsequent measurements of NERP-4 action? It is unclear.

3. Throughout the paper, plots are shown that appear to describe underpowered studies and what is more, many outliers can be seen, usually above the mean values shown. The analyses are thus not credible without assessing the impact of these outliers to determine whether they are true outliers and their removal does not result in a loss of significance.

4. There seems to be a plethora of possible effects that are not considered. For the whole animal studies shown in Fig. 6 for example, does NERP-4 treatment affect food intake? Glucagon levels, body weight?

5. There does not seem to be a clear explanation for the relationship between doses of NERP-4 used in binding studies to the various functional studies; this should be discussed in detail.

6. With regards to the hypothesis put forward, could the need for NERP-4 be mitigated by increasing the exposure to amino acid concentrations tested?

7. Minor: the authors should have the paper edited by a native English speaker to improve the grammar. I was struck in many places where the sentences used I think conveyed the exact opposite meaning than intended, unfortunately. For instance, they say "....restored these alterations" when I think the authors meant "reversed or prevented" which adds confusion.

Dear reviewers of *Nature Communications*,

We sincerely appreciate your careful reading of our manuscript and your valuable comments and advice, based on which we have revised our original version. We have also expanded several experiments and have provided details in the current version of the manuscript. Our responses to the reviewers' comments are listed below.

Reviewer 1

1) Under several experimental conditions adopted, insulin secretion is stimulated. Yet, with the exception of results shown in Figs. 2 and 3, insulin levels are not checked. How can authors exclude that the effects recorded involve also insulin?

Reply: As the reviewer mentioned, insulin itself has effects on β cells. We presented insulin levels in the revised manuscript (Fig. 4i, Fig. 6d, Fig. 9k–n, Extended Data Fig. 2b, Extended Data Fig. 4a, Extended Data Fig. 8c). In all these experiments, NERP-4 increased insulin in isolated human and mouse islets and MIN6-K8 cells, and we agree with the comment that the effects of NERP-4 involve insulin. We described these results below and added Fig. 4i, Fig. 6d, Fig. 9k–n, Extended Data Fig. 2b, Extended Data Fig. 4a, and Extended Data Fig. 8c.

Lines 109–111 in Text

Co-administration of NERP-4 and NERP-2[VGF 310–347], another VGF-derived insulinotropic peptide, to *Vgf* knockdown MIN6-K8 cells additively increased GSIS (Extended Data Fig. 2a, b).

Lines 150–152 in Text

Palmitate reduced ATP content, the mRNA levels of the ATP synthases *Atp5e* and *Atp5j2*, and GSIS in MIN6-K8 cells (Fig. 4c–e, Extended Data Fig. 4a), while NERP-4 reversed these reductions (Fig. 4c–e, Extended Data Fig. 4a).

Lines 172–175 in Text

Two weeks after NERP-4 administration, a glucose tolerance test (GTT) demonstrated lowered blood glucose and elevated plasma insulin (Fig. 6c, d), but an insulin tolerance test (ITT) showed no alteration in insulin sensitivity (Extended Data Fig. 5c).

Lines 232–238 in Text

Given that NERP-4 increased GSIS by stimulating amino acid uptake, we assumed that β -cell exposure to higher glutamine concentrations could mitigate the effects of NERP-4 on GSIS. We studied the insulinotropic activity of NERP-4 in MIN6-K8 cells at glutamine concentrations from 1 μ M to 1 mM in amino acid-free HEPES Krebs–Ringer bicarbonate (HKRB) buffer. Glutamine concentrations equal

to or higher than 1 μ M increased GSIS in a concentration-dependent manner (Extended Data Fig. 8c).

NERP-4 augmented GSIS at glutamine concentrations between 1 μ M and 100 μ M, but not at 1 mM glutamine concentration (Extended Data Fig. 8c).

Lines 240–241 in Text

MeAIB also abolished NERP-4–induced GSIS in human and C57BL/6J mouse islets (Fig. 9m, n).

Lines 309–311 in Discussion

Insulin itself has protective effects on β cells⁴⁴. In the present study, NERP-4 recovered the suppression of insulin secretion caused by metabolic stress. NERP-4–induced insulin secretion could contribute to the effect of NERP-4 on β -cell maintenance.

2) Many effects are only checked or inferred from changes in gene expression, several of which are really modest. An evaluation of protein expression should be added.

Reply: As the reviewer mentioned, changes in protein expression or activity are meaningful. We added western blot studies to determine the protein expression of CHOP and SOD2 in isolated C57BL/6J mouse islets (Fig. 3f) and *db/db* mouse islets (Fig. 7f). NERP-4 increased the SOD2 protein level and reduced the CHOP protein level in palmitate-treated C57BL/6J mouse islets and *db/db* mouse islets. We also determined the protein expression of cleaved caspase-3 and the translocation of Nrf2, as mentioned in our response to the next comment (# 3). We added these results in the text and in Fig. 3f and Fig. 7f.

Lines 130–133 in Text (palmitate-treated C57BL/6J mouse islets)

Palmitate increased the protein and mRNA levels of the ER stress marker CHOP, while NERP-4 reduced them (Fig. 3f, Extended Data Fig. 3). Palmitate reduced the protein and mRNA levels of the antioxidative response marker SOD2, while NERP-4 reversed these alterations (Fig. 3f, Extended Data Fig. 3).

Lines 186–188 in Text (*db/db* mouse islets)

NERP-4 reduced the protein and mRNA levels of CHOP (Fig. 7f, Extended Data Fig. 6), and increased the protein and mRNA levels of SOD2 and the mRNA level of *Nrf2* (Fig. 7f, Extended Data Fig. 6).

3) Moreover, in some instances, the functional consequences of changes in gene expression are simply presumed but not demonstrated. For instance, a 50% increase in caspase-3 expression can really explain the decrease in cell viability and the increase in TUNEL positivity detected? Actually, caspase-3 activation, rather than expression, should be linked to cell death. The same kind of argument should be raised for Nrf2.

Reply: mRNA of the apoptotic marker *caspase-3* was upregulated in cytokine-treated MIN6-K8 cells (original version Fig. 4g). As the reviewer pointed out, the inactive zymogen of caspase-3 is activated by proteolytic cleavage. We added a western blot analysis of cleaved caspase-3 to examine its protein expression (Fig. 4h). Cytokine treatment increased cleaved caspase-3 protein and NERP-4 reduced it (Fig. 4h). Nrf2 translocates from the cytosol to the nucleus in response to oxidative stress, thereby inducing antioxidant genes. In the revised manuscript, we studied the translocation of Nrf2 protein in palmitate-treated mouse islets by western blot analysis. Palmitate reduced Nrf2 translocation to the nucleus and *Nrf2* mRNA in C57BL/6J mouse islets (Fig. 3g, Extended Data Fig. 3). NERP-4 reversed these reductions (Fig. 3g, Extended Data Fig. 3). These results imply that NERP-4-induced alterations

of cleaved caspase-3 and Nrf2 could be involved in reducing apoptosis and oxidative stress facilitated by NERP-4. We added these results in the text and in Fig. 3g and Fig. 4h.

Lines 134–136 in Text

Palmitate reduced Nrf2 translocation to the nucleus and *Nrf2* mRNA level (Fig. 3g, Extended Data Fig. 3). NERP-4 reversed these alterations (Fig. 3g, Extended Data Fig. 3).

Lines 159–162 in Text

A cytokine cocktail (tumour necrosis factor- α , interleukin-1 β , and interferon- γ) upregulated the apoptotic marker protein cleaved caspase-3 and its mRNA, reduced GSIS and cell viability, and induced cell death in MIN6-K8 cells (Fig. 4h–k, Extended Data Fig. 4b). NERP-4 reversed these alterations.

4) The only Figure in which the effects of NERP-4 are evaluated under control conditions is Figure 4c. In this Figure, the peptide increases ATP independently of palmitate, and the effect is roughly comparable in palmitate-treated and untreated models, casting some doubts on the specificity of the claimed anti-lipotoxic effect. I think that this kind of control (i.e. NERP-4 effect under not-stressed conditions) should be made whenever stressful conditions are adopted.

Reply: In the original version, we showed the effects of NERP-4 on oxidative and ER stress in palmitate-treated mouse islets (original version Fig. 3b–g) and palmitate- (original version Fig. 4d, e) or cytokine- (original version Fig. 4g) treated MIN6-K8 cells. We agree with the reviewer’s comment and studied the effects of NERP-4 under non-stressed conditions in isolated C57BL/6J mouse islets and MIN6-K8 cells. NERP-4 increased each of the following in C57BL/6J mouse islets: mRNA expression of *Vgf*, *Sod2*, and *Nrf2*; the protein level of SOD2; and Nrf2 protein translocation to the nucleus (Fig. 3e–g, Extended Data Fig. 3). NERP-4 increased mRNA expression of *Vgf*, *Atp5e*, and *Atp5j2* and cell viability in MIN6-K8 cells (Fig. 4d–f, j). NERP-4 decreased production of ROS (Fig. 4g). We added these results in the text and in Fig. 3e–g, Fig. 4d–f, Fig. 4j, and Extended Data Fig. 3.

Lines 136–140 in Text

Under non-stressed conditions, NERP-4 administration to isolated C57BL/6J mouse islets increased the mRNA levels of *Vgf*, *Sod2*, and *Nrf2* (Fig. 3e, Extended Data Fig. 3). NERP-4 also decreased the

protein and mRNA levels of CHOP (Fig. 3f, Extended Data Fig. 3), increased the SOD2 protein level (Fig. 3f), and enhanced Nrf2 nuclear translocation (Fig. 3g).

Lines 154–156 in Text

NERP-4 administration to naïve MIN6-K8 cells also increased ATP content, the mRNA levels of *Atp5e* and *Atp5j2*, and cell viability, and decreased ROS production (Fig. 4c–e, g, j).

Lines 350–351 in Discussion

Under both stressed and non-stressed conditions, NERP-4 administration to MIN6-K8 cells improved β -cell viability and function.

5) The expression of *Snat2-5* is reported in Fig. 8. What about *Snat1*? It is claimed that this is present in the Discussion (line 241) but no data are reported. Given the very high homology between SNAT1 and SNAT2, this omission is important for a correct interpretation of the results presented in Figure 8f.

Reply: We determined the mRNA expression of *Snat1* in addition to that of *Snat2* to *Snat5* in C57BL/6J mouse islets and MIN6-K8 cells (Fig. 8e). *Snat1* was not expressed in either sample, consistent with a previous study in which no expression of *Snat1* was observed in mouse islets or MIN6-K8 cells (Reference 22 below). We added the results for *Snat1* mRNA in Fig. 8e.

Lines 206–209 in Text

SNAT1, SNAT2, and SNAT4 are system A amino acid transporters that facilitate the uptake of neutral amino acids³⁴. We detected the mRNAs of *Snat2*, *Snat3*, *Snat4*, and *Snat5*, but not *Snat1*, in C57BL/6J mouse islets and MIN6-K8 cells, consistent with a previous study²² (Fig. 8e).

Reference 22. Hashim, M., et al. Inhibition of SNAT5 induces incretin-responsive state from incretin-unresponsive state in pancreatic β -cells: Study of β -cell spheroid clusters as a model. *Diabetes* **67**, 1795–1806 (2018)

6) Authors report an increase in Gln transport induced by NERP-4. Yet, in many cell models Gln interacts with several transporters. On the other hand, MeAIB only inhibits a portion of Gln transport (Fig. 9j, k). What are the other transporters involved? Do they change upon NERP-4 stimulation?

Reply: As the reviewer mentioned, glutamine and alanine are imported via several transporters in many types of cells. In rat islets, glutamine is transported by SNAT2, SNAT4, ASCT2 (SLC1A5), and B⁰AT1 (SLC6A19) (Reference 5 below). We performed new experiments by SNAT2 inhibition and *Snat2* knockdown and verified that NERP-4 acted only on SNAT2. α -Methylaminoisobutyric acid (MeAIB) is a non-metabolisable L-alanine analogue that inhibits system A transporter activity (Reference 35 below). We studied [¹⁴C]-L-glutamine and [¹⁴C]-L-alanine uptake into si*Snat2*- or MeAIB-treated MIN6-K8 cells. Glutamine and alanine uptake was reduced by 13.6% and 30.3%, respectively, in si*Snat2*-treated MIN6-K8 cells compared with siSCR MIN6-K8 cells (Fig. 9e, f). Glutamine and alanine uptake was reduced by 17.8% and 40.7%, respectively, in MeAIB-treated MIN6-K8 cells (Fig. 9g, h). These results suggest that other transporters also mediate glutamine and alanine uptake. NERP-4 did not increase glutamine or alanine uptake into si*Snat2*- or MeAIB-treated MIN6-K8 cells (Fig. 9e–h), implying that NERP-4 acts only on SNAT2 to stimulate amino acid uptake. We added these results in the text and in Fig. 9e–h.

Lines 220–230 in Text

Snat2 knockdown in MIN6-K8 cells abrogated NERP-4–induced [¹⁴C]-L-glutamine and [¹⁴C]-L-alanine uptake (Fig. 9e, f, Extended Data Fig. 8b). α -Methylaminoisobutyric acid (MeAIB), a non-metabolisable L-alanine analogue, is an orthosteric inhibitor of system A transporter³⁵. MeAIB also abrogated NERP-4–induced [¹⁴C]-L-glutamine and [¹⁴C]-L-alanine uptake (Fig. 9g, h). These results imply that NERP-4 acts only on SNAT2 to stimulate amino acid uptake. Glutamine and alanine are imported via several transporters in many types of cells⁵. In rat islets, SNAT2, SNAT4, ASC-1 (SLC1A10), ASCT2 (SLC1A5), and B⁰AT1 (SLC6A19) transport glutamine and alanine⁵. *Snat2* knockdown reduced glutamine and alanine uptake by 13.6% and 30.3%, respectively, compared with siSCR treatment (Fig. 9e, f). MeAIB reduced glutamine and alanine uptake by 17.8% and 40.7%, respectively (Fig. 9g, h). These results suggest that transporters other than SNAT2 also mediate glutamine and alanine uptake in β cells.

Reference 5. Bröer, S. Amino acid transporters as modulators of glucose homeostasis. *Trends Endocrinol. Metab.* **33**, 120–135 (2022).

Reference 35. Prentki M. & Renold A. E..Neutral amino acid transport in isolated rat pancreatic islets. *J. Biol. Chem.* **258**, 14239–14244 (1983).

7) SNAT2 is claimed to be operative in glutamine and alanine (+ cysteine) transport. Yet, the specificity of the transporter is much larger. For instance, a preferential natural substrate of the transporter is Pro, which poorly interacts with most of the other transporters for neutral amino acids, thus constituting a reliable indicator of SNAT2 activity. Have the authors any data about changes in Pro transport induced by NERP-4?

Reply: As the reviewer mentioned, proline is the preferential natural substrate of SNAT2. We studied the effect of NERP-4 on proline uptake in MIN6-K8 cells. NERP-4 increased proline uptake (Extended Data Fig. 8a), supporting the finding that NERP-4 acts on SNAT2 as a target. We added this result in the text and in Extended Data Fig. 8a.

Lines 216–217 in Text

Next, MIN6-K8 cells were administered NERP-4 with 1 μM [^{14}C]-L-alanine or 1 μM [^3H]-L-proline. NERP-4 also increased their intracellular contents (Fig. 9d, Extended Data Fig. 8a).

Lines 218–219 in Text

Among system A transporters, proline is the preferential natural substrate of SNAT2, but not SNAT1 or SNAT4³⁴.

8) Have the authors any evidence that NERP-4 effectively change the intracellular concentration of amino acids? Have they measured the cell content of Gln (or Ala) in NERP-4-treated models?

Reply: We administered 1 μM [^{14}C]-L-glutamine, [^{14}C]-L-alanine, or [^3H]-L-proline to MIN6-K8 cells in the presence or absence of NERP-4. We determined their intracellular contents 2 min after these administrations. NERP-4 increased the contents of glutamine, alanine, and proline (Fig. 9c, d, Extended Data Fig. 8a). We added the results in the text and in Fig. 9c, d, and Extended Data Fig. 8a.

Lines 213–217 in Text

We next investigated the effect of NERP-4 on amino acid transport activity. Human and C57BL/6J mouse islets and MIN6-K8 cells were administered NERP-4 with 1 μM [^{14}C]-L-glutamine. NERP-4 increased the intracellular contents of [^{14}C]-L-glutamine in both islets and MIN6-K8 cells (Fig. 9a–c). Next, MIN6-K8 cells were administered NERP-4 with 1 μM [^{14}C]-L-alanine or 1 μM [^3H]-L-proline. NERP-4 also increased their intracellular contents (Fig. 9d, Extended Data Fig. 8a).

9) The results reported in Fig. 9e are quite expected, and cannot be taken per se as an evidence for a process coupled with insulin secretion. Obviously, when incubated in amino acid-free saline, amino acids go out of cells down their transmembrane gradient. Have the authors any evidence that the amino acid concentrations they measure outside are larger in conditions of insulin secretion stimulated (e.g. high glucose) compared to control (low glucose)? Moreover, an interesting functional proof of their model would be to demonstrate that this efflux is stimulated by NERP-4. Have the authors checked other amino acids? In other words, the three amino acid reported are the only released?

Reply: We quantified amino acids in the media after GSIS experiments under low- (2.8 mM) or high- (16.7 mM) glucose conditions. We detected glutamine, alanine, proline, glutamate, leucine, arginine and other amino acids at similar concentrations in both low- and high-glucose media (Extended Data Table 1a). Glutamine, glutamate, alanine, leucine, and arginine were previously shown to stimulate insulin secretion (Reference 2 below). NERP-4 did not augment insulin secretion from isolated C57BL/6J mouse islets under 2.8 mM glucose, even when amino acids were present in the medium (Fig. 2b, c). Another research group demonstrated that 2 mM glutamine administration to MIN6-K8 cells under 2.8 mM glucose did not increase intracellular Ca^{2+} and had no effect on GSIS (Reference 23 below). Combined together, these findings indicate that under high-glucose conditions, NERP-4 functions as an insulinotropic peptide by enhancing the uptake of amino acids released from β cells. NERP-4 does not act on SNAT3 or SNAT5 which stimulates the efflux of glutamine from β cells. We

thus consider that glutamine efflux is independent of NERP-4. We added these results in the text and in Extended Data Table 1.

Lines 243–254 in Text

We used amino acid–free HKRB buffer in GSIS experiments. SNAT3, SNAT5, and ASCT2 function as exporters of glutamine under conditions in which extracellular glutamine is absent or very low^{36,37}.

To explore the source of amino acids that stimulate NERP-4–induced GSIS, we quantified amino acids released into the medium after GSIS experiments. MIN6-K8 cells released glutamine, alanine, proline, and other amino acids under both 2.8 mM and 16.7 mM glucose conditions (Extended Data Table 1).

The concentrations of these amino acids were comparable between low and high glucose conditions.

NERP-4 did not augment Ca^{2+} influx or GSIS under 2.8 mM glucose even when amino acids were present in the medium (Fig. 2b, c, Extended Data Fig. 1a). Another research group demonstrated that

2 mM glutamine administration to MIN6-K8 cells in medium containing 2.8 mM glucose did not affect

GSIS or Ca^{2+} influx²³. Together, these findings indicate that under high glucose conditions, NERP-4

functions as an insulinotropic peptide by enhancing uptake of amino acids released from β cells.

Reference 2. Newsholme, P., Bender, K., Kiely, A. & Brennan, L. Amino acid metabolism, insulin secretion and diabetes. *Biochem. Soc. Trans.* **35**, 1180–1186 (2007).

Reference 23. Han G., et al. Glutamate is an essential mediator in glutamine-amplified insulin secretion.

J. Diabetes Investig. **12**, 920–930 (2021).

10) More importantly, authors state that NERP-4 stimulates GSIS even under amino acid-free conditions (line 250-251). Thus, how can SNAT2-mediated Gln transport works under those conditions? It is very unlikely that the extremely low quantities of Gln released from cells (in the order of microM) can justify a significant influx mediated by SNAT2 (whose k_m for Gln is > 1 mM).

Reply: To respond to this important comment, we added the following experiments. First, we verified that amino acids released from β cells are imported through the activity of the NERP-4–SNAT2 axis. We administered $1 \mu\text{M}$ [^{14}C]-L-glutamine, [^{14}C]-L-alanine, or [^3H]-L-proline to MIN6-K8 cells in the presence or absence of NERP-4, and 2 min later determined their intracellular contents. NERP-4 increased the intracellular contents of these amino acids (Fig. 9c, d, Extended Data Fig. 8a). These findings imply that low quantities of amino acids (i.e., $1 \mu\text{M}$ concentration) are imported into β cells by NERP-4.

Next, we verified that NERP-4–induced amino acid uptake mediated Ca^{2+} influx and GSIS under amino acid–free conditions. *Snat2* knockdown abolished NERP-4–induced glutamine and alanine uptake (Fig. 9e, f), Ca^{2+} influx (Fig. 9i), and GSIS (Fig. 9k). MeAIB also abrogated NERP-4–induced glutamine and alanine uptake (Fig. 9g, h), Ca^{2+} influx (Fig. 9j), and GSIS (Fig. 9l). We conclude that NERP-4 acts on SNAT2 to cause amino acid uptake that induces Ca^{2+} influx and GSIS.

We further analysed the kinetic properties of concentration-dependent uptake of [^{14}C]-MeAIB in the range of $1 \mu\text{M}$ to 2mM in the presence or absence of NERP-4. [^{14}C]-MeAIB uptake showed a

good fit to a Michaelis–Menten curve, with a K_m of $881.1 \pm 220.7 \mu\text{M}$ and a V_{max} of $2,240 \pm 242.3$ pmol/mg protein/min (Fig. 9o). NERP-4 administration lowered the K_m of [^{14}C]-MeAIB ($412.6 \pm 113.5 \mu\text{M}$) without changing the V_{max} , implying that NERP-4 modulates the binding affinity of SNAT2 to amino acids. As we showed in the original manuscript, MeAIB, an orthosteric inhibitor of SNAT2, did not change [^{125}I]-Y-NERP-4[8–19] binding to MIN6-K8 cells (Extended Data Fig. 8f). This supports the finding that the NERP-4 binding site is different from the substrate binding site. Combined with the kinetics analysis and binding study, NERP-4 acts as a positive allosteric modulator (PAM).

Lastly, we confirmed that NERP-4 augmented GSIS in the presence of μM -order concentrations of glutamine in MIN6-K8 cells. One micromolar glutamine increased GSIS, and co-administration of NERP-4 and $1 \mu\text{M}$ glutamine further increased GSIS (Extended Data Fig. 8c). We thus think that the impact of NERP-4 on SNAT2 is significant under low quantities of glutamine. We added these results in Extended Data Table 1, Extended Data Fig. 8a, Fig. 9e, f, i–k, and o.

Lines 220–222 in Text

Snat2 knockdown in MIN6-K8 cells abrogated NERP-4–induced [^{14}C]-L-glutamine and [^{14}C]-L-alanine uptake in MIN6-K8 cells (Fig. 9e, f, Extended Data Fig. 8b).

Lines 236–242 in Text

Glutamine concentrations equal to or higher than $1 \mu\text{M}$ increased GSIS in a concentration-dependent

manner (Extended Data Fig. 8c). NERP-4 augmented GSIS at glutamine concentrations between 1 μ M and 100 μ M, but not at 1 mM glutamine (Extended Data Fig. 8c). Both *Snat2* knockdown and MeAIB abrogated NERP-4–induced Ca^{2+} influx (Fig. 9i, j, Extended Data Fig. 8d) and GSIS (Fig. 9k, l) in MIN6-K8 cells. MeAIB also abolished NERP-4–induced GSIS in human and C57BL/6J mouse islets (Fig. 9m, n). These findings reveal that NERP-4 acts specifically on SNAT2 to induce amino acid uptake, Ca^{2+} influx, and GSIS.

Lines 245–249 in Text

To explore the source of amino acids that stimulate NERP-4–induced GSIS, we quantified amino acids released into the medium after GSIS experiments. MIN6-K8 cells released glutamine, alanine, proline, and other amino acids under both 2.8 mM and 16.7 mM glucose conditions (Extended Data Table 1). The concentrations of these amino acids were comparable between low and high glucose conditions.

Lines 256–265 in Text

To study the kinetic properties of SNAT2 activation by NERP-4, we determined the concentration-dependent uptake of [^{14}C]-MeAIB in the range of 1 μ M to 2 mM at 2 min in the presence or absence of NERP-4. [^{14}C]-MeAIB uptake showed a good fit to a Michaelis–Menten curve, with a K_m of 881.1 ± 220.7 μ M and a V_{max} of $2,240 \pm 242.3$ pmol/mg protein/min (Fig. 9o). NERP-4 lowered the K_m of [^{14}C]-MeAIB (412.6 ± 113.5 μ M) without changing the V_{max} . Furthermore, MeAIB did not alter

[¹²⁵I]-Y-NERP-4[8–19] binding to MIN6-K8 cells (Extended Data Fig. 8f), supporting the finding that the NERP-4 binding site is different from the substrate binding site. Taken together, these results show that NERP-4 is considered to modulate the binding affinity of SNAT2 to amino acids, implying that NERP-4 acts as a positive allosteric modulator (PAM).

11) It has been known since many years that SNAT2 is insulin-sensitive. This notion is briefly mentioned (line 258) but it should be more extensively discussed. Is there any evidence (original or in literature) that SNAT2 is induced by insulin in beta cells?

Reply: As the reviewer pointed out, insulin induces SNAT2 in skeletal muscle cells, and this enhances amino acid supplies and insulin-dependent protein synthesis (Reference 38 below). We have not found any studies mentioning that SNAT2 is induced by insulin in β cells. We examined whether SNAT2 in β cells was induced by insulin. Insulin upregulated *Snat2* mRNA in naïve MIN6-K8 cells and isolated C57BL/6J mouse islets (Extended Data Fig. 9a, b). We added these results as follows.

Lines 268–270 in Text

Insulin increases protein and mRNA levels of SNAT2 to enhance the amino acid supply and insulin-dependent protein synthesis³⁸. We found that insulin increased *Snat2* mRNA in naïve MIN6-K8 cells and C57BL/6J mouse islets (Extended Data Fig. 9a, b).

Reference 38, Kashiwagi H, Yamazaki K, Takekuma Y, Ganapathy V, Sugawara M. Regulatory mechanisms of SNAT2, an amino acid transporter, in L6 rat skeletal muscle cells by insulin, osmotic shock and amino acid deprivation. *Amino Acids* **36**, 219–230 (2009).

12) From the results presented, it is tentative to assume two different time schedules for NERP-4 effects. One, rapid, for the stimulation of insulin secretion, attributable to the allosteric mechanism proposed in the manuscript, and the other, slower, for effects on cell viability and metabolism. Under this regard, it would be very interesting to know if NERP-4 stimulates the expression of SNAT2? This is an easy question but, apparently, there is no clear answer. The only data are, apparently, those in Fig. 10, where, however, NERP-4 effects are seen in palmitate-treated islets (again the experiment in control conditions is missing) or in db mice. What about human cells and normal conditions or models?

Reply: As summarized well by the reviewer, we hypothesize that the effects of NERP-4 occur during two different phases. In the rapid phase, NERP-4 stimulates amino acid uptake into β cells, thereby enhancing insulin secretion. In the slower phase, NERP-4 acts on β -cell viability and metabolism. To determine whether NERP-4 stimulates SNAT2 expression under both normal and stressed conditions in the slower phase, we performed new experiments.

We demonstrated that NERP-4 upregulated *Snat2* mRNA in naïve MIN6-K8 cells and isolated C57BL/6J mouse islets under non-stressed conditions (Extended Data Fig. 9c, d). NERP-4 also upregulated *Snat2* mRNA in palmitate- or cytokine-treated MIN6-K8 cells (Extended Data Fig. 9e, f). These findings indicate that NERP-4 stimulates SNAT2 expression under normal and stressed conditions. We added these results as shown below. We were unable to analyze human islets because relatively few pancreatic transplants are currently being performed.

Lines 270–281 in Text

NERP-4 also increased *Snat2* mRNA in MIN6-K8 cells and C57BL/6J mouse islets under non-stressed conditions (Extended Data Fig. 9c, d). SNAT2 is upregulated in β cells in diabetes as a response to the translational repression associated with ER stress⁴. We investigated *Snat2* expression in palmitate-treated C57BL/6J mouse islets and *db/db* mouse islets. The *Snat2* mRNA level in C57BL/6J mouse islets was increased by palmitate and was elevated to an even greater extent by NERP-4 (Fig. 10a). The *Snat2* mRNA level was higher in *db/db* mice compared with *db/+* mice (Fig. 10b). Two-week NERP-4 administration increased the *Snat2* mRNA level in *db/db* islets (Fig. 10c). The *Snat2* mRNA level was also increased in palmitate- or cytokine-treated MIN6-K8 cells after NERP-4 administration for 48 or 24 h, respectively (Extended Data Fig. 9e, f). SNAT2 upregulation induced by long-term NERP-4 administration under stress conditions could contribute to the effects of NERP-4 on β -cell viability and metabolism.

13) If Gln uptake in beta cells were mainly due to SNAT2 operation (but this, however, awaits experimental demonstration), then the effects of SNAT2 knockdown (Fig. 10) would be those generically expected for an amino acid deprivation response rather than specifically attributes to an interference with NERP-4 effects.

Reply: As the reviewer mentioned, amino acid uptake into β cells is carried out redundantly by several transporters. SNAT2, SNAT4, ASC-1 (SLC1A10), ASCT2 (SLC1A5), and B⁰AT1 (SLC6A19), all of which are expressed in β cells, transport glutamine and alanine (Reference 5 below). To explore whether glutamine and alanine uptake in β cells is mainly due to SNAT2, we studied amino acid uptake in MIN6-K8 cells treated by *Snat2* knockdown or MeAIB administration. *Snat2* knockdown reduced glutamine and alanine uptake by 13.6% and 30.3%, respectively, compared with control MIN6-K8 cells (Fig. 9e, f). MeAIB also reduced glutamine and alanine uptake by 17.8% and 40.7%, respectively (Fig. 9g, h). These results suggest that transporters other than SNAT2 also mediate glutamine and alanine uptake. We thus think that SNAT2 deletion and inhibition attributed to an interference with NERP-4 effects. We added these results in Fig. 9e–h.

Lines 219–231 in Text

To demonstrate that the effect of NERP-4 depends only on SNAT2, we performed SNAT2 deletion and inhibitor experiments. *Snat2* knockdown in MIN6-K8 cells abrogated NERP-4–induced [¹⁴C]-L-glutamine and [¹⁴C]-L-alanine uptake (Fig. 9e, f, Extended Data Fig. 8b). α -Methylaminoisobutyric acid (MeAIB), a non-metabolisable L-alanine analogue, is an orthosteric inhibitor of system A transporter³⁵. MeAIB also abrogated NERP-4–induced [¹⁴C]-L-glutamine and [¹⁴C]-L-alanine uptake (Fig. 9g, h). These results imply that NERP-4 acts only on SNAT2 to stimulate amino acid uptake. Glutamine and alanine are imported via several transporters in many types of cells⁵. In rat islets,

SNAT2, SNAT4, ASC-1 (SLC1A10), ASCT2 (SLC1A5), and B⁰AT1 (SLC6A19) transport glutamine and alanine⁵. *Snat2* knockdown reduced glutamine and alanine uptake by 13.6% and 30.3%, respectively, compared with siSCR treatment (Fig. 9e, f). MeAIB reduced glutamine and alanine uptake by 17.8% and 40.7%, respectively (Fig. 9g, h). These results suggest that transporters other than SNAT2 also mediate glutamine and alanine uptake in β cells.

Reference 5, Bröer S., Amino acid transporters as modulators of glucose homeostasis. *Trends Endocrinol. Metab.* **33**, 120–135 (2022).

Minor

1) Results. The first sentence of the second paragraph (Line 103) refers to results presented in the first paragraph and should be omitted.

Reply: We omitted the following sentence (Line 103 in the original manuscript) to avoid duplication.

NERP-4 induced calcium mobilisation into β cells, suggesting its potential role in GSIS.

2) The characteristics of MIN6-K8 cells should be briefly given, especially as far as the responsiveness of their insulin secretion to amino acids is concerned

Reply: The MIN6-K cell line is a pancreatic β -cell line established from mouse insulinoma (Reference 61 below). MIN6-K8 cells, which constitute a subclone of MIN6-K cell lines, have been used to study the mechanism of glutamine-amplified Ca^{2+} influx and insulin secretion (References 21–23 below). Here we studied MIN6-K8 cells to explore the interaction between amino acid uptake and β -cell function. We described this point as follows.

Lines 85–88 in Text

NERP-4-induced Ca^{2+} influx into β cells was investigated in MIN6-K8 cells, which are mouse insulinoma-derived β cells²¹ that have been used to study the mechanism of glutamine-amplified Ca^{2+} influx and insulin secretion^{22,23}.

Reference 21. Iwasaki M., et al. Establishment of new clonal pancreatic β -cell lines (MIN6-K) useful for study of incretin/cyclic adenosine monophosphate signaling. *J. Diabetes Investig.* **1**, 137–142 (2010).

Reference 22. Hashim, M., et al. Inhibition of SNAT5 induces incretin-responsive state from incretin-unresponsive state in pancreatic β -cells: Study of β -cell spheroid clusters as a model. *Diabetes* **67**,

1795–1806 (2018)

Reference 23. Han G., et al. Glutamate is an essential mediator in glutamine-amplified insulin secretion.

J. Diabetes Investig. **12**, 920–930 (2021).

3) “several orders of magnitude” should be cautionary “2-3 orders of magnitude” (line 100)

Reply: We revised the text to “2–3 orders of magnitude”. (line 103)

Reviewer 2

Major:

1) Figure 1 shows that NERP4 mediates rapid calcium influx. However, how is unclear. Critically, the relationship between NERP4-mediated calcium influx and NERP4-mediated activation of SNAT2 is uncertain. SNAT2 is an AA-Na²⁺ exchanger but I don't think that SNAT2 activation can elicits calcium influx directly. Conversely, the kinetic of calcium flux in figure 1 is compatible with cation channel activation mechanism (ion channel coupled to another GPCR? Ion channel receptor?), while it appears incompatible with an indirect mechanism as proposed in in fig 10i. The major problem here is that if calcium influx is independent from SNAT2, NERP4 might activate a second binding partner (a membrane receptor?), and thus the mechanism proposed here for insulin secretion remains incompletely characterized. This must be ruled out, e.g. by showing that calcium influx is abrogated by SNAT2 deletion? Otherwise it must be recognized that NERP4 mediated effects are not only mediated by SNAT2 but by other mechanisms as well (see #3).

We sincerely appreciate your careful reading of our manuscript and your valuable comments and advice.

Reply: We are sincerely sorry that the Fig. 1b legend in the original manuscript was insufficient. In the original manuscript, Fig. 1b showed NERP-4-evoked transient luminescence in a piece of

pancreatic tissue from apoaequorin transgenic mice. The arrows in Fig. 1b indicated the tested substances, each of which was administered before the time indicated by the corresponding arrow. We manually performed this assay; we removed the cuvette from the machine, applied the substance to the tissue, returned the cuvette to the machine, and then measured the fluorescence. This assay was developed as a screening test to explore peptides that increase intracellular Ca^{2+} in the tissues of apoaequorin transgenic mice. In this figure, we could not show the exact time from substance application to luminescence measurement. We added this explanation in the Fig. 1 legend in the revised manuscript.

We next investigated NERP-4-induced Ca^{2+} influx into pancreatic β cells using a Ca^{2+} indicator. In the revised manuscript, we substituted a widely used Ca^{2+} indicator, Fura-2, for the Ca^{2+} indicator Fluo-4 mentioned in the original manuscript. To examine Ca^{2+} influx under the same glucose conditions used in the GSIS experiment, we first exposed MIN6-K8 cells to HKRB containing 2.8 mM glucose, then to HKRB containing 16.7 mM glucose. In Fig. 1c of the revised manuscript, we present the Fura-2 ratio from 2.8 mM glucose to 16.7 mM glucose.

To demonstrate that NERP-4-induced Ca^{2+} influx is dependent on SNAT2, we studied Ca^{2+} influx in MIN6-K8 cells that had undergone *Snat2* knockdown or that were treated with MeAIB (a SNAT2 inhibitor). NERP-4-induced Ca^{2+} influx was abolished under both conditions (Fig. 9i, j). We also showed that *Snat2* knockdown and MeAIB abolished NERP-4-induced amino acid uptake (Fig. 9e–h) and GSIS (Fig. 9k, l). Based on these findings, we conclude that the effects of NERP-4 in β

cells are mediated via SNAT2. We described these results and added Fig. 9e–l and Extended Data Fig. 8d. We also modified Fig. 10i and its legend to explain the mechanism of NERP-4–induced Ca^{2+} influx.

Lines 219–225 in Text

To demonstrate that the effect of NERP-4 depends only on SNAT2, we performed SNAT2 deletion and inhibitor experiments. *Snat2* knockdown in MIN6-K8 cells abrogated NERP-4–induced [^{14}C]-L-glutamine and [^{14}C]-L-alanine uptake (Fig. 9e, f, Extended Data Fig. 8b). α -Methylaminoisobutyric acid (MeAIB), a non-metabolisable L-alanine analogue, is an orthosteric inhibitor of system A transporter³⁵. MeAIB also abrogated NERP-4–induced [^{14}C]-L-glutamine and [^{14}C]-L-alanine uptake (Fig. 9g, h). These results imply that NERP-4 acts only on SNAT2 to stimulate amino acid uptake.

Lines 238–242 in Text

Both *Snat2* knockdown and MeAIB abrogated NERP-4–induced Ca^{2+} influx (Fig. 9i, j, Extended Data Fig. 8d) and GSIS (Fig. 9k, l) in MIN6-K8 cells. MeAIB also abolished NERP-4–induced GSIS in human and C57BL/6J mouse islets (Fig. 9m, n). These findings reveal that NERP-4 acts specifically on SNAT2 to induce amino acid uptake, Ca^{2+} influx, and GSIS.

Lines 817–821 in Fig. 1 legend

b, Representative profile of the relative luminescence evoked in the apoaequorin transgenic mouse pancreas. Arrows (A–D) represent the tested substances. Medium (A), 1 μ M NERP-4 (B), 100 μ M ATP (C), and 2.5% Triton X-100 (D) were administered before the time indicated by the corresponding arrow. ATP was used as a positive control.

Lines 947–949 in Fig. 10 legend

NERP-4 binds to SNAT2 to stimulate glutamine and alanine uptake into β cells, thereby enhancing mitochondrial ATP production, glucose-induced Ca^{2+} mobilisation into β cells, and GSIS.

2) The identification of a protein-mediated mechanism for NER4 is very exciting because for only 1 other VGF-derived peptide, i.e. TLQP-21, a receptor-mediated mechanism has been identified (see <https://pubmed.ncbi.nlm.nih.gov/34626205/>). Thus, this project fills an important gap of knowledge in the field. However, the evidence that NERP4's sole binding partner is SNAT2 is not that strong in fig 8a; and its role as an allostatic activator is weak (see #4). Also, while pharmacological data convincingly demonstrate that SNAT2 activity is required for NERP4 biological effect, the evidence that NERP4-mediate amino acid uptake and insulin secretion is exerted uniquely via SNAT2 is not that strong (Fig 9f only).

Reply: To clarify the evidence that the effects of NERP4 are dependent on SNAT2, we added experiments involving SNAT2 deletion and pharmacological inhibition with MeAIB. Both *siSnat2* and MeAIB treatments in MIN6-K8 cells abolished NERP-4-induced glutamine and alanine uptake (Fig. 9e–h), Ca²⁺ influx (Fig. 9i, j), and insulin secretion (Fig. 9k, l). MeAIB also abolished NERP-4-induced insulin secretion in human and mouse islets (Fig. 9m, n). We think that NERP-4 acts specifically on SNAT2. We described these results as follows and added Fig. 9e–n and Extended Data Fig. 8d as mentioned in our response to the # 1 comment. We explained that NERP-4 exerts as a positive allosteric modulator to SNAT2 by adding new experiments described in # 4.

Lines 219–225 in Text

To demonstrate that the effect of NERP-4 depends only on SNAT2, we performed SNAT2 deletion and inhibitor experiments. *Snat2* knockdown in MIN6-K8 cells abrogated NERP-4–induced [¹⁴C]-L-glutamine and [¹⁴C]-L-alanine uptake (Fig. 9e, f, Extended Data Fig. 8b). α-Methylaminoisobutyric acid (MeAIB), a non-metabolisable L-alanine analogue, is an orthosteric inhibitor of system A transporter³⁵. MeAIB also abrogated NERP-4–induced [¹⁴C]-L-glutamine and [¹⁴C]-L-alanine uptake (Fig. 9g, h). These results imply that NERP-4 acts only on SNAT2 to stimulate amino acid uptake.

Lines 238–242 in Text

Both *Snat2* knockdown and MeAIB abrogated NERP-4–induced Ca²⁺ influx (Fig. 9i, j, Extended Data Fig. 8d) and GSIS (Fig. 9k, l) in MIN6-K8 cells. MeAIB also abolished NERP-4–induced GSIS in human and C57BL/6J mouse islets (Fig. 9m, n). These findings reveal that NERP-4 acts specifically on SNAT2 to induce amino acid uptake, Ca²⁺ influx, and GSIS.

3) Several C-terminal VGF-derived peptides (TLQP-62 in particular) are potent insulin secretagogues and mediate GSIS as well as they increase VGF mRNA expression in beta cells (<https://pubmed.ncbi.nlm.nih.gov/25917832/>). The mechanism and relevance of the autocrine effect of VGF peptides on VGF expression is still unclear. However, authors should quantify VGF mRNA (in addition to Ins and other genes) in all of their experiments (in vivo and in vitro). If VGF mRNA is indeed induced by NERP4 [and thus all VGF peptides are potentially produced and secreted] as well, some of its long-term effects (in vitro and in vivo) can be mediated by other VGF-derived peptides as well.

Reply: As the reviewer mentioned, VGF produces several insulinotropic peptides, including TLQP-21, TLQP-62, and NERP-2. TLQP-21 also enhances β -cell survival and function during the development of type 2 diabetes (Reference 43 below). We studied the long-term effects of NERP-4 on *Vgf* mRNA expression in *db/db* mouse islets, palmitate-treated mouse islets, and cytokines-treated MIN6-K8 cells. NERP-4 administration increased *Vgf* mRNA levels in these *in vivo* and *in vitro* experiments (Fig. 3e, Fig. 6j, and Extended Data Fig. 4b). We agree with the reviewer's comment that other VGF-derived peptides whose levels are increased by NERP-4 administration could partially mediate NERP-4's long-term effects on β -cell maintenance and survival. We described these results as follows and added Fig. 3e, Fig. 6j, and Extended Data Fig. 4b.

Lines 129–130 in Text

Palmitate increased the *Vgf* mRNA level and NERP-4 elevated it to an even greater extent (Fig. 3e).

Lines 162–163 in Text

A cytokine cocktail increased the *Vgf* mRNA level and NERP-4 further elevated it (Extended Data Fig. 4b).

Lines 177–179 in Text

NERP-4 also increased the following in *db/db* islets: *Ins1*, *Ins2*, *Iapp*, *Vgf*, and *Pdx-1* mRNA levels; insulin content; Ki67-positive cell number; and insulin staining intensity (Fig. 6g–o).

Lines 304–309 in Discussion

A previous study showed that VGF-derived C-terminal peptides, TLQP-21[VGF 556–576] and TLQP-62[VGF 556–617], are potent insulin secretagogues^{19, 42}. These two peptides also enhanced β -cell survival and function during the development of type 2 diabetes^{19, 42, 43}. NERP-4 increased *Vgf* mRNA in *db/db* mouse islets, palmitate- or cytokine-treated β cells, and naïve β cells. NERP-4-induced upregulation of VGF could mediate partly the long-term effects of NERP-4 on β -cell maintenance and survival.

Reference 43. Hannedouche, S., et al. Identification of the C3a receptor (C3AR1) as the target of the VGF-derived peptide TLQP-21 in rodent cells. *J. Biol. Chem.* **288**, 27434–27443 (2013).

4) Much emphasis is given in the discussion to NERP4 being a PAM. However, this conclusion is not supported by rigorous data, but only inferred from indirect evidence (Fig 9k). At the minimum an in-silico model of the mode of binding on SNAT2 (+/- MeAIB)-NERP4 should be presented if authors want to sustain this conclusion.

Reply: To respond to this comment, we added a kinetics experiment to demonstrate that NERP-4 acts as a PAM. We analysed concentration-dependent uptake of [¹⁴C]-MeAIB in the range of 1 μM to 2 mM in the presence or absence of NERP-4. [¹⁴C]-MeAIB uptake showed a good fit to a Michaelis–Menten curve, with a Km of 881.1 ± 220.7 μM and a Vmax of 2,240 ± 242.3 pmol/mg protein/min (Fig. 9o). NERP-4 administration lowered the Km of [¹⁴C]-MeAIB (412.6 ± 113.5 μM) without changing the Vmax, implying that NERP-4 modulates the binding affinity of SNAT2 to amino acids. As we showed in the original manuscript, MeAIB, an orthosteric inhibitor of SNAT2, did not change [¹²⁵I]-Y-NERP-4[8–19] binding to MIN6-K8 cells (Extended Data Fig. 8f). This supports the finding that the NERP-4 binding site is different from the substrate binding site. Combined with the kinetics analysis and binding study, NERP-4 acts as a positive allosteric modulator (PAM).

Based on protein prediction approaches and homology modelling, SNAT2 is expected to have an intracellular N terminus, 11 putative transmembrane helices, and an extracellular C terminus (Reference 70 below). We searched the AlphaFold Protein Structure Database entry for SNAT2, but the topological structure and amino acid transport site of SNAT2 have yet to be determined. Future structural analyses of the molecular features of the SNAT2-binding site of NERP-4 could help establish an *in silico* model of the mode of binding to SNAT2. We described these results and added Fig. 9o.

Lines 255–265 in Text

NERP-4 augmented [^{14}C]-L-alanine uptake 1 min after its administration to MIN6-K8 cells (Extended Data Fig. 8e). To study the kinetic properties of SNAT2 activation by NERP-4, we determined the concentration-dependent uptake of [^{14}C]-MeAIB in the range of 1 μM to 2 mM at 2 min in the presence or absence of NERP-4. [^{14}C]-MeAIB uptake showed a good fit to a Michaelis–Menten curve, with a K_m of $881.1 \pm 220.7 \mu\text{M}$ and a V_{max} of $2,240 \pm 242.3 \text{ pmol/mg protein/min}$ (Fig. 9o). NERP-4 lowered the K_m of [^{14}C]-MeAIB ($412.6 \pm 113.5 \mu\text{M}$) without changing the V_{max} . Furthermore, MeAIB did not alter [^{125}I]-Y-NERP-4[8–19] binding to MIN6-K8 cells (Extended Data Fig. 8f), supporting the finding that the NERP-4 binding site is different from the substrate binding site. Taken together, these results show that NERP-4 is considered to modulate the binding affinity of SNAT2 to amino acids, implying that NERP-4 acts as a positive allosteric modulator (PAM).

Reference 70. Schiöth H. B., Roshanbin S., Hägglund M.G. & Fredriksson R. Evolutionary origin of amino acid transporter families SLC32, SLC36 and SLC38 and physiological, pathological and therapeutic aspects. *Mol. Aspects Med.* **34**, 571–585 (2013).

Specific comments:

5) The result section is difficult to read. Authors are encouraged to use a more narrative style of presenting the results.

Reply: We are grateful for this comment. We added many sentences to clarify the results.

6) The section on ER stress and mito dysfunction is not well connected to the rest of the story. I'm not sure it is really required to support the main conclusion and mechanism.

Reply: We think that NERP-4 effects occur during two different phases. One is the rapid phase, in which NERP-4 stimulates Ca^{2+} influx and insulin secretion. The other is a slower phase characterized by NERP-4-induced upregulation of genes involved in β -cell maintenance and recovery from metabolic stress. In the revised manuscript, we studied the effects of NERP-4 on the expression of proteins (CHOP, SOD2, Nrf2, and cleaved caspase-3) involved in apoptosis, oxidative stress, and ER stress (Fig. 3f, g, Fig. 4h, and Fig. 7f). In the original manuscript, we also demonstrated that *Snat2* knockdown abolished the abilities of NERP-4 to increase the OCR, the ATP production, and cell viability and to decrease reactive oxidative stress production (Fig. 10e–h). We added these findings to clarify NERP-4 effects on β -cell maintenance and survival. We added Fig. 3f, g, Fig. 4h, and Fig. 7f.

Lines 131–140 in Text

Palmitate reduced the protein and mRNA levels of the antioxidative response marker SOD2, while NERP-4 reversed these alterations (Fig. 3f, Extended Data Fig. 3). Nrf2 responds to oxidative stress by translocating from the cytosol to the nucleus and then inducing antioxidant genes²⁷. Palmitate reduced Nrf2 translocation to the nucleus and *Nrf2* mRNA level (Fig. 3g, Extended Data Fig. 3). NERP-4 reversed these alterations (Fig. 3g, Extended Data Fig. 3). Under non-stressed conditions,

NERP-4 administration to isolated C57BL/6J mouse islets increased the mRNA levels of *Vgf*, *Sod2*, and *Nrf2* (Fig. 3e, Extended Data Fig. 3). NERP-4 also decreased the protein and mRNA levels of CHOP (Fig. 3f, Extended Data Fig. 3), increased the SOD2 protein level (Fig. 3f), and enhanced Nrf2 nuclear translocation (Fig. 3g).

Lines 157–164 in Text

We also investigated the roles of NERP-4 in a model of cytokine-induced β -cell impairment. Interleukin-1 β was previously shown to induce oxidative and ER stress in β cells³⁰. A cytokine cocktail (tumour necrosis factor- α , interleukin-1 β , and interferon- γ) upregulated the apoptotic marker protein cleaved caspase-3 and its mRNA, reduced GSIS and cell viability, and induced cell death in MIN6-K8 cells (Fig. 4h–k, Extended Data Fig. 4b). NERP-4 reversed these alterations. A cytokine cocktail increased the *Vgf* mRNA level and NERP-4 further elevated it (Extended Data Fig. 4b). Collectively, these data indicate that NERP-4 suppresses oxidative and ER stress, thereby improving β -cell mitochondrial function.

7) Page 9. The “criteria for specific binding” is not explained in the manuscript (or I cannot find it).

As I mentioned above the specificity/selectivity of NERP4-SNAT2 binding is not very convincing in my opinion.

Reply: We have now presented the "criteria for specific binding" to identify candidate targets for NERP-4 in the LRC-TriCEPS experiment. The criteria for a candidate that interacts with the ligand of interest are an adjusted *P* value < 0.01 and fold change > 3.5, based on the manufacturer's recommendation (Reference 33 below). The fold change of SNAT2 (S38A2) was the highest (fold change, 4.0), followed by that of SCRB2 (fold change, 3.7) (Fig. 8a). Only SNAT2 and SCRB2 met the aforementioned criteria.

We verified the specificity and selectivity of NERP4 binding to SNAT2 in knockdown and pharmacological inhibition experiments (Fig. 8c–f and Fig. 9e–n). NERP-4 bound only to SNAT2-overexpressing HEK293 cells, and not to HEK293 cells expressing other SNATs (Fig. 8f). Knockdown and pharmacological inhibition of SNAT2 abolished NERP-4's effects on glutamine and alanine uptake (Fig. 9e–h), Ca²⁺ influx (Fig. 9i, j), and GSIS (Fig. 9k–n). We described these results in the text. We also mentioned the "criteria for specific binding" of TriCEPS in the Methods.

Lines 220–222 in Text

Snat2 knockdown in MIN6-K8 cells abrogated NERP-4–induced [¹⁴C]-L-glutamine and [¹⁴C]-L-alanine uptake in MIN6-K8 cells (Fig. 9e, f, Extended Data Fig. 8b).

Lines 238–242 in Text

Both *Snat2* knockdown and MeAIB abrogated NERP-4–induced Ca²⁺ influx (Fig. 9i, j, Extended Data Fig. 8d) and GSIS (Fig. 9k, l) in MIN6-K8 cells. MeAIB also abolished NERP-4–induced GSIS in human and C57BL/6J mouse islets (Fig. 9m, n). These findings reveal that NERP-4 acts specifically on SNAT2 to induce amino acid uptake, Ca²⁺ influx, and GSIS.

Lines 732–737 in Methods

The results are presented as a volcano plot, with the *X*-axis representing the mean ratio fold change (on a log₂ scale) and the *Y*-axis representing the statistical significance *P* value of the ratio fold change for each protein (on a $-\log_{10}$ scale). The criteria for considering a protein as a candidate for interacting with the ligand of interest were an adjusted *P* value < 0.01 and a fold change > 3.5 for NERP-4 compared to NERP-2, according to the manufacturer's recommendation³³.

Reference 33. Frei, A. P., et al. Direct identification of ligand-receptor interactions on living cells and tissues. *Nat. Biotechnol.* **30**, 997–1001 (2012).

8) The statistical section describes the use of Bonferroni post hoc while the figure legends describe the Tukey. Please clarify.

Reply: As the reviewer pointed out, our original description in the statistical section was insufficient.

We used the following: two-way ANOVA followed by Bonferroni's post-test for multiple comparisons; one-way ANOVA with Tukey's multiple comparisons test; one-way ANOVA with Fisher's LSD test; and the unpaired two-tailed Student's *t*-test. We corrected the statistics section accordingly.

Lines 805–812 in Methods

Statistical analysis

Statistical analyses were performed with GraphPad Prism 7 statistical software (GraphPad) using two-way ANOVA followed by Bonferroni's post-test for multiple comparisons, One-way ANOVA and Tukey's multiple comparisons test, one-way ANOVA and Fisher's LSD test, and the unpaired two-tailed Student's *t*-test. Outliers were identified by ROUT method with Prism in all experiments. The statistical results were the same in all the experiments when outliers were included or excluded. We showed the figures that include outliers. All data are expressed as means \pm s.e.m. $P < 0.05$ was considered statistically significant.

9) Fig 1k seems irrelevant, it is quite expected that beta cells express insulin to a level higher than VGF.

Reply: Following the reviewer's comment, we moved Fig. 1k to Extended Data Fig. 1b.

Reviewer 3

1. While the manuscript describes potentially mechanistic studies, the calcium studies shown early in the paper (Fig 1) are insufficiently described and their importance is not clear for the secretory phenotype. For example, in Fig 1b it is shown that the addition of NERP-4 raises free calcium in 'pancreatic extracts', leading to a sharply rising spike that declines over 100 seconds. It is not clear what the preparation is here. Do they mean 'islets' or cells from the pancreas of some kind? Plus, how was a dose of 1 μ M chosen here while the dose used in the center panel (on MIN6-K8, what is K8, a subclone?) was only 0.1 nM? The MIN6-K8 run was carried out in 22 mM glucose, which is well above the physiological range and yet no transients or oscillations in MIN6 calcium were seen during the first measurement period, which is odd. The authors also present no evidence that the rise in Ca to NERP-4 is due to intracellular Ca release and not influx—or both? Properly, the experiment should be run in Ca free media to ensure the rise in Ca is due to release. As the authors no doubt know, Ca is not the only regulator of glucose induced insulin secretion and it is possible that NERP-4 affects not the triggering but the amplification pathway.

We sincerely appreciate your careful reading of our manuscript and your valuable comments and advice.

1) Reply to “the different doses of NERP-4 used between apoaequorin transgenic mice and MIN6-K8 cells”

The luminescence assay with apoaequorin transgenic mice has been developed as a screening test to explore peptides which increase intracellular Ca^{2+} . We corrected the unclear description of apoaequorin transgenic mice. We used a piece of pancreatic tissue from apoaequorin transgenic mice to study the luminescence-evoking activity of NERP-4 (Fig. 1b). We revised “pancreatic extracts” (Line 89 in the original draft) to “a piece of pancreatic tissue” (Line 86 in the revised manuscript). For this screening test, we stimulated the piece of pancreatic tissue with a high concentration of peptides. The MIN6-K cell line is a mouse insulinoma-derived β -cell line (Reference 21 below). MIN6-K8 cells, a subclone of MIN6-K cell lines, have been used to study the mechanism of glutamine-amplified Ca^{2+} influx and insulin secretion (References 21–23 below). A concentration of 0.1 nM NERP-4 was sufficient to study NERP-4 activity in MIN6-K8 cells; however, 1 μM NERP-4 was needed to evoke luminescence in the pancreatic tissue from apoaequorin transgenic mice. We also showed that NERP-4 did not enhance the glucose-induced elevation of intracellular cAMP, a key second messenger of the amplification pathway (Fig. 2f). We described these points as follows.

Lines 553–588 in Methods

Apoaequorin transgenic mice

We generated apoaequorin transgenic mice to explore transient luminescence caused by intracellular Ca^{2+} mobilisation as described elsewhere¹⁰. Mouse pancreatic tissues were minced in RPMI-1640 (Invitrogen) to prepare small pieces. In order to reconstitute aequorin, 1–2 mm³ pieces were incubated with coelenterazine (Molecular Probes) for 3 h. Pancreatic pieces were sequentially treated with 1 μM NERP-4, 100 μM ATP, and 2.5% Triton X-100.

Lines 85–88 in Text

NERP-4-induced Ca^{2+} influx into β cells was investigated in MIN6-K8 cells, which are mouse insulinoma-derived β cells²¹ that have been used to study the mechanism of glutamine-amplified Ca^{2+} influx and insulin secretion^{22,23}.

Lines 111–113 in Text

In MIN6-K8 cells, NERP-4 enhanced the glucose-induced elevation of ATP concentration, but not that of intracellular cAMP concentration (Fig. 2e, f).

2) Reply to “the rise in Ca²⁺ to NERP-4 due to Ca²⁺ influx”

In the original manuscript, we studied Ca²⁺ influx using Fluo4-AM as a Ca²⁺ indicator. In the revised manuscript, we substituted a widely used Ca²⁺ indicator, Fura-2 for Fluo-4. To examine Ca²⁺ influx under the same glucose condition as used in the GSIS experiment, we first exposed MIN6-K8 cells to HKRB containing 2.8 mM glucose, then HKRB containing 16.7 mM glucose. A high glucose concentration (16.7 mM) caused a transient Ca²⁺ influx into MIN6-K8 cells, and NERP-4 enhanced Ca²⁺ influx (Fig. 1c). To determine if the rise in Ca²⁺ was due to Ca²⁺ influx via Ca²⁺ channels, we examined Ca²⁺ influx into MIN6-K8 cells treated with EGTA (a calcium chelator) or nifedipine (an L-type Ca²⁺ channel blocker). Both treatments suppressed NERP-4–induced Ca²⁺ influx (Fig. 1d, e). These results indicate that NERP-4 increased Ca²⁺ influx via Ca²⁺ channels expressed on the cell membrane. We described these results as follows and added Fig. 1c–e.

Lines 89–93 in Text

To explore whether the rise in Ca²⁺ was due to extracellular Ca²⁺ influx via Ca²⁺ channels, we studied Ca²⁺ influx into MIN6-K8 cells treated with EGTA (a calcium chelator) or nifedipine (an L-type Ca²⁺ channel blocker). Both treatments suppressed NERP-4–induced Ca²⁺ influx (Fig. 1d, e), implying that NERP-4 mediated Ca²⁺ influx via Ca²⁺ channels.

Reference 21. Iwasaki M., et al. Establishment of new clonal pancreatic β -cell lines (MIN6-K) useful

for study of incretin/cyclic adenosine monophosphate signaling. *J. Diabetes Investig.* **1**, 137–142 (2010).

Reference 22. Hashim, M., et al. Inhibition of SNAT5 induces incretin-responsive state from incretin-unresponsive state in pancreatic β -cells: Study of β -cell spheroid clusters as a model. *Diabetes* **67**, 1795–1806 (2018)

Reference 23. Han G., et al. Glutamate is an essential mediator in glutamine-amplified insulin secretion. *J. Diabetes Investig.* **12**, 920–930 (2021).

2. Another problem with the data in Fig. 1 is that even if true, it is not clear what the functional role of the Ca rise is for the data shown in the rest of the paper? Thus, is the Ca rise the result of binding of NERP-4 to the amino acid transporter and amino acid flux or is it unrelated and due to another pathway? What does preventing the Ca rise do to the subsequent measurements of NERP-4 action? It is unclear.

Reply: In response to the reviewer's comment, we added the results of new experiments in the revised manuscript. To examine the relationship between the amino acid transporter and Ca²⁺ rise, we studied NERP-4-induced Ca²⁺ influx in MIN6-K8 cells treated with *Snat2* knockdown or the SNAT2 inhibitor MeAIB. These treatments abolished NERP-4-induced amino acid influx (Fig. 9e–h) and Ca²⁺ influx (Fig. 9i, j), suggesting that Ca²⁺ rise was mediated by the binding of NERP-4 to SNAT2. Both SNAT2 deletion and pharmacological inhibition abolished NERP-4-induced GSIS in human and mouse islets and MIN6-K8 cells (Fig. 9k–n). To determine the relationship of NERP-4 action to Ca²⁺ rise in β cells, we studied insulin secretion with EGTA or nifedipine. NERP-4 did not increase GSIS in MIN6-K8 cells treated with these chemicals (Fig. 2g, h). We conclude that NERP-4 binds to SNAT2, thereby enhancing amino acid uptake and Ca²⁺ influx into β cells. We described these results and added new Fig. 2g, h, Fig. 9e–j, and Fig. 9l.

Lines 113–114 in Text

Both EGTA and nifedipine treatments abolished NERP-4–induced GSIS in MIN6-K8 cells (Fig. 2g, h).

Lines 219–225 in Text

To demonstrate that the effect of NERP-4 depends only on SNAT2, we performed SNAT2 deletion and inhibitor experiments. *Snat2* knockdown in MIN6-K8 cells abrogated NERP-4–induced [¹⁴C]-L-glutamine and [¹⁴C]-L-alanine uptake (Fig. 9e, f, Extended Data Fig. 8b). α -Methylaminoisobutyric acid (MeAIB), a non-metabolisable L-alanine analogue, is an orthosteric inhibitor of system A transporter³⁵. MeAIB also abrogated NERP-4–induced [¹⁴C]-L-glutamine and [¹⁴C]-L-alanine uptake (Fig. 9g, h). These results imply that NERP-4 acts only on SNAT2 to stimulate amino acid uptake.

Lines 238–242 in Text

Both *Snat2* knockdown and MeAIB abrogated NERP-4–induced Ca²⁺ influx (Fig. 9i, j, Extended Data Fig. 8d) and GSIS (Fig. 9k, l) in MIN6-K8 cells. MeAIB also abolished NERP-4–induced GSIS in human and C57BL/6J mouse islets (Fig. 9m, n). These findings reveal that NERP-4 acts specifically on SNAT2 to induce amino acid uptake, Ca²⁺ influx, and GSIS.

3. Throughout the paper, plots are shown that appear to describe underpowered studies and what is more, many outliers can be seen, usually above the mean values shown. The analyses are thus not

credible without assessing the impact of these outliers to determine whether they are true outliers and their removal does not result in a loss of significance.

Reply: As the reviewer indicated, we assessed the impact of the outliers. We used the ROUT method with Prism to identify outliers in all experiments. We performed statistical tests that excluded these outliers and found the same significant results as in the original tests that had included the outliers. For this reason, we still show the figures that include the outliers. We explain these points as follows.

Lines 809–811 in Methods

Outliers were identified by ROUT method with Prism in all experiments. The statistical results were the same in all the experiments when outliers were included or excluded. We showed the figures that include outliers.

4. There seems to be a plethora of possible effects that are not considered. For the whole animal studies shown in Fig. 6 for example, does NERP-4 treatment affect food intake? Glucagon levels, body weight?

Reply: In the original manuscript, we showed that NERP-4 did not change the glucagon mRNA level or the area of glucagon immunoreactivity (Extended Data Fig. 5d, f in the revised version). In the

revised manuscript, we studied food intake, body weight, and plasma glucagon levels in *db/db* mice administered NERP-4. NERP-4 did not affect these parameters (Extended Data Fig. 5a, b, e). We described these results as follows and added Extended Data Fig. 5a, b, e.

Lines 170–172 in Text

Two-week NERP-4 administration to *db/db* mice decreased fasting blood glucose and plasma insulin levels compared with saline treatment, but did not change body weight or food intake (Fig. 6a, b, Extended Data Fig. 5a, b).

Lines 179–180 in Text

NERP-4 did not change the *Gcg* mRNA level, plasma glucagon level, or the area of glucagon immunoreactivity (Extended Data Fig. 5d–f).

5. There does not seem to be a clear explanation for the relationship between doses of NERP-4 used in binding studies to the various functional studies; this should be discussed in detail.

Reply: We used [¹²⁵I]-tyrosine-NERP-4[8–19] in the binding study and full-size NERP-4 in the functional studies. As the reviewer pointed out, we used different doses of these two peptides in the binding studies and functional studies. Tyrosine-NERP-4[8–19] is a C-terminally tyrosyl dodecapeptide of NERP-4 corresponding to amino acid positions 8 to 19. We used NERP-4[8–19] as an antigen to produce an antiserum for NERP-4. We thus synthesized tyrosine-NERP-4[8–19] for the radioiodination experiments, which included the binding experiment. The binding of tyrosine-NERP-4[8–19] was weaker than that of full-size NERP-4. We therefore tested 0.5 nM–5 nM [¹²⁵I]-tyrosine-NERP-4[8–19] in the binding experiment. In the various functional studies, we used 0.1 nM full-size NERP-4. To clarify the difference between the doses of tyrosine-NERP-4[8–19] and NERP-4, we mentioned that tyrosine-NERP-4[8–19] was used in the binding experiments.

Lines 200–206 in Text

We studied the binding of NERP-4 to SNAT2 with a radioisotope-labelled C-terminally tyrosyl fragment of NERP-4, [¹²⁵I]-Y-NERP-4[8–19], which was used in the RIA of NERP-4. [¹²⁵I]-Y-NERP-4[8–19] bound to SNAT2-overexpressing HEK293 cells in a concentration-dependent manner (Fig. 8c, Extended Data Fig. 7). [¹²⁵I]-Y-NERP-4[8–19] binding was reduced by the addition of excessive

NERP-4 (Fig. 8c), suggesting that the binding was specific for NERP-4. [¹²⁵I]-Y-NERP-4[8–19] binding was detected in the membrane fraction of SNAT2-overexpressing HEK293 cells (Fig. 8d).

Lines 745–754 in Methods

Binding assay

SNAT2-expressing HEK293 cells were incubated in HEPES buffer (DMEM containing 0.1% bovine serum albumin, 0.1% NaN₃ and 50 mM HEPES (pH 7.4)) containing 0.5–16 nM [¹²⁵I]-Y-NERP-4[8–19] with or without unlabelled NERP-4 for 1 h at 37 °C. The radioactivity of cell lysates was measured with an automatic γ -counter (AccuFLEX γ ARC-7001, Hitachi). Nonspecific binding was determined in the presence of a 500-fold excess of unlabelled NERP-4. Cell membrane proteins (4 μ g) were extracted from SNAT2-expressing HEK293 cells, as described previously⁶⁶, and incubated with 2 nM [¹²⁵I]-Y-NERP-4[8–19] for 3 h at 25 °C. Bound [¹²⁵I]-Y-NERP-4[8–19] was isolated by vacuum filtration, and the radioactivity was measured. SNAT3-, SNAT4-, or SNAT5-expressing HEK293 cells were also studied in the above binding assay.

6. With regards to the hypothesis put forward, could the need for NERP-4 be mitigated by increasing the exposure to amino acid concentrations tested?

Reply: Given that NERP-4 increased GSIS by stimulating amino acid uptake, we assumed that exposure of β cells to higher amino acid concentrations could mitigate the effect of NERP-4 on GSIS. We compared GSIS from MIN6-K8 cells under different glutamine concentrations (1 μ M to 1 mM). Concentrations equal to or higher than 1 μ M increased GSIS (Extended Data Fig. 8c). NERP-4 augmented GSIS in the presence of 1 μ M glutamine, but NERP-4-induced GSIS was mitigated by 1 mM glutamine (Extended Data Fig. 8c). We added these results in the text and Extended Data Fig. 8c.

Lines 232–238 in Text

Given that NERP-4 increased GSIS by stimulating amino acid uptake, we assumed that β -cell exposure to higher glutamine concentrations could mitigate the effects of NERP-4 on GSIS. We studied the insulinotropic activity of NERP-4 in MIN6-K8 cells at glutamine concentrations from 1 μ M to 1 mM in amino acid-free HEPES Krebs–Ringer bicarbonate (HKRB) buffer. Glutamine concentrations equal to or higher than 1 μ M increased GSIS in a concentration-dependent manner (Extended Data Fig. 8c). NERP-4 augmented GSIS at glutamine concentrations between 1 μ M and 100 μ M, but not at 1 mM glutamine (Extended Data Fig. 8c).

7. Minor: the authors should have the paper edited by a native English speaker to improve the grammar.

I was struck in many places where the sentences used I think conveyed the exact opposite meaning than intended, unfortunately. For instance, they say "...restored these alterations" when I think the authors meant "reversed or prevented" which adds confusion.

Reply: According to the reviewer's comments, we replaced "restored" with "reversed."

We had the manuscript edited by a native English speaker and attached the certification.

REVIEWER COMMENTS

Reviewer #1 (Remarks to the Author):

Authors have done a great work in addressing my remarks, adding new, and very interesting, data, in particular on transport studies.

While most of my concerns have been successfully overcome, I still have some doubts on the mechanistic aspects (see my comments 8-9-10). In summary, I think that the results now clearly indicate a major (and possibly exclusive) role of SNAT2 in the effect but I wonder if it is due to the metabolic effect of the transported substrates (I doubt that this is likely, see my comments 8-9-10, and lack of data on possible changes in the intracellular content of amino acids) or to some transduction pathway activated by the transporter (as suggested in other contexts, see the concept of transceptors proposed by Hundal's group several years ago).

I'm going through my original remarks and authors' rebuttal.

Remarks 1)-2)-3)-4)-5)-6)

Comment: Ok.

7) SNAT2 is claimed to be operative in glutamine and alanine (+ cysteine) transport. Yet, the specificity of the transporter is much larger. For instance, a preferential natural substrate of the transporter is Pro, which poorly interacts with most of the other transporters for neutral amino acids, thus constituting a reliable indicator of SNAT2 activity. Have the authors any data about changes in Pro transport induced by NERP-4?

Reply: As the reviewer mentioned, proline is the preferential natural substrate of SNAT2. We studied the effect of NERP-4 on proline uptake in MIN6-K8 cells. NERP-4 increased proline uptake (Extended Data Fig. 8a), supporting the finding that NERP-4 acts on SNAT2 as a target.

Comment: Interesting novel results. Could they point to a physiological role for Pro?

8) Have the authors any evidence that NERP-4 effectively change the intracellular concentration of amino acids? Have they measured the cell content of Gln (or Ala) in NERP-

4-treated models?

Reply: We administered 1 μ M [14C]-L-glutamine, [14C]-L-alanine, or [3H]-L-proline to MIN6-K8 cells in the presence or absence of NERP-4. We determined their intracellular contents 2 min after these administrations. NERP-4 increased the contents of glutamine, alanine, and proline (Fig. 9c, d, Extended Data Fig. 8a). We added the results in the text and in Fig. 9c, d, and Extended Data Fig. 8a.

Lines 213–217 in Text

We next investigated the effect of NERP-4 on amino acid transport activity. Human and C57BL/6J mouse islets and MIN6-K8 cells were administered NERP-4 with 1 μ M [14C]-L-glutamine. NERP-4 increased the intracellular contents of [14C]-L-glutamine in both islets and MIN6-K8 cells (Fig. 9a–c).

Next, MIN6-K8 cells were administered NERP-4 with 1 μ M [14C]-L-alanine or 1 μ M [3H]-L-proline. NERP-4 also increased their intracellular contents (Fig. 9d, Extended Data Fig. 8a).

Comment: I must consider authors' reply unsatisfactory. What they show is not the intracellular concentration or content of the amino acids but, rather, the 2-, 5-, or 10-min uptake of the labelled substrates. Since they measure in another part of the study the extracellular levels of amino acids (see "Amino acid determination" in Methods), it should be easily feasible to determine if NERP-4 effectively changes the cell content of amino acids. Alternatively, I suggest that they should replace "intracellular contents" with "2-, 5- or 10-min uptake", as specified in the experiment. However, from a mechanistic point of view, it would be important (see below, comments 9 and 10) to know if NERP-4 increase the intracellular amino acid content and if the increase is restricted or not to SNAT2 substrates.

9) The results reported in Fig. 9e are quite expected, and cannot be taken per se as an evidence for a process coupled with insulin secretion. Obviously, when incubated in amino acid-free saline, amino acids go out of cells down their transmembrane gradient. Have the authors any evidence that the amino acid concentrations they measure outside are larger in conditions of insulin secretion stimulated (e.g. high glucose) compared to control (low glucose)? Moreover, an interesting functional proof of their model would be to demonstrate that this efflux is stimulated by NERP-4. Have the authors checked other amino acids? In other words, the three amino acid reported are the only released?

Reply: We quantified amino acids in the media after GSIS experiments under low- (2.8 mM) or high-(16.7 mM) glucose conditions. We detected glutamine, alanine, proline, glutamate, leucine, arginine and other amino acids at similar concentrations in both low- and high-glucose media (Extended Data Table 1a). Glutamine, glutamate, alanine, leucine, and arginine were previously shown to stimulate insulin secretion (Reference 2 below). NERP-4 did not augment insulin secretion from isolated C57BL/6J mouse islets under 2.8 mM glucose, even when amino acids were present in the medium (Fig. 2b, c). Another research group demonstrated that 2 mM glutamine administration to MIN6-K8 cells under 2.8 mM glucose did not increase intracellular Ca²⁺ and had no effect on GSIS (Reference 23 below). Combined together, these findings indicate that under high-glucose conditions, NERP-4 functions as an insulinotropic peptide by enhancing the uptake of amino acids released from β cells.

NERP-4 does not act on SNAT3 or SNAT5 which stimulates the efflux of glutamine from β cells. We thus consider that glutamine efflux is independent of NERP-4. We added these results in the text and in Extended Data Table 1.

Lines 243–254 in Text

We used amino acid-free HKRB buffer in GSIS experiments. SNAT3, SNAT5, and ASCT2 function as exporters of glutamine under conditions in which extracellular glutamine is absent or very low. To explore the source of amino acids that stimulate NERP-4-induced GSIS, we quantified amino acids released into the medium after GSIS experiments. MIN6-K8 cells released glutamine, alanine, proline, and other amino acids under both 2.8 mM and 16.7 mM glucose conditions (Extended Data Table 1). The concentrations of these amino acids were comparable between low and high glucose conditions. NERP-4 did not augment Ca²⁺ influx or GSIS under 2.8 mM glucose even when amino acids were present in the medium (Fig. 2b, c, Extended Data Fig. 1a). Another research group demonstrated that 2 mM glutamine administration to MIN6-K8 cells in medium containing 2.8 mM glucose did not affect GSIS or Ca²⁺ influx²³. Together, these findings indicate that under high glucose conditions, NERP-4 functions as an insulinotropic peptide by enhancing uptake of amino acids released from β cells.

Reference 2. Newsholme, P., Bender, K., Kiely, A. & Brennan, L. Amino acid metabolism, insulin secretion and diabetes. *Biochem. Soc. Trans.* 35, 1180–1186 (2007).

Reference 23. Han G., et al. Glutamate is an essential mediator in glutamine-amplified

insulin secretion. *J. Diabetes Investig.* 12, 920–930 (2021).

Comment: The experiment performed confirms that the concentration of the amino acids released by the cells is in the micromolar range (see below, comment 10).

10) More importantly, authors state that NERP-4 stimulates GSIS even under amino acid-free conditions (line 250-251). Thus, how can SNAT2-mediated Gln transport works under those conditions? It is very unlikely that the extremely low quantities of Gln released from cells (in the order of microM) can justify a significant influx mediated by SNAT2 (whose K_m for Gln is > 1 mM).

Reply: To respond to this important comment, we added the following experiments. First, we verified that amino acids released from β cells are imported through the activity of the NERP-4–SNAT2 axis. We administered $1 \mu\text{M}$ [^{14}C]-L-glutamine, [^{14}C]-L-alanine, or [^3H]-L-proline to MIN6-K8 cells in the presence or absence of NERP-4, and 2 min later determined their intracellular contents. NERP-4 increased the intracellular contents of these amino acids (Fig. 9c, d, Extended Data Fig. 8a). These findings imply that low quantities of amino acids (i.e., $1 \mu\text{M}$ concentration) are imported into β cells

by NERP-4. Next, we verified that NERP-4–induced amino acid uptake mediated Ca^{2+} influx and GSIS under amino acid–free conditions. Snat2 knockdown abolished NERP-4–induced glutamine and alanine uptake (Fig. 9e, f), Ca^{2+} influx (Fig. 9i), and GSIS (Fig. 9k). MeAIB also abrogated NERP-4–induced glutamine and alanine uptake (Fig. 9g, h), Ca^{2+} influx (Fig. 9j), and GSIS (Fig. 9l). We conclude that NERP-4 acts on SNAT2 to cause amino acid uptake that induces Ca^{2+} influx and GSIS. We further analysed the kinetic properties of concentration-dependent uptake of [^{14}C]-MeAIB in the range of $1 \mu\text{M}$ to 2 mM in the presence or absence of NERP-4. [^{14}C]-MeAIB uptake showed a good fit to a Michaelis–Menten curve, with a K_m of $881.1 \pm 220.7 \mu\text{M}$ and a V_{max} of $2,240 \pm 242.3$

$\text{pmol/mg protein/min}$ (Fig. 9o). NERP-4 administration lowered the K_m of [^{14}C]-MeAIB ($412.6 \pm 113.5 \mu\text{M}$) without changing the V_{max} , implying that NERP-4 modulates the binding affinity of SNAT2 to amino acids. As we showed in the original manuscript, MeAIB, an orthosteric inhibitor of SNAT2, did not change [^{125}I]-Y-NERP-4[8–19] binding to MIN6-K8 cells (Extended Data Fig. 8f). This supports the finding that the NERP-4 binding site is

different from the substrate binding site. Combined with the kinetics analysis and binding study, NERP-4 acts as a positive allosteric modulator (PAM). Lastly, we confirmed that NERP-4 augmented GSIS in the presence of μM -order concentrations of glutamine in MIN6-K8 cells. One micromolar glutamine increased GSIS, and coadministration of NERP-4 and 1 μM glutamine further increased GSIS (Extended Data Fig. 8c). We thus think that the impact of NERP-4 on SNAT2 is significant under low quantities of glutamine.

We added these results in Extended Data Table 1, Extended Data Fig. 8a, Fig. 9e, f, i–k, Lines 220–222 in Text

Snat2 knockdown in MIN6-K8 cells abrogated NERP-4–induced [^{14}C]-L-glutamine and [^{14}C]-L-alanine uptake in MIN6-K8 cells (Fig. 9e, f, Extended Data Fig. 8b).

Lines 236–242 in Text

Glutamine concentrations equal to or higher than 1 μM increased GSIS in a concentration-dependent manner (Extended Data Fig. 8c). NERP-4 augmented GSIS at glutamine concentrations between 1 μM and 100 μM , but not at 1 mM glutamine (Extended Data Fig. 8c). Both Snat2 knockdown and MeAIB abrogated NERP-4–induced Ca^{2+} influx (Fig. 9i, j, Extended Data Fig. 8d) and GSIS (Fig. 9k, l) in MIN6-K8 cells. MeAIB also abolished NERP-4–induced GSIS in human and C57BL/6J mouse islets (Fig. 9m, n). These findings reveal that NERP-4 acts specifically on SNAT2 to induce amino acid uptake, Ca^{2+} influx, and GSIS.

Lines 245–249 in Text

To explore the source of amino acids that stimulate NERP-4–induced GSIS, we quantified amino acids released into the medium after GSIS experiments. MIN6-K8 cells released glutamine, alanine, proline, and other amino acids under both 2.8 mM and 16.7 mM glucose conditions (Extended Data Table 1).

The concentrations of these amino acids were comparable between low and high glucose conditions.

Lines 256–265 in Text

To study the kinetic properties of SNAT2 activation by NERP-4, we determined the concentration dependent uptake of [^{14}C]-MeAIB in the range of 1 μM to 2 mM at 2 min in the presence or absence of NERP-4. [^{14}C]-MeAIB uptake showed a good fit to a Michaelis–Menten curve, with a K_m of $881.1 \pm 220.7 \mu\text{M}$ and a V_{max} of $2,240 \pm 242.3 \text{ pmol/mg}$

protein/min (Fig. 9o). NERP-4 lowered the K_m of [14C]-MeAIB ($412.6 \pm 113.5 \mu\text{M}$) without changing the V_{max} . Furthermore, MeAIB did not alter [125I]-Y-NERP-4[8–19] binding to MIN6-K8 cells (Extended Data Fig. 8f), supporting the finding that the NERP-4 binding site is different from the substrate binding site. Taken together, these results show that NERP-4 is considered to modulate the binding affinity of SNAT2 to amino acids, implying that NERP-4 acts as a positive allosteric modulator (PAM).

Comment: I appreciate the additional data provided by Authors. However, the kinetic analysis provided confirm (Fig. 9o (insert)) that in the micromolar range the difference in transport is very modest (as expected from the interesting K_m -pure change shown by Authors). This renders more important a correct estimation of intracellular content (see below, comments 8 and 9), because even a modest initial velocity change may lead, if sustained by an active transporter such as SNAT2, a significant change in the intracellular content of substrates.

11) It has been known since many years that SNAT2 is insulin-sensitive. This notion is briefly mentioned (line 258) but it should be more extensively discussed. Is there any evidence (original or in literature) that SNAT2 is induced by insulin in beta cells?

Reply: As the reviewer pointed out, insulin induces SNAT2 in skeletal muscle cells, and this enhances amino acid supplies and insulin-dependent protein synthesis (Reference 38 below). We have not found any studies mentioning that SNAT2 is induced by insulin in β cells. We examined whether SNAT2 in β cells was induced by insulin. Insulin upregulated Snat2 mRNA in naïve MIN6-K8 cells and isolated C57BL/6J mouse islets (Extended Data Fig. 9a, b).

Comment: OK. Very interesting.

12) From the results presented, it is tentative to assume two different time schedules for NERP-4 effects. One, rapid, for the stimulation of insulin secretion, attributable to the allosteric mechanism proposed in the manuscript, and the other, slower, for effects on cell viability and metabolism. Under this regard, it would be very interesting to know if NERP-4 stimulates the expression of SNAT2? This is an easy question but, apparently, there is no

clear answer. The only data are, apparently, those in Fig. 10, where, however, NERP-4 effects are seen in palmitate-treated islets (again the experiment in control conditions is missing) or in db mice. What about human cells and normal conditions or models?

Reply: As summarized well by the reviewer, we hypothesize that the effects of NERP-4 occur during two different phases. In the rapid phase, NERP-4 stimulates amino acid uptake into β cells, thereby enhancing insulin secretion. In the slower phase, NERP-4 acts on β -cell viability and metabolism. To determine whether NERP-4 stimulates SNAT2 expression under both normal and stressed conditions in the slower phase, we performed new experiments. We demonstrated that NERP-4 upregulated *Snat2* mRNA in naïve MIN6-K8 cells and isolated C57BL/6J mouse islets under non-stressed conditions (Extended Data Fig. 9c, d). NERP-4 also upregulated *Snat2* mRNA in palmitate- or cytokine-treated MIN6-K8 cells (Extended Data Fig. 9e, f). These findings indicate that NERP-4 stimulates SNAT2 expression under normal and stressed conditions. We added these results as shown below. We were unable to analyze human islets because relatively few pancreatic transplants are currently being performed.

Comment: OK, very interesting.

13) If Gln uptake in beta cells were mainly due to SNAT2 operation (but this, however, awaits experimental demonstration), then the effects of SNAT2 knockdown (Fig. 10) would be those generically expected for an amino acid deprivation response rather than specifically attributes to an interference with NERP-4 effects.

Reply: As the reviewer mentioned, amino acid uptake into β cells is carried out redundantly by several transporters. SNAT2, SNAT4, ASC-1 (SLC1A10), ASCT2 (SLC1A5), and B0AT1 (SLC6A19), all of which are expressed in β cells, transport glutamine and alanine (Reference 5 below). To explore whether glutamine and alanine uptake in β cells is mainly due to SNAT2, we studied amino acid uptake in MIN6-K8 cells treated by *Snat2* knockdown or MeAIB administration. *Snat2* knockdown reduced glutamine and alanine uptake by 13.6% and 30.3%, respectively, compared with control MIN6-K8 cells (Fig. 9e, f). MeAIB also reduced glutamine and alanine uptake by 17.8% and 40.7%, respectively (Fig. 9g, h). These results suggest that transporters other than SNAT2 also mediate glutamine and alanine uptake. We thus think that SNAT2 deletion and inhibition attributed

to an interference with NERP-4 effects.

Comment: OK

Minor

1), 2), 3)

Comment: OK

Reviewer #2 (Remarks to the Author):

Authors satisfactorily addressed most of my comments. The revised manuscript is much stronger and, easier to read.

There are still 2 aspects that I believe would benefit some clarification.

1. New data clearly show that NERP4 increases VGF expression. Thus, it is possible that some of the effects are mediated by other VGF-derived peptides being released and acting in an autocrine/paracrine manner on the beta cells. The SNAP2 KD/inhibition experiments support a specific mechanism of NERP4; yet, basal aminoacid uptake is also diminished by the SNAT2 KD and pharmacological inhibitor, so the story might be more complicated. Unfortunately, it is impossible at present to block the effect of all VGF peptides (and VGF KO will affect beta cell function, so it will not be a good experiment). Perhaps, the addition of a set of experiments in which cells are stimulated with NERP4 in the presence/absence of the C3aR1 antagonist SB290157, which has been shown to inhibit TLQP-21-mediated biological activity (including calcium influx from the extracellular compartment) in 3T3-L1 and other cells, would be a nice addition.

Alternatively, this possible indirect effect mediated by other VGF peptides should be addressed more strongly through the manuscript, and the potential for TLQP-62 and AQEE-30 (via an unknown mechanism), TLQP-21 (via C3aR1 activation) or other NERP peptides identified by this group should be clarified.

2. New pharmacological experiments add support to the specificity of SNAT2 as the binding

partner of NERP4. However, I'm still not very convinced this is unique, and that NRP4 is a PAM. I understand that the experiments required to narrow this down 100% would be extremely complex and open ended. Thus, my suggestion is simply to reduce emphasis and major conclusions on this aspect.

This concern is justified by: i) the absence of direct molecular evidence of the mode of binding, and the position of the the allosteric binding site; ii) it is not a general and expected result that allosteric modulators change affinity of orthosteric modulators <https://pubmed.ncbi.nlm.nih.gov/35298129/>. Thus, the competitive binding experiments only provide indirect support; iii) 2 proteins, SNAT2 and SCRB2 showed comparable binding specificity. Yet, the study neglects to investigate SCRB2. iii) SNAT2 KD or pharmacological inhibition appear to affect at least some basal beta cell functions, thus partially limiting the validity of these loss of function approaches to validate the specificity of NERP4-SNAT2 interaction.

Reviewer #3 (Remarks to the Author):

The authors revised their paper on the mechanisms utilized by NERP-4, which increased Ca²⁺ influx in pancreatic pieces and Min6 cells and glucose-induced insulin secretion. The results are novel and the manuscript is much improved. I commend the authors for their thorough revision of the manuscript according to the critiques of the reviewers.

I do have reservations still about the use of islet extracts or pieces of minced pancreas, though, as this is not a standard preparation in the field. The authors should move on from this to make proper isolated islets like the rest of the field as this would remove concern about possible roles for the exocrine tissue remaining in such extracts.

Dear Reviewers of *Nature Communications*,

We sincerely appreciate your careful reading of the revised manuscript and your valuable comments and advice. We have expanded several experiments and have provided details in the current version of the manuscript. Our responses to the reviewers' comments are listed below. We added Dr. Valery Gmyr to the author list because of his contribution to the revised experiments. The minor modifications in the revised manuscript can also be found on Line 3, Lines 13–14, Lines 208–210, Lines 210–213, Lines 335–337, Line 367, and Line 410, as follows.

Line 3

Yuichiro Mita^{2, 14}, Ryota Tanida^{2, 15}, Satoshi Hirako³, Seiji Shioda⁴, **Valery Gmyr⁵**,

Lines 13–14

⁵ **Université de Lille, Inserm, Campus Hospitalo-Universitaire de Lille, Institut Pasteur de Lille, U1190-EGID, F-59000 Lille, France.**

Lines 208–210

We **therefore investigated** SNAT2, a neutral amino acid transporter that enhances **amino acid** uptake into β cells to stimulate insulin secretion⁵.

Lines 210–213

We studied the binding of NERP-4 to SNAT2. [¹²⁵I]-Y-NERP-4[8–19] bound to SNAT2-overexpressing HEK293 cells in a concentration-dependent manner (Fig. 8d, Extended Data Fig. 7). [¹²⁵I]-Y-NERP-4[8–19] binding was reduced by the addition of excessive NERP-4 (Fig. 8d), suggesting that the binding was specific for NERP-4.

Lines 335–337

The only endogenous PAM that has been identified thus far is the glutamate transporter–associated protein 3-18 (GTRAP3-18), which acts directly on an AAT named excitatory amino acid carrier 1 (EAAC1)⁵⁰.

Line 367

Identifying a novel mechanism whereby the peptide–amino acid transporter axis regulates amino acid transport could increase our understanding of how amino acids regulate β -cell biology.

Line 410

Minamino N, Gmyr V, Kerr-Conte JA, and Pattou F conducted the experiments and analysed the data;

REVIEWER COMMENTS

Reviewer #1 (Remarks to the Author):

Authors have done a great work in addressing my remarks, adding new, and very interesting, data, in particular on transport studies.

While most of my concerns have been successfully overcome, I still have some doubts on the mechanistic aspects (see my comments 8-9-10). In summary, I think that the results now clearly indicate a major (and possibly exclusive) role of SNAT2 in the effect but I wonder if it is due to the metabolic effect of the transported substrates (I doubt that this is likely, see my comments 8-9-10, and lack of data on possible changes in the intracellular content of amino acids) or to some transduction pathway activated by the transporter (as suggested in other contexts, see the concept of transceptors proposed by Hundal's group several years ago).

I'm going through my original remarks and authors' rebuttal.

Reply:

We sincerely appreciate the reviewer's careful reading of our revised manuscript and your valuable comments and advice. We measured the intracellular contents of amino acids under the presence or absence of NERP-4. NERP-4 increased the contents of glutamine, alanine, proline, and glycine. As the reviewer noted, Dr. Hundal's group proposed the concept of transceptors, which transport amino acids and activate transduction pathways to support cell growth and proliferation⁶¹⁻⁶³. We have mentioned in the Discussion, as shown below, that SNAT2 possibly acts as a transceptor.

Lines 346–354 in the Discussion

In addition to its role in amino acid transfer, SNAT2 possesses a transceptor-like function⁶¹. It detects whether there are sufficient amounts of the amino acids that regulate its expression and stability^{61, 62}. SNAT2-mediated Na⁺ incorporation plays a potential role in membrane depolarisation and increases

both intracellular Ca^{2+} levels and insulin secretion⁵⁹. SNAT2 activation can also activate several transduction pathways that support cell growth and proliferation^{61–63}. In the present study, NERP-4 increased SNAT2 expression, suggesting that NERP-4 not only contributes to the transport of Na^+ and amino acids via SNAT2, but also increases SNAT2 expression that can enhance its downstream signalling.

Reference 61. Hyde, R, Cwiklinski, E. L., MacAulay, K., Taylor, P. M. & Hundal, H. S. Distinct sensor pathways in the hierarchical control of SNAT2, a putative amino acid transceptor, by amino acid availability. *J. Biol. Chem.* **282**, 19788–19798 (2007).

Reference 62. Hoffmann, T. M., et al. Effects of Sodium and Amino Acid Substrate Availability upon the Expression and Stability of the SNAT2 (SLC38A2) Amino Acid Transporter. *Front. Pharmacol.* **9**, 63 (2018).

Reference 63. Pinilla, J. et al. SNAT2 transceptor signalling via mTOR: a role in cell growth and proliferation? *Front Biosci (Elite Ed)*. **3**, 1289–1299 (2011).

Reference 59. Turbitt J., et al. NKCC transport mediates the insulinotropic effects of taurine and other small neutral amino acids. *Life Sci.* **316**, 121402 (2023).

Remarks 1)-2)-3)-4)-5)-6)

Comment: Ok.

7) SNAT2 is claimed to be operative in glutamine and alanine (+ cysteine) transport. Yet, the specificity of the transporter is much larger. For instance, a preferential natural substrate of the transporter is Pro, which poorly interacts with most of the other transporters for neutral amino acids, thus constituting a reliable indicator of SNAT2 activity. Have the authors any data about changes in Pro transport induced by NERP-4?

Reply: As the reviewer mentioned, proline is the preferential natural substrate of SNAT2. We studied the effect of NERP-4 on proline uptake in MIN6-K8 cells. NERP-4 increased proline uptake (Extended Data Fig. 8a), supporting the finding that NERP-4 acts on SNAT2 as a target.

Comment: Interesting novel results. Could they point to a physiological role for Pro?

Reply:

Proline plays key roles in protein structure and function, and also in the maintenance of cellular redox homeostasis⁶⁶. Proline also stimulates insulin secretion through plasma membrane depolarisation as well as glycolytic and oxidative metabolism^{58, 59}. We mentioned the roles of proline on Lines 343–344 and Lines 361–364 of the Discussion section, as follows.

Lines 343–344 in the Discussion

Among the SNAT2 substrates, glutamine, alanine, and proline regulate insulin secretion^{2, 58, 59}.

Lines 361–364 in the Discussion

Proline plays key roles in protein structure and function, and also in the maintenance of cellular redox homeostasis⁶⁶. Proline also stimulates insulin secretion through plasma membrane depolarisation as well as cell glycolytic and oxidative metabolism^{58, 59}.

Reference 66. Vettore, L. A., Westbrook, R. L. & Tennant, D. A. Proline metabolism and redox; maintaining a balance in health and disease. *Amino Acids* **53**, 1779–1788 (2021)

Reference 58. McClenaghan, N. H., Barnett, C. R. & Flatt, P. R. Na⁺ cotransport by metabolizable and nonmetabolizable amino acids stimulates a glucose-regulated insulin-secretory response. *Biochem. Biophys. Res. Commun.* **249**, 299–303 (1998).

Reference 59. Turbitt, J., et al. NKCC transport mediates the insulinotropic effects of taurine and other small neutral amino acids. *Life Sci.* **316**, 121402 (2023).

8) Have the authors any evidence that NERP-4 effectively change the intracellular concentration of amino acids? Have they measured the cell content of Gln (or Ala) in NERP-4-treated models?

Reply: We administered 1 μ M [14C]-L-glutamine, [14C]-L-alanine, or [3H]-L-proline to MIN6-K8 cells in the presence or absence of NERP-4. We determined their intracellular contents 2 min after these administrations. NERP-4 increased the contents of glutamine, alanine, and proline (Fig. 9c, d, Extended Data Fig. 8a). We added the results in the text and in Fig. 9c, d, and Extended Data Fig. 8a. Lines 213–217 in Text

We next investigated the effect of NERP-4 on amino acid transport activity. Human and C57BL/6J mouse islets and MIN6-K8 cells were administered NERP-4 with 1 μ M [14C]-L-glutamine. NERP-4 increased the intracellular contents of [14C]-L-glutamine in both islets and MIN6-K8 cells (Fig. 9a–c).

Next, MIN6-K8 cells were administered NERP-4 with 1 μ M [14C]-L-alanine or 1 μ M [3H]-L-proline. NERP-4 also increased their intracellular contents (Fig. 9d, Extended Data Fig. 8a).

Comment: I must consider authors' reply unsatisfactory. What they show is not the intracellular concentration or content of the amino acids but, rather, the 2-, 5-, or 10-min uptake of the labelled substrates. Since they measure in another part of the study the extracellular levels of amino acids (see "Amino acid determination" in Methods) , it should be easily feasible to determine if NERP-4 effectively changes the cell content of amino acids. Alternatively, I suggest that they should replace "intracellular contents" with "2-, 5- or 10-min uptake", as specified in the experiment. However, from a mechanistic point of view, it would be important (see below, comments 9 and 10) to know if NERP-4 increase the intracellular amino acid content and if the increase is restricted or not to SNAT2 substrates.

Reply to comments 8–10:

We quantified the amounts of amino acids in MIN6-K8 cells under high glucose with or without NERP-4. NERP-4 increased the intracellular contents of the SNAT2 substrates glutamine, alanine, proline, glycine, threonine, and serine. We described this result on Lines 225–228 of the text and on Lines 364–365 in the Discussion section, and added Extended Data Fig. 8b and Extended Data Table

1. We also replaced "intracellular contents" with "uptake" on Lines 223–225 of the text.

Lines 223–228 in the text

NERP-4 increased [^{14}C]-L-glutamine uptake in both islets and MIN6-K8 cells (Fig. 9a–c). Next, we administered NERP-4 with 1 μM [^{14}C]-L-alanine or 1 μM [^3H]-L-proline to MIN6-K8 cells. NERP-4 also increased their uptake (Fig. 9d, Extended Data Fig. 8a). NERP-4 increased the intracellular contents of glutamine, alanine, proline, glycine, and glutamic acid in MIN6-K8 cells (Extended Data Fig. 8b, Extended Data Table 1).

Lines 364–365 in the Discussion

In this study, NERP-4 increased the intracellular contents of SNAT2 substrates and stimulated GSIS from MIN6-K8 cells prepared under amino acid-free conditions.

9) The results reported in Fig. 9e are quite expected, and cannot be taken per se as an evidence for a process coupled with insulin secretion. Obviously, when incubated in amino acid-free saline, amino acids go out of cells down their transmembrane gradient. Have the authors any evidence that the amino acid concentrations they measure outside are larger in conditions of insulin secretion stimulated (e.g. high glucose) compared to control (low glucose)? Moreover, an interesting functional proof of their model would be to demonstrate that this efflux is stimulated by NERP-4. Have the authors checked other amino acids? In other words, the three amino acid reported are the only released?

Reply: We quantified amino acids in the media after GSIS experiments under low- (2.8 mM) or high- (16.7 mM) glucose conditions. We detected glutamine, alanine, proline, glutamate, leucine, arginine and other amino acids at similar concentrations in both low- and high-glucose media (Extended Data Table 1a). Glutamine, glutamate, alanine, leucine, and arginine were previously shown to stimulate insulin secretion (Reference 2 below). NERP-4 did not augment insulin secretion from isolated C57BL/6J mouse islets under 2.8 mM glucose, even when amino acids were present in the medium (Fig. 2b, c). Another research group demonstrated that 2 mM glutamine administration to MIN6-K8 cells under 2.8 mM glucose did not increase intracellular Ca²⁺ and had no effect on GSIS (Reference 23 below). Combined together, these findings indicate that under high-glucose conditions, NERP-4 functions as an insulinotropic peptide by enhancing the uptake of amino acids released from β cells. NERP-4 does not act on SNAT3 or SNAT5 which stimulates the efflux of glutamine from β cells. We thus consider that glutamine efflux is independent of NERP-4. We added these results in the text and in Extended Data Table 1.

Lines 243–254 in Text

We used amino acid-free HKRB buffer in GSIS experiments. SNAT3, SNAT5, and ASCT2 function as exporters of glutamine under conditions in which extracellular glutamine is absent or very low. To explore the source of amino acids that stimulate NERP-4-induced GSIS, we quantified amino acids released into the medium after GSIS experiments. MIN6-K8 cells released glutamine, alanine, proline, and other amino acids under both 2.8 mM and 16.7 mM glucose conditions (Extended Data Table 1). The concentrations of these amino acids were comparable between low and high glucose conditions. NERP-4 did not augment Ca²⁺ influx or GSIS under 2.8 mM glucose even when amino acids were present in the medium (Fig. 2b, c, Extended Data Fig. 1a). Another research group demonstrated that 2 mM glutamine administration to MIN6-K8 cells in medium containing 2.8 mM glucose did not affect GSIS or Ca²⁺ influx²³. Together, these findings indicate that under high glucose conditions, NERP-4 functions as an insulinotropic peptide by enhancing uptake of amino acids released from β cells.

Reference 2. Newsholme, P., Bender, K., Kiely, A. & Brennan, L. Amino acid metabolism, insulin secretion and diabetes. *Biochem. Soc. Trans.* 35, 1180–1186 (2007).

Reference 23. Han G., et al. Glutamate is an essential mediator in glutamine-amplified insulin secretion. *J. Diabetes Investig.* 12, 920–930 (2021).

Comment: The experiment performed confirms that the concentration of the amino acids released by

the cells is in the micromolar range (see below, comment 10).

10) More importantly, authors state that NERP-4 stimulates GSIS even under amino acid-free conditions (line 250-251). Thus, how can SNAT2-mediated Gln transport works under those conditions? It is very unlikely that the extremely low quantities of Gln released from cells (in the order of microM) can justify a significant influx mediated by SNAT2 (whose K_m for Gln is > 1 mM).

Reply: To respond to this important comment, we added the following experiments. First, we verified that amino acids released from β cells are imported through the activity of the NERP-4–SNAT2 axis. We administered 1 μ M [14C]-L-glutamine, [14C]-L-alanine, or [3H]-L-proline to MIN6-K8 cells in the presence or absence of NERP-4, and 2 min later determined their intracellular contents. NERP-4 increased the intracellular contents of these amino acids (Fig. 9c, d, Extended Data Fig. 8a). These findings imply that low quantities of amino acids (i.e., 1 μ M concentration) are imported into β cells by NERP-4. Next, we verified that NERP-4–induced amino acid uptake mediated Ca^{2+} influx and GSIS under amino acid–free conditions. Snat2 knockdown abolished NERP-4–induced glutamine and alanine uptake (Fig. 9e, f), Ca^{2+} influx (Fig. 9i), and GSIS (Fig. 9k). MeAIB also abrogated NERP-4–induced glutamine and alanine uptake (Fig. 9g, h), Ca^{2+} influx (Fig. 9j), and GSIS (Fig. 9l). We conclude that NERP-4 acts on SNAT2 to cause amino acid uptake that induces Ca^{2+} influx and GSIS. We further analysed the kinetic properties of concentration-dependent uptake of [14C]-MeAIB in the range of 1 μ M to 2 mM in the presence or absence of NERP-4. [14C]-MeAIB uptake showed a good fit to a Michaelis–Menten curve, with a K_m of 881.1 ± 220.7 μ M and a V_{max} of $2,240 \pm 242.3$ pmol/mg protein/min (Fig. 9o). NERP-4 administration lowered the K_m of [14C]-MeAIB (412.6 ± 113.5 μ M) without changing the V_{max} , implying that NERP-4 modulates the binding affinity of SNAT2 to amino acids. As we showed in the original manuscript, MeAIB, an orthosteric inhibitor of SNAT2, did not change [125I]-Y-NERP-4[8–19] binding to MIN6-K8 cells (Extended Data Fig. 8f). This supports the finding that the NERP-4 binding site is different from the substrate binding site. Combined with the kinetics analysis and binding study, NERP-4 acts as a positive allosteric modulator (PAM). Lastly, we confirmed that NERP-4 augmented GSIS in the presence of μ M-order concentrations of glutamine in MIN6-K8 cells. One micromolar glutamine increased GSIS, and coadministration

of NERP-4 and 1 μ M glutamine further increased GSIS (Extended Data Fig. 8c). We

thus think that the impact of NERP-4 on SNAT2 is significant under low quantities of glutamine.

We added these results in Extended Data Table 1, Extended Data Fig. 8a, Fig. 9e, f, i–k, Lines 220–222 in Text

Snat2 knockdown in MIN6-K8 cells abrogated NERP-4–induced [14C]-L-glutamine and [14C]-L-alanine uptake in MIN6-K8 cells (Fig. 9e, f, Extended Data Fig. 8b).

Lines 236–242 in Text

Glutamine concentrations equal to or higher than 1 μ M increased GSIS in a concentration-dependent manner (Extended Data Fig. 8c). NERP-4 augmented GSIS at glutamine concentrations between 1 μ M

and 100 μM , but not at 1 mM glutamine (Extended Data Fig. 8c). Both Snat2 knockdown and MeAIB abrogated NERP-4-induced Ca^{2+} influx (Fig. 9i, j, Extended Data Fig. 8d) and GSIS (Fig. 9k, l) in MIN6-K8 cells. MeAIB also abolished NERP-4-induced GSIS in human and C57BL/6J mouse islets (Fig. 9m, n). These findings reveal that NERP-4 acts specifically on SNAT2 to induce amino acid uptake, Ca^{2+} influx, and GSIS.

Lines 245–249 in Text

To explore the source of amino acids that stimulate NERP-4-induced GSIS, we quantified amino acids released into the medium after GSIS experiments. MIN6-K8 cells released glutamine, alanine, proline, and other amino acids under both 2.8 mM and 16.7 mM glucose conditions (Extended Data Table 1). The concentrations of these amino acids were comparable between low and high glucose conditions.

Lines 256–265 in Text

To study the kinetic properties of SNAT2 activation by NERP-4, we determined the concentration dependent uptake of [14C]-MeAIB in the range of 1 μM to 2 mM at 2 min in the presence or absence of NERP-4. [14C]-MeAIB uptake showed a good fit to a Michaelis–Menten curve, with a K_m of $881.1 \pm 220.7 \mu\text{M}$ and a V_{max} of $2,240 \pm 242.3 \text{ pmol/mg protein/min}$ (Fig. 9o). NERP-4 lowered the K_m of [14C]-MeAIB ($412.6 \pm 113.5 \mu\text{M}$) without changing the V_{max} . Furthermore, MeAIB did not alter [125I]-Y-NERP-4[8–19] binding to MIN6-K8 cells (Extended Data Fig. 8f), supporting the finding that the NERP-4 binding site is different from the substrate binding site. Taken together, these results show

that NERP-4 is considered to modulate the binding affinity of SNAT2 to amino acids, implying that NERP-4 acts as a positive allosteric modulator (PAM).

Comment: I appreciate the additional data provided by Authors. However, the kinetic analysis provided confirm (Fig. 9o (insert)) that in the micromolar range the difference in transport is very modest (as expected from the interesting K_m -pure change shown by Authors). This renders more important a correct estimation of intracellular content (see below, comments 8 and 9), because even a modest initial velocity change may lead, if sustained by an active transporter such as SNAT2, a significant change in the intracellular content of substrates.

Reviewer #2 (Remarks to the Author):

Authors satisfactorily addressed most of my comments. The revised manuscript is much stronger and, easier to read.

There are still 2 aspects that I believe would benefit some clarification.

1. New data clearly show that NERP4 increases VGF expression. Thus, it is possible that some of the effects are mediated by other VGF-derived peptides being released and acting in an autocrine/paracrine manner on the beta cells. The SNAP2 KD/inhibition experiments support a specific mechanism of NERP4; yet, basal aminoacid uptake is also diminished by the SNAT2 KD and pharmacological inhibitor, so the story might be more complicated. Unfortunately, it is impossible at present to block the effect of all VGF peptides (and VGF KO will affect beta cell function, so it will not be a good experiment). Perhaps, the addition of a set of experiments in which cells are stimulated with NERP4 in the presence/absence of the C3aR1 antagonist SB290157, which has been shown to inhibit TLQP-21-mediated biological activity (including calcium influx from the extracellular compartment) in 3T3-L1 and other cells, would be a nice addition.

Alternatively, this possible indirect effect mediated by other VGF peptides should be addressed more strongly through the manuscript, and the potential for TLQP-62 and AQEE-30 (via an unknown mechanism), TLQP-21 (via C3aR1 activation) or other NERP peptides identified by this group should be clarified.

Reply:

We sincerely appreciate the reviewer's careful reading of the revised manuscript and your valuable comments and advice. In the second revision of our manuscript, we have clarified that the VGF-derived peptides TLQP-21, TLQP-62, NERP-2, and AQEE-30 influence β -cell behaviour. We agree with the reviewer's comment that *Vgf* KO affects β -cell function, as *Vgf* knockdown in MIN6-K8 cells in this study suppressed insulin secretion (Extended Data Fig. 2b).

We here confirmed that the C3aR1 antagonist SB290157 blocked TLQP-21-induced insulin secretion (Fig. A, bottom). NERP-4 administration recovered ROS production as well as palmitate-

induced suppression of GSIS and ATP production (Fig. 3b; Fig. 4c, g; Extended Data Fig. 4a). We investigated the possible involvement of TLQP-21 in this setting of the experiments. Co-administration of SB290157 with NERP-4 did not affect GSIS, ATP production, or ROS production (Fig. B–D, bottom). We thought that the pharmacological concentration of NERP-4 used in this study would counteract the effects of other VGF-derived peptides released from MIN6-K8 cells. However, we recognize that VGF-derived peptides whose expression was elevated by NERP-4 administration play a role in β -cell biology. We mentioned the potential effects mediated by other VGF peptides in the Discussion section, as follows.

Lines 313–323 in the Discussion

In the present study, NERP-4 additively enhanced GSIS when administered together with NERP-2, suggesting that the target protein of NERP-4 differs from that of NERP-2. VGF also produces three other insulintropic peptides, namely TLQP-21, TLQP-62[VGF 556–617], and AQEE-30[VGF 588–617]^{19, 43, 44}. Chronic administration of TLQP-21 to prediabetic Zucker diabetic fatty rats preserved islet β -cell mass and slowed diabetes onset¹⁹. *C3aRI* knockout abolished the TLQP-21–induced anti-obesity effect seen in wild mice⁴⁵. TLQP-62 induced a rapid increase in intracellular calcium mobilization, further increasing TLQP-62 secretion in a self-reinforcing manner⁴³. AQEE-30 increased the phosphorylation of Akt and GSK3 β in streptozotocin-treated β cells and suppressed β -cell death⁴⁶. NERP-4 increased *Vgf* mRNA in *db/db* mouse islets, palmitate- or cytokine-treated β cells, and naïve

β cells. VGF-derived peptides being induced by NERP-4 potentially mediate the long-term effects of NERP-4 on β -cell maintenance and function.

Reference 45. Cero, C., et al. The neuropeptide TLQP-21 opposes obesity via C3aR1-mediated enhancement of adrenergic-induced lipolysis. *Mol. Metab.* **6**, 148–158 (2017).

Reference 46. Hirakida, H., et al. VGF nerve growth factor inducible has the potential to protect pancreatic β -cells. *J. Endocrinol.* **257**, e220267 (2023).

A, GSIS from MIN6-K8 cells treated with or without SB290157 ($n = 3$). SB290157 abolished TLQP-21-induced but not NERP-4-induced GSIS. B–D, GSIS (B, $n = 8$), ATP production (C, $n = 6$), and ROS production (D, $n = 8$) in MIN6-K8 cells after 48-h exposure to palmitate and NERP-4 with or without SB290157. Co-administration of SB290157 with NERP-4 did not affect GSIS, ATP production, or ROS production. Results are representative of two independent experiments (A–D). Data are mean \pm s.e.m (A–D). One-way ANOVA and Tukey’s multiple comparisons test (A). Unpaired two-tailed Student’s t -test (B–D).

2. New pharmacological experiments add support to the specificity of SNAT2 as the binding partner of NERP4. However, I'm still not very convinced this is unique, and that NERP4 is a PAM. I understand that the experiments required to narrow this down 100% would be extremely complex and open ended. Thus, my suggestion is simply to reduce emphasis and major conclusions on this aspect. This concern is justified by: i) the absence of direct molecular evidence of the mode of binding, and the position of the the allosteric binding site; ii) it is not a general and expected result that allosteric modulators change affinity of orthosteric modulators <https://pubmed.ncbi.nlm.nih.gov/35298129/>. Thus, the competitive binding experiments only provide indirect support; iii) 2 proteins, SNAT2 and SCRB2 showed comparable binding specificity. Yet, the study neglects to investigate SCRB2. iii) SNAT2 KD or pharmacological inhibition appear to affect at least some basal beta cell functions, thus partially limiting the validity of these loss of function approaches to validate the specificity of NERP4-SNAT2 interaction.

Reply:

We thank the reviewer for the constructive suggestions and feedback that have helped us strengthen our manuscript. As the reviewer pointed out, the topological binding site of SNAT2 with NERP-4 has been unidentified. We referred to research investigating the impact of allosteric modulators on substrate affinity⁵³. We added the following text on Lines 333–335 in the Discussion section: “The binding of allosteric modulators to these target proteins had little impact on either the orthosteric or allosteric binding pockets⁵³.”

SCRB2 is a type III glycoprotein located primarily in limiting membranes of lysosomes and endosomes³⁴. SCRB2 does not interact with extracellular ligands because of its subcellular localisation. We detected NERP-4 binding in the cell membrane protein fraction (Fig. 8c). These findings suggest that SCRB2 is not the target protein of NERP-4. We explained the subcellular localization of SCRB2 and its functions in the endosomal/lysosomal compartment on Lines 200–210 in the text.

We agree with the reviewer's comment that *Snat2* knockdown and pharmacological inhibition attenuated amino acid uptake (Fig. 9e–h). However, *Snat2* knockdown did not affect insulin secretion (Fig. 9k), intracellular ATP (Fig. 10f), ROS production (Fig. 10g), or cell viability (Fig. 10h). MeAIB treatment also did not affect insulin secretion (Fig. 9n). Under preserved β -cell function, both *Snat2* knockdown and MeAIB abolished NERP-4 activities in these five experiments (Fig. 9k, n, Fig. 10f–h). These results imply that the two inhibition experiments could validate the specificity of NERP-4–SNAT2 interaction. Also, on Lines 273–275 in the text and Lines 330–332 in the Discussion section, as shown below, we mentioned that NERP-4 is possibly a PAM.

Lines 200–210 in the text

Only SNAT2 and lysosome membrane protein 2 (SCRIB2) satisfied the criteria for specific binding to TriCEPS (Fig. 8b). We detected NERP-4 binding in the cell membrane protein fraction of SNAT2-overexpressing HEK293 cells with a radioisotope-labelled C-terminally tyrosyl fragment of NERP-4[8–19], [¹²⁵I]-Y-NERP-4[8–19], which was used in the RIA of NERP-4 (Fig. 8c). SCRIB2 is a type III glycoprotein located primarily in limiting membranes of lysosomes and endosomes³⁴. SCRIB2 participates in membrane transportation and the reorganization of the endosomal/lysosomal compartment³⁴. Our findings suggest that SCRIB2 is not the target protein of NERP-4. Amino acid transporters expressed on the cell membrane of β cells regulate β -cell behaviour by mediating amino

acid uptake and release^{3, 4}. We **therefore investigated** SNAT2, a neutral amino acid transporter that enhances **amino acid** uptake into β cells to stimulate insulin secretion⁵.

Lines 273–275 in the text

Taken together, these results show that NERP-4 **may** modulate the binding affinity of SNAT2 to amino acids, implying that NERP-4 **could act** as a positive allosteric modulator (PAM).

Lines 330–332 in the Discussion

Considering the kinetics analysis of the NERP-4–SNAT2 interaction and the lack of MeAIB interference with the binding between NERP-4 and SNAT2, NERP-4 **may act** as a PAM of SNAT2.

Lines 333–335 in the Discussion

The binding of allosteric modulators to these target proteins had little impact on either the orthosteric or allosteric binding pockets⁵³.

Reference 34. Canton, J., Neculai, D. & Grinstein, S. Scavenger receptors in homeostasis and immunity. *Nat. Rev. Immunol.* **13**, 621–634 (2013).

Reference 53. Chen, C. J., et al. How do modulators affect the orthosteric and allosteric binding Pockets? *ACS Chem. Neurosci.* **13**, 959–977 (2022).

Reviewer #3 (Remarks to the Author):

The authors revised their paper on the mechanisms utilized by NERP-4, which increased Ca²⁺ influx in pancreatic pieces and Min6 cells and glucose-induced insulin secretion.

The results are novel and the manuscript is much improved. I commend the authors for their thorough revision of the manuscript according to the critiques of the reviewers.

I do have reservations still about the use of islet extracts or pieces of minced pancreas, though, as this is not a standard preparation in the field. The authors should move on from this to make proper isolated islets like the rest of the field as this would remove concern about possible roles for the exocrine tissue remaining in such extracts.

Reply:

We thank the reviewer for their constructive suggestions and feedback that have helped us strengthen our manuscript. Aequorin-expressing tissues from apoaequorin transgenic mice were used to detect elevations in intracellular Ca²⁺ concentrations by ligands of interest, such as novel compounds and peptides¹⁰. We used pancreatic pieces because the islets would lose their original function once they were incubated with coelenterazine for 3 h. As the reviewer pointed out, some substances could activate pancreatic exocrine tissue. We therefore used a Fura-2 fluorescence assay to confirm that NERP-4 stimulated Ca²⁺ mobilization into β cells (Fig. 1c; Fig. 9i, j; Extended Data Fig. 8g). We mentioned the luminescence assay in aequorin-expressing tissues in the text as follows.

Lines 85–88 in the text

Aequorin-expressing tissues from apoaequorin transgenic mice were used to detect the elevation of intracellular Ca^{2+} concentrations by ligands of interest, such as novel compounds and peptides¹⁰. We used pancreatic pieces because the islets would lose their original function once they were incubated with coelenterazine for 3 h.

Reference 10 . Yamano, K., et al. Identification of the functional expression of adenosine A3 receptor in pancreas using transgenic mice expressing jellyfish apoaequorin. *Transgenic Res.* **16**, 429–435 (2007).

REVIEWERS' COMMENTS

Reviewer #1 (Remarks to the Author):

I think that authors have adequately addressed most of my remaining concerns and that the study has now markedly improved.

Reviewer #2 (Remarks to the Author):

Overall, Authors satisfactorily addressed my additional comments. This paper is a relevant contribution to the field!

Reviewer #3 (Remarks to the Author):

I am satisfied that the authors did a very reasonable job responding to my specific critiques related to intracellular free Ca measurements. I think the new data support their conclusions as they predicted.

Dear Reviewers of *Nature Communications*,

We sincerely appreciate your careful reading of the revised manuscript and your valuable comments. The manuscript has been subjected to revision carefully and accordingly. Our responses to the reviewers' comments are listed below.

REVIEWERS' COMMENTS

Reviewer #1 (Remarks to the Author):

I think that authors have adequately addressed most of my remaining concerns and that the study has now markedly improved.

Reply:

We sincerely thank the reviewer for the constructive comments and advice in strengthening our manuscript.

Reviewer #2 (Remarks to the Author):

Overall, Authors satisfactorily addressed my additional comments. This paper is a relevant contribution to the field!

Reply:

We sincerely thank the reviewer for raising many valuable points that have helped us strengthen our manuscript.

Reviewer #3 (Remarks to the Author):

I am satisfied that the authors did a very reasonable job responding to my specific critiques related to intracellular free Ca measurements. I think the new data support their conclusions as they predicted.

Reply:

We sincerely thank the reviewer for the helpful feedback, which has substantially strengthened our manuscript.